# COMMUNICATION-EFFICIENT HETEROGENEOUS FEDERATED LEARNING WITH GENERALIZED HEAVY-BALL MOMENTUM

## ABSTRACT

Federated Learning (FL) has emerged as the state-of-the-art approach for learning from decentralized data in privacy-constrained scenarios. However, system and statistical challenges hinder real-world applications, which demand efficient learning from edge devices and robustness to heterogeneity. Despite significant research efforts, existing approaches (i) are not sufficiently robust, (ii) do not perform well in large-scale scenarios, and (iii) are not communication efficient. In this work, we propose a novel *Generalized Heavy-Ball Momentum* (GHBM), proving that it enjoys an improved theoretical convergence rate w.r.t. existing FL methods based on classical momentum in *partial participation*, without relying on bounded data heterogeneity. Then, we present FEDHBM as an adaptive, communication-efficient by-design instance of GHBM. Extensive experimentation on vision and language tasks, in both controlled and realistic large-scale scenarios, confirms our theoretical findings, showing that GHBM substantially improves the state of the art, especially in large scale scenarios with high data heterogeneity and low client participation [1].

## 1 INTRODUCTION

Federated Learning (FL) (McMahan et al., 2017) is a paradigm to learn from decentralized data in which a central server orchestrates an iterative two-step training process that involves 1) local training, potentially on a large number of clients, each with its own private data, and 2) the aggregation of these updated local models on the server into a single, shared global model. This process is repeated over several communication rounds. While the inherent privacy-preserving nature of FL is appealing for decentralized applications where data sharing is restricted, it also introduces some challenges. Since local data reflects characteristics of individual clients, limiting the optimization to use only the client's data can lead to issues caused by *statistical heterogeneity*. This becomes problematic when multiple optimization steps are performed before models are synchronized, causing clients to *drift* away from the ideal global updates (Karimireddy et al., 2020). Indeed, heterogeneity has been shown to hinder the convergence of FEDAVG (Hsu et al., 2019), increasing the number of communication rounds to reach a target model quality (Reddi et al., 2021) and impacting final performance.

Several studies have proposed solutions to mitigate the effects of heterogeneity. For instance, SCAFFOLD (Karimireddy et al., 2020) relies on additional control variables to correct the local client's updates, while FEDDYN (Acar et al., 2021) uses ADMM to align the global and local client solutions. Albeit theoretically grounded, experimentally these methods are not sufficiently robust to handle cases of extreme heterogeneity, low client participation, or large-scale problems, exhibiting slow convergence and instabilities (Varno et al., 2022; Reddi et al., 2021).

Momentum-based FL methods appear to be a promising solution for addressing these challenges. By accumulating past update directions, momentum can help clients overcome the inconsistencies of local objectives introduced by heterogeneous data. Several works explored incorporating momentum in FL, either at the server (Hsu et al., 2019) or at client-level to correct local updates Ozfatura et al. (2021); Xu et al. (2021). Notably, MIME (Karimireddy et al., 2020; 2021) has been proposed as a framework to make clients mimic the updates of a centralized model trained on i.i.d. data by leveraging extra server statistics at client side. While the theoretical benefits of momentum in FL have been demonstrated in *full participation* Cheng et al. (2024), existing FL methods based

---

[1]Code is provided for the review process and will be released upon acceptance

solely on momentum still theoretically rely on *bounded heterogeneity* in *partial participation* and, as our work demonstrates, experimentally present limitations in settings with low participation, high heterogeneity and real-world large-scale problems. Moreover, current approaches often incur increased communication costs due to the exchange of additional information required to correct local updates (Karimireddy et al., 2020; 2021; Xu et al., 2021; Ozfatura et al., 2021). This can be a significant drawback in communication-constrained environments, further hindering the practical adoption of FL in real-world scenarios and underscoring the critical need for more robust, effective, and communication-efficient FL algorithms.

In this work, we present both empirical evidence and theoretical justification for the failure cases of previous momentum-based FL approaches. Specifically, we demonstrate that the interplay of data heterogeneity and partial participation is not properly addressed, leading to the classical momentum term employed in these methods to be updated with a biased estimate of the global gradient, which diminishes its effectiveness in correcting client drift. To address these challenges, we propose a novel *Generalized Heavy-Ball* (GHBM) formulation of momentum that consists of calculating it as a decayed average of $\tau$ past momentum terms. This design choice makes the momentum term not to be biased towards most recently selected clients, and allows GHBM to converge under arbitrary heterogeneity even in partial participation. Then, we present FEDHBM as an adaptive, communication-efficient by-design instance of GHBM, and show experimentally significantly improved performance over the state of the art.

**Contributions.** We summarize our main results below.

- We present a novel formulation of momentum called *Generalized Heavy-Ball* (GHBM) momentum, which extends the classical heavy-ball (Polyak, 1964), and propose variants that are robust to heterogeneity and communication-efficient by design.
- We establish the theoretical convergence rate of GHBM for non-convex functions, extending the previous result of Cheng et al. (2024) of classical momentum, showing that GHBM converges under arbitrary heterogeneity even (and most notably) in *partial participation*.
- We empirically show that existing FL algorithms suffer severe limitations in extreme non-iid scenarios and real-world settings. In contrast, FEDHBM is extremely robust and achieves higher model quality with significantly faster convergence speeds than other client-drift correction methods.

## 2 RELATED WORKS

**The problem of statistical heterogeneity.** The detrimental effects of non-iid data in FL were first observed by (Zhao et al., 2018), who proposed mitigating performance loss by broadcasting a small portion of public data to reduce the divergence between clients' distributions. Alternatively, (Li & Wang, 2019) uses server-side public data for knowledge distillation. Both approaches rely on the strong assumption of readily available and suitable data. Recognizing weight divergence as a source of performance loss FEDPROX (Li et al., 2020) adds a regularization term to penalize divergence from the global model. Nevertheless, this was proved ineffective in addressing data heterogeneity Caldarola et al. (2022). Other works (Kopparapu & Lin, 2020; Zaccone et al., 2022; Zeng et al., 2022; Caldarola et al., 2021) explored grouping clients based on their data distribution to mitigate the challenges of aggregating divergent models.

**Stochastic Variance Reduction in FL.** Stochastic variance reduction techniques have been applied in FL (Chen et al., 2021; Li et al., 2019) with SCAFFOLD Karimireddy et al. (2020) providing for the first time convergence guarantees for arbitrarily heterogeneous data. The authors also shed light on the *client-drift* of local optimization, which results in slow and unstable convergence. SCAFFOLD uses control variates to estimate the direction of the server model and clients' models and to correct the local update. This approach requires double the communication to exchange the control variates, and it is not robust enough to handle large-scale scenarios akin to cross-device FL (Reddi et al., 2021; Karimireddy et al., 2021). Conversely, our novel formulation of momentum yields a graceful decay of old and stale gradients while achieving robustness to extreme heterogeneity and low participation. Based on it, we propose an algorithm that does not require any additional data exchange.

**ADMM and adaptivity.** Other methods are based on the Alternating Direction Method of Multipliers (Chen et al., 2022; Gong et al., 2022; Wang et al., 2022). In particular, FEDDYN(Acar et al., 2021) dynamically modifies the loss function such that the model parameters converge to stationary points of the global empirical loss. Although technically it enjoys the same convergence properties

of SCAFFOLD without suffering from its increased communication cost, in practical cases it has displayed problems in dealing with pathological non-iid settings (Varno et al., 2022). Other works explored the use of adaptivity to speed up the convergence of FedAvg and reduce the communication overhead (Xie et al., 2019; Reddi et al., 2021).

**Use of momentum as local correction.** As a first attempt, Hsu et al. (2019) adopted momentum at server-side to reduce the impact of heterogeneity. However, it has been proven of limited effectiveness under high heterogeneity, because the drift happens at the client level. This motivated later approaches that apply server momentum at each local step (Ozfatura et al., 2021; Xu et al., 2021), and the more general approach by Karimireddy et al. (2021) to adapt any centralized optimizer to cross-device FL. It employs a combination of control variates and server optimizer state (*e.g.* momentum) at each client step, which lead to increased communication bandwidth and frequency. A recent similar approach (Das et al., 2022) employs compressed updates, still requiring significantly more computation client-side. Rather differently from previous works, FEDHBM is based on our novel *Generalized Heavy-Ball Momentum* (GHBM): it consists in a decayed average of the previous $\tau$ momentum terms instead of considering only the last one, and it is designed to more steadily incorporate the descent information of clients selected at past rounds, to be used into local steps as client drift correction. Indeed, the classical heavy-ball (Polyak, 1964) is a special case of GHBM. Remarkably, we show that our formulation is crucial to effectively counteract the effects of statistical heterogeneity and client sampling, and it is communication efficient by design.

**Lowering communication requirements in FL.** Researchers have studied methods to reduce the memory needed for exchanging gradients in the distributed setting, for example by quantization (Alistarh et al., 2017) or by compression (Mishchenko et al., 2019; Koloskova et al., 2020). In the context of FL, such ideas have been developed to meet the communication and scalability constraints (Reisizadeh et al., 2020), and to take into account heterogeneity (Sattler et al., 2020). Our work focuses on the efficient use of the information already being sent in vanilla FEDAVG, so additional techniques to compress that information remain orthogonal to our approach.

## 3 METHOD

### 3.1 SETUP

In FL a server and a set $\mathcal{S}$ of clients collaboratively solve a learning problem, with $|\mathcal{S}| = K \in \mathbb{N}^+$. At each round $t \in [T]$, a fraction of $C \in (0, 1]$ clients from $\mathcal{S}$ is selected to participate to the learning process: we denote this portion as $\mathcal{S}^t \subseteq \mathcal{S}$. Each client $i \in \mathcal{S}^t$ receives the server model $\theta_i^{t,0} \equiv \theta^{t-1}$, and performs $J_i$ local optimization steps, using stochastic gradients $\tilde{g}_i^{t,j}$ evaluated on local parameters $\theta_i^{t,j-1}$ and a batch $d_{i,j}$, sampled from its local dataset $\mathcal{D}_i$. During local training, $\theta_i^{t,j}$ is the model of client $i$ at round $t$ after the $j$-th optimization step, while $\theta_i^t \equiv \theta^{t,J_i}$ is the model sent back to the server. The server then aggregates the client updates $\tilde{g}_i^t := (\theta^{t-1} - \theta_i^t)$, building *pseudo-gradients* (Reddi et al., 2021) $\tilde{g}^t$ that are used to update the model.

### 3.2 ADDRESSING CLIENT DRIFT WITH MOMENTUM

One of the core propositions of federated optimization is to take advantage of local clients' work, by running multiple optimization steps on local parameters before synchronization. This has been proven effective for speeding up convergence when local datasets are i.i.d. with respect to a global distribution (Stich, 2019; Lin et al., 2020; McMahan et al., 2017), and is particularly important for improving communication efficiency, which is the bottleneck when learning in decentralized settings. However, the statistical heterogeneity of clients' local datasets causes local models to *drift* from the ideal trajectory of server parameters. One way of addressing such drift is to use momentum during local optimization, based on the idea that a moving average of past server pseudo-gradients can correct local optimization towards the solution of the global problem. At each round, FL methods based on momentum typically use the gradients of the selected clients, whether computed at local (Xu et al., 2021; Ozfatura et al., 2021) or global (Karimireddy et al., 2021) parameters, to update the momentum term server-side.

**Partial Participation and Biased Momentum.** We claim that existing momentum-based methods overlook a critical aspect of federated learning: *partial client participation*. Indeed, when only a portion of clients participate in the training rounds, the server pseudo-gradient used to update the momentum estimate can be biased towards the previously selected clients, hampering its corrective

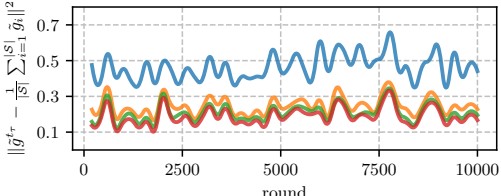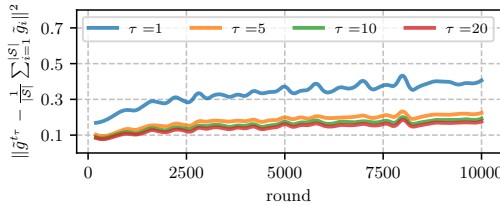

Figure 1: The error of reusing past gradients for updating the momentum, for different values of $\tau$, on CIFAR-100 with RESNET-20, in non-iid ($\alpha = 0$, left) and iid ($\alpha = 10k$, right) settings. The plot shows the empirical measure of the deviation between (i) the average of the last $\tau$ server pseudo-gradient (at different parameters) and (ii) the server-pseudo gradient calculated over all the clients (at the same parameters). Reusing old gradients is beneficial despite the introduced lag.

benefit to local optimization. This effect is particularly pronounced in settings with high data heterogeneity and low client participation (common in cross-device FL), where, as our experiments demonstrate, conventional momentum fails to correct the drift and improve over vanilla FedAvg.

**Main contribution.** To address the challenges posed by partial participation, we propose a novel momentum-based approach that explicitly accounts for client sampling. Our key idea is to update the momentum term using a pseudo-gradient that approximates the true global gradient over all clients, even those not participating in the current round. Our approach effectively mitigates the bias introduced by partial participation by integrating the descent directions from past rounds into local updates resulting in a more accurate and robust momentum estimate. Importantly, our momentum retains a heavy-ball form similar to classical momentum. This can be exploited in FL to avoid sending additional data from server to clients, preserving the same communication complexity as FedAvg.

### 3.3 Generalized Heavy-Ball Momentum (GHBM)

In this section, we introduce our novel formulation for momentum, which we call *Generalized Heavy-Ball Momentum* (GHBM). First, we recall that classical momentum consists of a moving average of past gradients, and it is commonly expressed as in eq. (1), which can be equivalently expressed in a version commonly referred to as *heavy-ball momentum* in eq. (2) (see lemma B.2):

**Heavy-Ball Momentum (HBM)**

$$\tilde{m}^t \leftarrow \beta \tilde{m}^{t-1} + \tilde{g}^t(\theta^{t-1}) \quad (1) \qquad \tilde{m}^t \leftarrow (\theta^{t-1} - \theta^{t-2}) \quad (2)$$
$$\theta^t \leftarrow \theta^{t-1} - \eta \tilde{m}^t \qquad\qquad\qquad \theta^t \leftarrow \theta^{t-1} - \eta \tilde{g}^t(\theta^{t-1}) + \beta \tilde{m}^t$$

Let us notice that, when applied to FL optimization, the gradient referred to above as $\tilde{g}^t$ is built from updates of clients $i \in \mathcal{S}^t$, which are usually a small portion of all the clients participating in the training. Consequently, at each round the momentum is updated using a direction biased towards the distribution of clients selected in that round. Indeed, the prerequisites for this update to reflect the objectives of the other clients are (i) iidness of local datasets or (ii) high client participation. Both conditions are rarely met in practice, and lead to ineffectiveness of existing momentum-based FL methods in realistic scenarios. Our objective is to update the momentum term at each round with a reliable estimate of the gradient w.r.t. the global data distribution of all clients. In practice, the desired update rule for momentum would use the average gradient of all clients selected in the last $\tau$ rounds at current parameters $\theta^{t-1}$, as in eq. (3).

**Desired momentum update**          **Practical momentum update**

$$\tilde{m}^t \leftarrow \beta \tilde{m}^{t-1} + \frac{1}{\tau} \sum_{k=t-\tau+1}^{t} \tilde{g}^k(\theta^{t-1}) \quad (3) \qquad \tilde{m}^t \leftarrow \beta \tilde{m}^{t-1} + \frac{1}{\tau} \sum_{k=t-\tau+1}^{t} \tilde{g}^k(\theta^{k-1}) \quad (4)$$

While eq. (3) cannot be implemented in partial participation because clients selected in rounds $k \in [t - \tau + 1, t)$ do not have access to model parameters $\theta^{t-1}$, it is possible to reuse old gradients calculated at parameters $\theta^{k-1}$ as their approximation, as in eq. (4). This introduces a *lag* due to using old gradients, yet experimentally the effect in heterogeneity reduction greatly compensate, and so the deviation w.r.t. a gradient calculated over all the clients is reduced (see Fig. 1).

With this idea in mind, our proposed formulation consists of calculating the momentum term as the decayed average of past $\tau$ momentum terms, instead of explicitly using the server pseudo-gradients at the last $\tau$ rounds, as shown in eq. (5). This formulation is close to the update rule sketched in eq.

(4) and has the additional advantage of enjoying a heavy-ball form similar to eq. (2) (see lemma B.3), which will be useful to derive communication efficient FL algorithms. In practice, the difference w.r.t. eq. (2) consists ins considering a delta $\tau > 1$ between the two parameters:

**GENERALIZED HEAVY-BALL MOMENTUM (GHBM)**

$$\tilde{m}_\tau^t \leftarrow \frac{1}{\tau} \sum_{k=1}^{\tau} \beta \tilde{m}_\tau^{t-k} + \tilde{g}^t(\theta^{t-1}) \quad (5) \qquad \tilde{m}_\tau^t \leftarrow \frac{1}{\tau} \left( \theta^{t-1} - \theta^{t-\tau-1} \right) \quad (6)$$

$$\theta^t \leftarrow \theta^{t-1} - \eta \tilde{m}_\tau^t \qquad\qquad \theta^t \leftarrow \theta^{t-1} - \eta \tilde{g}^t(\theta^{t-1}) + \beta \tilde{m}_\tau^t$$

As it is trivial to notice, GHBM with $\tau = 1$ recovers the classical momentum, hence it can be considered as a generalized formulation. The GHBM term is then embedded into local updates using the heavy-ball form shown in eq. 6, leading to the following update rule:

$$\text{CLIENT STEP:} \qquad \theta_i^{t,j} \leftarrow \theta_i^{t,j-1} - \eta_l \tilde{g}_i^{t,j}(\theta_i^{t,j-1}) + \underbrace{\hat{\beta} \left( \theta^{t-1} - \theta^{t-\tau-1} \right)}_{\tau-\text{GHBM}} \qquad (7)$$

where $\hat{\beta} := \frac{\beta}{\tau J}$ is the momentum factor scaled by the number of local steps $J$ (see Algorithm 1).

**Discussion on $\tau$.** The $\tau$ hyperparameter in GHBM plays a crucial role, since it controls the number of server pseudo-gradients to average for estimating the update to the momentum term. Intuitively, considering only the effect on heterogeneity reduction, the optimal value is the one that gives the average over all the clients. That value, under proper assumptions on client sampling, is equal to $\tau = 1/C$, which is the inverse of client participation. As we prove, that property is the key factor that enables GHBM to converge under arbitrary heterogeneity, achieving in *partial participation* the same rate that methods based on classical momentum can only obtain by imposing *full participation*. We show this property with a theoretical experiment in Appendix C.9. However, since GHBM reuses old gradients , this introduces a *lag* which grows with $\tau$. For this reason the optimal choice of $\tau$ comes from an inevitable trade-off between the heterogeneity reduction effect and other sources of error we will discuss in Sec. 4.2.

## 3.4 COMMUNICATION COMPLEXITY OF GHBM AND EFFICIENT VARIANTS

---

**Algorithm 1:** GHBM, LOCALGHBM and FEDAVG

---

**Require:** initial model $\theta^0$, $K$ clients, $C$ participation ratio, $T$ number of total round, $\eta$ and $\eta_l$ learning rates.
1: **for** $t = 1$ to $T$ **do**
2: $\quad \mathcal{S}^t \leftarrow$ subset of clients $\sim \mathcal{U}(\mathcal{S}, \max(1, K \cdot C))$
3: $\quad$ Send $\theta^{t-1}, \theta^{t-\tau-1}$ to all clients $i \in \mathcal{S}^t$
4: $\quad$ **for** $i \in \mathcal{S}^t$ **in parallel do**
5: $\qquad \theta_i^{t,0} \leftarrow \theta^{t-1}$
6: $\qquad$ Retrieve $\theta^{t-\tau_i-1}$ from local storage
7: $\qquad \tilde{m}_\tau^t \leftarrow \frac{1}{\tau J}(\theta^{t-1} - \theta^{t-\tau-1})$
8: $\qquad \tilde{m}_{\tau_i}^t \leftarrow \frac{1}{\tau_i J}(\theta^{t-1} - \theta^{t-\tau_i-1})$
9: $\qquad$ **for** $j = 1$ to $J$ **do**
10: $\qquad\quad$ sample a mini-batch $d_{i,j}$ from $\mathcal{D}_i$
11: $\qquad\quad \theta_i^{t,j} \leftarrow \theta_i^{t,j-1} - \eta_l \tilde{g}_i^{t,j} + \beta \tilde{m}_\tau^t + \beta \tilde{m}_{\tau_i}^t$
12: $\qquad$ **end for**
13: $\qquad$ Save model $\theta^{t-1}$ into local storage
14: $\quad$ **end for**
15: $\quad \tilde{g}^t \leftarrow \frac{1}{|\mathcal{S}^t|} \sum_{i=1}^{|\mathcal{S}^t|} \left( \theta^{t-1} - \theta_i^{t,J} \right)$
16: $\quad \theta^t \leftarrow \theta^{t-1} - \eta \tilde{g}^t$
17: **end for**

---

As it is possible to notice from Algorithm 1, GHBM requires the server to additionally send the past model $\theta^{t-\tau-1}$, which is used to calculate the momentum term in eq. (7). Alternatively, the server could send the momentum term $\tilde{m}_\tau^t$ but, in both cases, this introduces a communication overhead of $1.5\times$ w.r.t. FE-DAVG, as momentum is usually applied to all model parameters. However, this overhead can be avoided by leveraging the observation that the choice of $\tau = 1/C$ is expected to be optimal. Indeed, it is sufficient to notice that, if clients participate cyclically, *i.e.* the period between each subsequent sampling is equal for all clients, the frequency at which each client is selected for training is exactly $1/C$. Notice that this is still true on average under uniform client sampling, *i.e.* calling $\tau_i$ the sampling period for client $i$, $\mathbb{E}[\tau_i] = \tau = 1/C$. Leveraging those observations and exploiting the fact that GHBM has an equivalent heavy-ball form, the additional requirement on communication can be traded by allowing clients to maintain persistent storage, and keep the model received by the server across rounds, as shown in Algorithm 1. In this algorithm, that we call **LOCALGHBM**, $\tau_i$ is adaptive and determined stochastically by client participation. The space complexity of LOCALGHBM is constant in the size of model parameters for the clients and recovers the original communication complexity of FedAvg.

Table 1: Comparison of convergence rates of FL algorithms. Our GHBM improves the rate of classical momentum by attaining, in *partial participation*, the same rate of classical momentum in *full participation*. Remind that $L$ is the smoothness constant of objective functions, $\Delta = f(\theta^0) - \min_\theta f(\theta)$ is the initialization gap, $\sigma^2$ is the gradient variance *within* clients, $|\mathcal{S}|$ is the number of clients, $C$ the participation ratio, $J$ is the number of local steps per round, and $T$ is the number of communication rounds. $\zeta = \sup_\theta \|\nabla f(\theta)\|$ and $G^2 := \sup_\theta \frac{1}{|\mathcal{S}|} \sum_{i=1}^{|\mathcal{S}|} \|\nabla f_i(\theta) - \nabla f(\theta)\|^2$ are uniform bounds of gradient norm and gradient dissimilarity.

| Algorithm | Convergence Rate $\frac{1}{T} \sum_{t=1}^{T} \mathbb{E}\left[\|\nabla f(\theta^t)\|^2\right] \lesssim$ | Additional Assumptions | Partial participation? |
|---|---|---|---|
| FEDAVG (Yang et al., 2021) | $\left(\frac{L\Delta\sigma^2}{\|\mathcal{S}\|JT}\right)^{1/2} + \frac{L\Delta}{T}$ | Bounded hetero.[1] | ✗ |
| (Yang et al., 2021) | $\left(\frac{L\Delta\sigma^2}{\|\mathcal{S}\|CJT}\right)^{1/2} + \frac{L\Delta}{T}$ | Bounded hetero.[1] | ✓ |
| FEDCM (Xu et al., 2021)[2] | $\left(\frac{L\Delta(\sigma^2 + \|\mathcal{S}\|CJ\zeta^2)}{\|\mathcal{S}\|CJT}\right)^{1/2} + \left(\frac{L\Delta(\sigma/\sqrt{J} + \sqrt{\|\mathcal{S}\|C}(\zeta+G))}{\sqrt{\|\mathcal{S}\|CT}}\right)^{2/3}$ | Bounded grad. Bounded hetero. | ✓ |
| (Cheng et al., 2024) | $\left(\frac{L\Delta\sigma^2}{\|\mathcal{S}\|JT}\right)^{1/2} + \frac{L\Delta}{T}$ | – | ✗ |
| SCAFFOLD-M (Cheng et al., 2024) | $\left(\frac{L\Delta\sigma^2}{\|\mathcal{S}\|CJT}\right)^{1/2} + \frac{L\Delta}{T}\left(1 + \frac{\|\mathcal{S}\|^{2/3}}{\|\mathcal{S}\|C}\right)$ | – | ✓ |
| **GHBM (Thm. 4.7)** | $\left(\frac{L\Delta\sigma^2}{\|\mathcal{S}\|JT}\right)^{1/2} + \frac{L\Delta}{T}$ | Cyclic participation | ✓ |

[1] The local learning rate vanishes to zero when gradient dissimilarity is unbounded, *i.e.*, $G \to \infty$.
[2] The work has not been published in peer-reviewed venues.

We empirically found that performance can be further improved by considering $\theta_{i,j}^t$ instead of $\theta^{t-1}$ and $\theta_i^{t-\tau_i}$ instead of $\theta^{t-\tau_i-1}$ when calculating $\tilde{m}_{\tau_i}^t$ (see Sec. 5.2). This adds a correction term specific to each client objective, such that it penalizes the direction of the last updates at round $t - \tau_i$ with respect to the progressive updates of local steps at the current round $t$. The final communication-efficient update rule is named **FEDHBM**.

**Applicability of GHBM-based algorithms in FL scenarios.** Albeit based on the same principle, our algorithms are suitable for different scenarios. FEDHBM and LOCALGHBM take advantage of the fact that clients participate multiple times in the training process to remove the need to send the momentum term from the server. As such, clients are *stateful*, as they require maintaining variables across rounds (Kairouz et al., 2021). On the other hand, GHBM has *stateless* clients, which makes it more suitable for cross-device FL or when additional system challenges prevent clients to store state variables. In Sec. 4.2 we analyze such trade-offs from the perspective of optimization, and in Sec. 5.3 we show that they always perform better than the state-of-art.

## 4 THEORETICAL DISCUSSION

In this section, we establish the theoretical foundations of our algorithms. Our analysis reveals that: (i) the momentum update rule implemented by GHBM in eq. (4) approximates an update with global gradient, with $\tau$ controlling the trade-off between heterogeneity reduction and the *lag* due to using old gradients; (ii) thanks to this algorithmic design choice, GHBM converges under arbitrary heterogeneity even in partial participation, whereas FL methods based on classical momentum inevitably require to assume bounded data heterogeneity. The proofs are deferred to Appendix B.

### 4.1 ASSUMPTIONS

For proving our results we rely on notions of stochastic gradient with bounded variance (4.1) and smoothness of the objective functions of the clients (4.2), common in deep learning.

**Assumption 4.1** (Unbiasedness and bounded variance of stochastic gradient)**.**

$$\mathbb{E}_{d_i \sim \mathcal{D}_i}\left[\tilde{g}_i(\theta, d_i)\right] = g_i(\theta, \mathcal{D}_i)$$
$$\mathbb{E}_{d_i \sim \mathcal{D}_i}\left[\|\tilde{g}_i(\theta, d_i) - g_i(\theta, \mathcal{D}_i)\|^2\right] \leq \sigma^2$$

**Assumption 4.2** (Smoothness of client's objectives)**.** Let it be a constant $L > 0$, then for any $i, \theta_1, \theta_2$ the following holds:

$$\|g_i(\theta_1) - g_i(\theta_2)\|^2 \leq L^2 \|\theta_1 - \theta_2\|^2$$

To simplify the problem of determining the clients participating at different rounds, we additionally assume clients participate in a cyclic manner (assumption 4.3). This is a technicality used in the proof and **it is not adopted in the experiments**, where we select clients randomly and uniformly.

**Assumption 4.3** (Cyclic Participation). Let it be $\mathcal{S}^t$ the set of clients participating at any round $t$. A sampling strategy respecting the following is denoted as *"cyclic"* with period $\tau = 1/C$:

$$\mathcal{S}^t = \mathcal{S}^{t-\tau} \qquad \forall\, t > \tau \quad \wedge \quad \mathcal{S}^k \cap \mathcal{S}^t = \varnothing \qquad \forall\, k \in (t-\tau, t)$$

### 4.2 Overcoming bounded gradient dissimilarity in partial participation

In this section, we explain the core elements used in our theory to guarantee convergence under arbitrary heterogeneity for GHBM.

**Bounding the participation-induced heterogeneity.** Let us recall the main idea behind GHBM: because of partial participation, at each round classical momentum is updated using a direction biased towards the distribution of clients selected in that round. Consequently, recalling that GHBM recovers classical momentum when $\tau = 1$, as a first analysis we bound the effect of heterogeneity induced by partial client participation in the estimate of momentum as function of $\tau$.

Let us assume we run federated optimization with one full gradient step in partial participation, and consider the momentum update in eq. (3). Then, the following lemma holds:

**Lemma 4.4** (Deviation of $\tau$-averaged gradient from true gradient). *Define $\mathcal{S}_\tau^t := \cup_{k=0}^{\tau-1}\mathcal{S}^{t-k}$ as the set of clients selected in the last $\tau$ rounds, and $g^{t_\tau} := 1/|\mathcal{S}_\tau^t| \sum_{i=1}^{|\mathcal{S}_\tau^t|} g_i^t(\theta^{t-1})$ as the average server pseudo-gradient. Call $G^2 := \sup_\theta 1/|\mathcal{S}| \sum_{i=1}^{|\mathcal{S}|} \|\nabla f_i(\theta) - \nabla f(\theta)\|^2$ the bound of gradient dissimilarity. The approximation of a gradient over the last $\tau$ rounds $g^{t_\tau}$ w.r.t. the true gradient is quantified by the following:*

$$\mathbb{E}\left[\left\|g^{t_\tau} - \nabla f(\theta^{t-1})\right\|^2\right] \leq 8\mathbb{E}\left[\left(\frac{|\mathcal{S}| - |\mathcal{S}_\tau^t|}{|\mathcal{S}|}\right)^2\right]\left(G^2 + \left\|\nabla f(\theta^{t-1})\right\|^2\right) \tag{8}$$

Lemma 4.4 shows that, as $\tau$ increases, the effect of heterogeneity reduces quadratically as the difference between the $|\mathcal{S}^t|$ and $|\mathcal{S}_\tau^t|$ approaches to zero. While in general determining the exact value of $\tau$ for which this condition is true is a complex problem[2], we can simplify the problem of sampling by assuming clients participate cyclically in the training process and state the following.

**Corollary 4.5.** *Consider lemma 4.4 and further assume that, at each round of FL training, clients are sampled according to a rule satisfying assumption 4.3. Then, for any $\tau \in \left(0, \frac{1}{C}\right]$:*

$$\left\|g^{t_\tau} - \nabla f(\theta^{t-1})\right\|^2 \leq 8\left(1 - \tau C\right)^2\left(G^2 + \left\|\nabla f(\theta^{t-1})\right\|^2\right)$$

Corollary 4.5 shows that, under cyclic participation, the error in lemma 4.4 quadratically decreases as $\tau$ increases until, when $\tau = \frac{1}{C}$, the error is equal to zero, as the two terms in the left-hand side (LHS) of the inequality are the same by definition (*i.e.* the bound of gradient dissimilarity is not necessary).

**Bounding the overall error in momentum update.** In the previous paragraph, we established the role of $\tau$ in GHBM for counteracting heterogeneity and derived its optimal value w.r.t. partial client participation. However, our analysis assumed that all clients selected in the last $\tau$ rounds compute a full gradient on the same server parameters. As discussed in Sec. 3.3, a more realistic update rule for momentum would reuse past gradients as in eq. (4), computed at local parameters. This is because clients selected in rounds $k \in [t - \tau + 1, t)$ do not have access to model parameters $\theta^{t-1}$. Consequently, increasing $\tau$ introduces additional sources of error to the momentum term, quantified in the following lemma.

**Lemma 4.6** (Bounded error of momentum update). *Consider the update rule in eq. (4), and call $\tilde{g}^{t_\tau} = \frac{1}{\tau}\sum_{k=t-\tau+1}^{t}\frac{1}{|\mathcal{S}^k|J}\sum_{i=1}^{|\mathcal{S}^k|}\sum_{j=1}^{J}\tilde{g}_i^{k,j}(\theta_i^{k,j-1})$ the client stochastic pseudo-gradient over the local optimization. Let also define the client drift $\mathcal{U}_t := \frac{1}{|\mathcal{S}|J}\sum_{j=1}^{J}\sum_{i=1}^{|\mathcal{S}|}\mathbb{E}\|\theta_i^{t,j} - \theta^{t-1}\|^2$ and the error of server update $\mathcal{E}_t := \mathbb{E}\|\nabla f(\theta^{t-1}) - \tilde{m}_\tau^{t+1}\|^2$. Under assumptions 4.1-4.2-4.3, it holds that:*

$$\mathbb{E}\left[\left\|\tilde{g}^{t_\tau} - g^{t_\tau}\right\|^2\right] \leq 3\left(\underbrace{\frac{\sigma^2}{|\mathcal{S}_\tau^t|J}}_{\text{(a) noise}} + \underbrace{\frac{L^2}{\tau}\sum_{k=t-\tau+1}^{t}\mathcal{U}_k}_{\text{(b) Client drift}} + \underbrace{2L^2\eta^2\sum_{k=t-\tau+1}^{t-1}\left(\mathbb{E}\left[\left\|\nabla f(\theta^{k-1})\right\|^2\right] + \mathcal{E}_k\right)}_{\text{(c) Gradient lag}}\right)$$

---

[2]Calculating the number of rounds needed to have sampled each client at least once is an instance of the *Batched Coupons Collector* problem (Stadje, 1990; Ferrante & Frigo, 2012; Ferrante & Saltalamacchia, 2014), for which a closed form solution is unknown.

Lemma 4.6 shows that the error affecting the GHBM momentum update rule can be decomposed in three main parts. The first term **(a)** is caused by clients taking stochastic gradients on mini-batched of data: the dependency shows that actually increasing $\tau$ has a positive effect until the gradients of all clients participate to the estimate (*i.e.* $\mathcal{S}_\tau^t = \mathcal{S}$). The second term **(b)** is the average client drift over the last $\tau$ rounds, and it is due to clients performing multiple local steps: the lemma shows a benign dependency, since increasing $\tau$ does not increase the overall error due to this term. The last term **(c)** is the *gradient lag*, that is the error due to using client pseudo-gradients taken at old parameters. While this may be the main source of error since it linearly increases with $\tau$, it depends on $\mathcal{E}_k$, which is the deviation of server update from the true gradient. If momentum succeed in correcting local optimization (*i.e.* $\mathcal{E}_k$ is small), this term will also be small and not hurt the optimization. From experimental verification, this turns out to be the case: the heterogeneity reduction achieved by increasing $\tau$ dominates over the error overall error bounded in lemma 4.6, as showed in Fig. 1, underscoring a notable robustness of this approach.

### 4.3 Convergence guarantees

We can now state the convergence result for GHBM for ***non-convex*** functions in (cyclic) partial participation. Comparison with recent related algorithms is provided in Tab. 1.

**Theorem 4.7.** *Under assumptions 4.1-4.2-4.3, if we take $\tilde{m}_\tau^0 = 0$, and $\beta$, $\eta$ and $\eta_l$ as in eq. (122), then GHBM with $\tau = 1/C$ converges as:*

$$\frac{1}{T}\sum_{t=1}^{T}\mathbb{E}\left[\left\|\nabla f(\theta^{t-1})\right\|^2\right] \lesssim \frac{L\Delta}{T} + \sqrt{\frac{L\Delta\sigma^2}{|\mathcal{S}|JT}}$$

where $\Delta := f(\theta^0) - \min_\theta f(\theta)$ and $\lesssim$ absorbs numeric constants.

**Discussion.** The convergence rate of GHBM matched the best-known rate for FL with non-convex objectives. The notable results we achieve is dismissing the bounded gradient dissimilarity assumption even in partial participation. Moreover, the dominant term on the right-hand side (RHS) scales with the size of all client population $|\mathcal{S}|$, instead of the clients selected in a single round $|\mathcal{S}|C$.

**Comparison with FedCM (Xu et al., 2021).** The best-known rate for FEDCM in partial participation Xu et al. (2021) relies both on bounded gradients and bounded gradient dissimilarity and it is asymptotically weaker than ours. As shown by our analysis, requiring bounded gradient dissimilarity is an intrinsic limit for classical momentum in partial participation, not a limit of their proof technique.

**Comparison with FedCM (Cheng et al., 2024).** In their work authors prove that, by incorporating (classical) momentum in FedAvg, FEDCM in *full participation* converges without requiring bounded client dissimilarity. Our results extend theirs in that we prove that GHBM can achieve the same convergence rate even in (cyclic) partial participation, because our novel momentum formulation approximates the update of momentum term with all the clients' gradients.

**Comparison with SCAFFOLD-M.** By integrating classical momentum into SCAFFOLD, SCAFFOLD-M accelerated its convergence rate while maintaining robustness to unbounded heterogeneity in partial participation. Besides our rate being better, SCAFFOLD-M inherits the same limitations of SCAFFOLD, that is the dependence on the number of participating clients, which affects its practical applicability in scenarios with very low client-participation (Reddi et al., 2021).

## 5 Experimental Results

We present evidence both in controlled and real-world scenarios, showing that: (i) the GHBM formulation is pivotal to enable momentum to provide an effective correction even in extreme heterogeneity, (ii) our adaptive LOCALGHBM effectively exploits client participation to enhance communication efficiency and (iii) our proposed algorithms are suitable for cross-device scenarios, with stark improvement on large datasets and architectures (*e.g.* VIT-B\16).

### 5.1 Setup

**Scenarios, Datasets and Models.** For the controlled scenarios, we employ CIFAR-10/100 as computer vision tasks, with RESNET-20 and the same CNN similar to a LeNet-5 commonly used in FL works (Hsu et al., 2020), and SHAKESPEARE dataset as NLP task following (Reddi et al., 2021; Karimireddy et al., 2021). For CIFAR-10/100, the local datasets are obtained by sampling the examples according to a Dirichlet distribution with concentration parameter $\alpha$, as is common practice

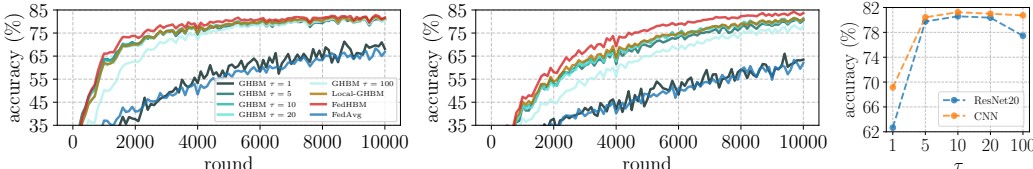

Figure 2: GHBM effectively counteracts the effects of heterogeneity: our momentum formulation ($\tau > 1$) is crucial for superior performance , with an optimal value $\tau = 1/C = 10$, as predicted in theory. Results on CIFAR-10 with CNN (left) and RESNET-20 (right), under worst-case heterogeneity.

Hsu et al. (2020) (additional details in Appendix C.2). We denote as NON-IID and IID respectively the splits corresponding to $\alpha = 0$ and $\alpha = 10.000$, while for SHAKESPEARE we use instead the predefined splits (Caldas et al., 2019). The datasets are partitioned among $K = 100$ clients, selecting a portion $C = 10\%$ of them at each round. As real-world scenarios, we adopt the large-scale GLDV2 and INATURALIST datasets as CV tasks, with both a VIT-B\16 (Dosovitskiy et al., 2021) and a MOBILENETV2 (Sandler et al., 2018) pretrained on ImageNet, and STACKOVERFLOW dataset as NLP task, following Reddi et al. (2021); Karimireddy et al. (2021). These settings are particularly challenging, because the learning tasks are complex, the number of client is high and the client participation (for convenience directly reported in Tab. 3) is scarce (see Appendix C.1 for details).

**Metrics and Experimental protocol.** As metrics, we consider *final model quality*, as the top-1 accuracy over the last 100 rounds of training (Tab. 2-3, Fig. 6), and *communication/computational efficiency*: this is evaluated by measuring the amount of exchanged bytes and the wall-clock time spent by an algorithm to reach the performance of FEDAVG (Tab. 4). Results are always reported as the average over 5 independent runs, performed on the best-performing hyperparameters extensively and carefully searched separately for all competitor algorithms. For additional details about the datasets, splits, model architectures, and algorithms' hyperparameters, see Appendix C.4.

### 5.2 COUNTERACTING CLIENT DRIFT WITH GHBM

Figure 2 validates our momentum design under worst-case heterogeneity: $\tau > 1$ is crucial to enable momentum to provide an effective correction to client drift. Indeed, previous momentum-based methods (Xu et al., 2021; Ozfatura et al., 2021), which are special cases of GHBM with $\tau = 1$, are observed to be ineffective in improving FEDAVG. The best value of $\tau$ is experimentally proven to be $\approx 1/C = 10$, and sub-optimal large values of $\tau$ only marginal affect performance (rightmost plot), confirming our theoretical analysis in Sec. 4.2. Our communication-efficient instances always match or surpass the best-tuned GHBM, confirming that their adaptive estimate of each client's momentum positively contributes in a scenario of stochastic client participation (see Sec. 4.2).

### 5.3 COMPARISON WITH THE STATE-OF-ART

**Results in controlled scenario** Our results in Tab. 2 clearly indicate that existing algorithms behave inconsistently when larger models are used (RESNET-20) and fail at improving FEDAVG. In particular, our experimentation reveals that estimating the momentum using full batch gradients as done by MIMEMOM (Karimireddy et al. (2021)) does not guarantee an effective correction in most difficult scenarios. Conversely, our algorithms outperform the FEDAVG with an impressive margin of $+20.6\%$ and $+14.4\%$ on RESNET-20 and CNN under worst-case heterogeneity, and consistently over less severe conditions (higher values of $\alpha$ in Fig. 3). We also report the performance of FEDCM in full participation, as upper bound for momentum in FL (see Sec. 3.3). Our methods closely match the upper bound in all cases except for CIFAR-100, in the non-iid setting. This is motivated by the algorithm being still far from the convergence point in the given round budget, and by the error introduced by using past gradients (see discussion in Sec. 3.3).

**Results in real-world large-scale scenarios** Extending the experimentation to settings characterized by extremely low client participation, we test both our GHBM with $\tau$ tuned via a grid-search and our adaptive FEDHBM, which exploits client participation to keep the same communication complexity of FEDAVG. As discussed in sections 3.3-4.2, under such extreme client participation patterns GHBM performs better because the trade-off between heterogeneity reduction and gradient lag is explicitly tuned by the choice of the best performing $\tau$, while FEDHBM will likely adopt a suboptimal value. However, results in Tab. 3 show a stark improvement over the state-of-art for both our algorithms, indicating that the design principle of our momentum formulation is remarkably robust and provides effective improvement even when client participation is very low (*e.g.* $C \leq 1\%$).

Table 2: Comparison with state-of-the-art in controlled setting (acc@10k-20k rounds for RESNET-20/CNN). NON-IID ($\alpha = 0$) and IID ($\alpha = 10k$). Best result in **bold**, second best underlined. ✗ indicates non-convergence.

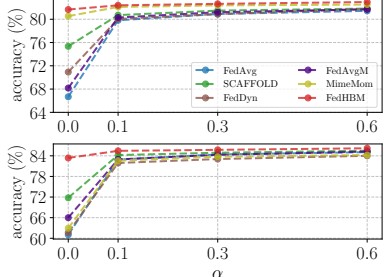

Figure 3: Final model quality at different values of $\alpha$ (lower $\alpha \to$ higher heterogeneity) on CIFAR-10, with CNN (top) and RESNET-20 (bottom).

| METHOD | CIFAR-100 (RESNET-20) | | CIFAR-100 (CNN) | | SHAKESPEARE | |
|---|---|---|---|---|---|---|
| | NON-IID | IID | NON-IID | IID | NON-IID | IID |
| FEDAVG | 21.9 ±0.9 | 58.6 ±0.4 | 35.6 ±0.2 | 49.7 ±0.2 | 47.3 ±0.1 | 47.1 ±0.2 |
| FEDPROX | 22.1 ±1.0 | 58.5 ±0.3 | 35.5 ±0.3 | 49.9 ±0.2 | 47.3 ±0.1 | 47.1 ±0.2 |
| SCAFFOLD | 30.7 ±1.3 | 58.0 ±0.6 | 45.5 ±0.1 | 49.4 ±0.4 | 50.2 ±0.1 | 50.1 ±0.1 |
| FEDDYN | 6.0 ±0.5 | 60.8 ±0.7 | ✗ | 51.9 ±0.2 | 50.7 ±0.2 | 50.8 ±0.2 |
| ADABEST | 8.4 ±2.0 | 55.6 ±0.3 | 35.6 ±0.3 | 49.7 ±0.2 | 47.3 ±0.1 | 47.1 ±0.2 |
| MIME | 9.0 ±0.4 | 59.0 ±0.3 | 36.3 ±0.5 | 50.9 ±0.4 | 48.3 ±0.2 | 48.5 ±0.1 |
| FEDAVGM | 22.8 ±0.8 | 58.7 ±0.9 | 35.2 ±0.9 | 50.7 ±0.2 | 50.0 ±0.0 | 50.4 ±0.1 |
| SCAFFOLD-M | 30.9 ±0.7 | 60.1 ±0.5 | 45.7 ±0.2 | 50.1 ±0.3 | 50.8 ±0.0 | 51.0 ±0.1 |
| FEDCM (GHBM $\tau$=1) | 22.2 ±1.0 | 53.1 ±0.2 | 36.0 ±0.3 | 50.2 ±0.5 | 49.2 ±0.1 | 50.4 ±0.1 |
| FEDADC (GHBM $\tau$=1) | 22.4 ±0.1 | 53.2 ±0.2 | 37.9 ±0.3 | 50.2 ±0.4 | 49.2 ±0.1 | 50.4 ±0.1 |
| MIMEMOM | 21.7 ±1.1 | 60.5 ±0.6 | 48.2 ±0.7 | 50.6 ±0.1 | 48.5 ±0.2 | 48.9 ±0.2 |
| MIMELITEMOM | 14.4 ±0.6 | 59.2 ±0.5 | 46.0 ±0.3 | 50.7 ±0.1 | 49.1 ±0.4 | 49.4 ±0.3 |
| FEDCM (full participation) | 51.4 ±1.2 | 62.2 ±0.3 | 50.5 ±0.3 | 51.9 ±0.1 | 51.3 ±0.1 | 51.5 ±0.1 |
| LOCALGHBM (ours) | 38.2 ±1.0 | 62.0 ±0.5 | 50.3 ±0.5 | 51.9 ±0.4 | 51.2 ±0.1 | 51.1 ±0.3 |
| FEDHBM (ours) | 42.5 ±0.8 | 62.5 ±0.5 | 50.4 ±0.5 | 52.0 ±0.4 | 51.3 ±0.1 | 51.4 ±0.2 |

Table 3: Test accuracy (%) comparison of best SOTA FL algorithms on large-scale and realistic settings. GHBM is the best algorithm when client participation is extremely low, while FEDHBM still improves the other competitors by a large margin. ✗ means that the algorithm did not converge.

| METHOD | MOBILENETV2 | | | | VIT-B\16 | | | |
|---|---|---|---|---|---|---|---|---|
| | GLDv2 | INATURALIST | | | GLDv2 | INATURALIST | | STACKOVERFLOW |
| | $C \approx 0.79\%$ | $C \approx 0.1\%$ | $C \approx 0.5\%$ | $C \approx 1\%$ | $C \approx 0.79\%$ | $C \approx 0.1\%$ | $C \approx 0.5\%$ | $C \approx 0.12\%$ |
| FEDAVG | 60.3 ±0.2 | 38.0 ±0.8 | 45.25 ±0.1 | 47.59 ±0.1 | 68.5 ±0.5 | 65.6 ±0.1 | 70.7 ±0.8 | 24.0 ±0.4 |
| SCAFFOLD | 61.0 ±0.1 | ✗ | ✗ | ✗ | 67.5 ±3.3 | ✗ | ✗ | 24.8 ±0.4 |
| FEDAVGM | 61.5 ±0.2 | 41.3 ±0.4 | 46.0 ±0.1 | 48.4 ±0.1 | 70.0 ±0.5 | 66.0 ±0.2 | 71.4 ±0.5 | 24.1 ±0.3 |
| MIMEMOM | ✗ | ✗ | ✗ | ✗ | ✗ | ✗ | ✗ | 24.9 ±0.6 |
| GHBM - best $\tau$ (ours) | 65.9 ±0.1 | 41.8 ±0.1 | 48.7 ±0.1 | 50.5 ±0.1 | 74.3 ±0.6 | 68.8 ±0.3 | 73.5 ±0.4 | 27.0 ±0.1 |
| FEDHBM (ours) | 65.4 ±0.2 | 41.6 ±0.2 | 47.3 ±0.0 | 49.8 ±0.0 | 73.1 ±0.9 | 66.7 ±0.7 | 72.1 ±0.5 | 24.5 ±0.4 |

**Communication efficiency** To demonstrate the communication-efficiency of our algorithms, in Tab. 4 we calculated the communication and computational cost of our simulations for reaching the performance of FEDAVG (details in Appendix C.3). These analyses reveal that our proposed algorithms lead to a dramatic reduction in both communication and computational cost, with an average saving of respectively $+67.5\%$ and $+62.5\%$. In practice, both our algorithms show faster convergence and higher final model quality: in particular, in settings with extremely low client participation (*e.g.* GLDv2 and INATURALIST), GHBM is more suitable for best accuracy, while FEDHBM is the best at lowering the communication cost.

Table 4: Communication and computational cost for reaching the final model quality of FEDAVG, across academic and real-world large-scale datasets (details in Appendix C.3). The coloured arrows indicate respectively a reduction (↓) and an increase (↑) of communication/computational cost.

| METHOD | COMM. OVERHEAD | COMMUNICATION COST (BYTES EXCHANGED) | | | | COMPUTATIONAL COST (WALL-CLOCK TIME HH:MM) | | | |
|---|---|---|---|---|---|---|---|---|---|
| | | CIFAR-100 ($\alpha = 0$) | | GLDv2 | | CIFAR-100 ($\alpha = 0$) | | GLDv2 | |
| | | CNN | RESNET-20 | MOBILENETV2 | VIT-B\16 | CNN | RESNET-20 | MOBILENETV2 | VIT-B\16 |
| FEDAVG | 1× | 30.9 GB | 10.3 GB | 89.8 GB | 483.7 GB | 02:05 | 03:36 | 13:51 | 13:56 |
| SCAFFOLD | 2× | 31.8 GB ↑3.0% | 12.1 GB ↑17.5% | 51.2 GB ↓43.0% | 967.4 GB ↑100.0% | 01:15 ↓40.0% | 02:27 ↓41.0% | 08:28 ↓38.9% | 15:15 ↑9.4% |
| FEDAVGM | 1× | 28.9 GB ↓6.5% | 9.2 GB ↓10.7% | 73.6 GB ↓18.0% | 403.1 GB ↓16.7% | 01:57 ↓6.5% | 03:14 ↓10.2% | 11:22 ↓18.0% | 11:37 ↓16.7% |
| MIMEMOM | 3× | 21.5 GB ↓30.4% | 30.9 GB ↑200.0% | 269.4 GB ↑200.0% | 1.417 TB ↑200.0% | 01:27 ↓30.4% | 10:42 ↑197.8% | 41:07 ↑197.8% | 41:30 ↑197.8% |
| GHBM (ours) | 1.5× | 6.4 GB ↓79.3% | 6.3 GB ↓38.8% | 48.5 GB ↓46.0% | 314.4 GB ↓35.0% | 00:19 ↓84.2% | 01:28 ↓59.3% | 05:20 ↓61.5% | 06:30 ↓53.3% |
| FEDHBM (ours) | 1× | 3.9 GB ↓87.4% | 3.7 GB ↓64.1% | 29.6 GB ↓67.0% | 234.4 GB ↓51.5% | 00:17 ↓86.0% | 01:18 ↓63.9% | 06:23 ↓54.0% | 07:31 ↓46.0% |

## 6 CONCLUSIONS

In this work, we propose a novel *Generalized Heavy-Ball Momentum* (GHBM), motivating its principled application in FL to counteract the effects of statistical heterogeneity. Based on GHBM, we present FEDHBM as an adaptive instance which is additionally communication-efficient by design. Our results in large-scale scenarios largely improve the state of art both in final model quality and communication efficiency. The generality and versatility of the novel GHBM formulation expands its potential applications to a wider range of scenarios where communication is a bottleneck, such as distributed learning.

## REPRODUCIBILITY STATEMENT

The authors are committed to ensuring the reproducibility of all results presented in this work. The main text provides a detailed algorithmic description of the proposed federated learning (FL) algorithms, along with comprehensive theoretical and experimental results. The Appendix expands on this by providing: (i) full formal proofs for the theoretical results (see Appendix B) and (ii) detailed descriptions of the datasets, model architectures, and hyperparameter tuning used in the experiments (see Appendix C). The code implementing the algorithms is included with the submission for the review process and will be made publicly available upon acceptance.

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

## A ADDITIONAL DISCUSSION

### A.1 EXTENDED RELATED WORKS

Recently, similarly based on variance reduction as SCAFFOLD, (Mishchenko et al., 2022) propose SCAFFNEW to achieve accelerated communication complexity in heterogeneous settings through control variates, guaranteeing convergence under arbitrary heterogeneity in full participation. The work by Mishchenko et al. (2024), under the assumption of second-order data heterogeneity, proposes an algorithm which can reduce client drift by estimating the global update direction as well as employing regularization. Similarly to the already discussed MIME (Karimireddy et al., 2021), Karagulyan et al. (2024) propose the SPAM algorithm and leverage momentum as a local correction term to benefit from second-order similarity.

Among momentum methods with similar guarantees than ours, SCAFFOLD-M (Cheng et al., 2024) integrates classical momentum into SCAFFOLD to attain a slightly better convergence rate and maintaining robustness to unbounded heterogeneity in partial participation. However, SCAFFOLD-M inherits the same limitations of SCAFFOLD: it requires clients to keep local client states across rounds, making the algorithm not well-suited for cross-device FL (Reddi et al., 2021); and it is limited by the ineffectiveness of variance reduction in deep learning (Defazio & Bottou, 2019). Conversely, momentum has proven fundamental to accelerate training in deep learning, and GHBM is the first algorithm that can use only momentum to converge under arbitrary heterogeneity in partial participation

### A.2 CROSS-SILO AND CROSS-DEVICE FL

**Setting Cross-silo FL.** In this setting, following the characterization in (Kairouz et al., 2021), the training nodes are expected to be different organizations or geo-distributed data centers. The number of such nodes is modest ($\mathcal{O}(10^2)$) and they are assumed to be almost always available and reliable. This makes it possible to maintain a state on nodes across two different rounds, and often the use of stateful clients is an indicator for an algorithm to be designed for this scenario. Usually, the problem of FL in such a setting is cast as a finite-sum optimization problem, where each function is the local clients' loss function (eq. 9)

**Setting cross-device FL.** Differently from cross-silo FL, in the cross-device setting the clients are assumed to be possibly unreliable edge devices, with only a fraction of them available at any given time. As such, communication is the primary bottleneck. Most importantly, they can be massive in number ($\mathcal{O}(10^{10})$), so this motivates the fact that they should be stateless since each client is likely to participate only once in the training procedure. Following the characterization in (Karimireddy et al., 2021), being the number of clients enormous, this problem can be modeled by introducing the stochasticity client-level, over the possibly sampled clients (eq. 10).

**CROSS-SILO:**

$$\arg \min_{\theta \in \mathbb{R}^d} \sum_{k \in \mathcal{S}} \frac{|\mathcal{D}_k|}{|\mathcal{D}_\mathcal{S}|} \mathbb{E}_{(x,y) \sim \mathcal{D}_k}[L(f_\theta; (x,y))] \quad (9)$$

**CROSS-DEVICE:**

$$\arg \min_{\theta \in \mathbb{R}^d} \mathbb{E}_{i \sim S} \left[ \sum_{j=1}^{|\mathcal{D}_i|} \frac{1}{|\mathcal{D}_i|} L(f_\theta; (x_j, y_j)) \right] \quad (10)$$

**Cross-silo and cross-device in practice.** The two aforementioned setups are however extreme cases, and real-world scenarios will likely enjoy some features from both settings. Previous FL works that address cross-silo FL usually experiment with a few hundred devices but account for low participation and unreliability, and treat communication as the primary bottleneck (Karimireddy et al., 2020; Acar et al., 2021). However, they are stateful, and this has raised concerns about their applicability in cross-device: in particular Karimireddy et al. (2021) noticed that the control variates in Karimireddy et al. (2020) get stale as clients are not seen again during training, and highlights that stateless clients reflect the different formulation in equations 10, 9. In this work we show that FEDHBM is robust to extremely low participation rates, and that it gets more effective as each client participates in the training process. Remarkably, our method succeeds in scenarios where state-of-art methods fail (see and tables 2-3).

### A.3 NOTES ON FAILURE CASES OF SOTA ALGORITHMS

In this paper, we evaluated our approach using the large-scale FL datasets proposed by (Hsu et al., 2020). Notably, several recent state-of-the-art FL algorithms failed to converge on these datasets. For

SCAFFOLD this result aligns with prior works (Reddi et al., 2021; Karimireddy et al., 2021), since it is unsuitable for cross-device FL with thousands of devices. Indeed, the client control variates can become stale, and may consequently degrade the performance. For MIMEMOM (Karimireddy et al., 2021), despite extensive hyperparameter tuning using the authors' original code, we were unable to achieve convergence. This finding is surprising since the approach has been proposed to tackle cross-device FL. To our knowledge, this is the first work to report these failure cases, likely due to the lack of prior evaluations on such challenging datasets. We believe these findings underscore the need for further investigation into the factors contributing to algorithm performance in large-scale, heterogeneous FL settings.

### A.4 BROADER IMPACT AND LIMITATIONS

The algorithms presented in this work offer a substantial advancement in federated training efficiency. By significantly improving performance while reducing computational, communication, and energy costs, our approach contributes to a more sustainable and scalable federated learning ecosystem. This marks a notable step towards wider adoption of FL in real-world applications, particularly in the challenging cross-device setting, where our methods have demonstrated remarkable flexibility and effectiveness. Despite these significant improvements over the state-of-the-art, challenges remain in fully realizing the potential of cross-device FL. Our results underscore the critical importance of accurately estimating the global direction for rapid algorithm convergence. Both GHBM and FEDHBM leverage this insight, correcting client drift through global direction estimation. However, the accurate estimation of this direction in extremely large-scale scenarios (*e.g.*, millions of clients with low participation rates) remains an open research problem.

## B PROOFS

### ALGORITHMS

To handle the proof, we analyze a simpler version of our algorithm, in which we use the update rule in eq. (4) instead of the one described in eq. (5). The resulting Algorithm 3 we analyze is reported along the plain GHBM (Algorithm 2) we used in the experiments. Both algorithms enjoy the same underlying idea: use the gradients of a larger portion of the clients to estimate the momentum term.

---

**Algorithm 2:** GHBM (PRACTICAL VERSION)

---

**Require:** initial model $\theta^0$, $K$ clients, $C$ participation ratio, $T$ number of total round, $B$ batch size, $\eta$ and $\eta_l$ learning rates.

1: **for** $t = 1$ to $T$ **do**
2:    $\mathcal{S}^t \leftarrow$ subset of clients $\sim \mathcal{U}(\mathcal{S}, \max(1, K \cdot C))$
3:    **for** $i \in \mathcal{S}^t$ **in parallel do**
4:       $\theta_i^{t,0} \leftarrow \theta^{t-1}$
5:       **for** $j = 1$ to $J$ **do**
6:          sample a mini-batch $d_{i,j}$ from $\mathcal{D}_i$
7:          $u_i^{t,j} \leftarrow \nabla f_i(\theta_i^{t,j-1}, d_{i,j}) + \beta \tilde{m}_\tau^t$
8:          $\theta_i^{t,j} \leftarrow \theta_i^{t,j-1} - \eta_l u_i^{t,j}$
9:       **end for**
10:    **end for**
11:    $u^t \leftarrow \frac{1}{|\mathcal{S}^t|} \sum_{i=1}^{|\mathcal{S}^t|} \left( \theta^{t-1} - \theta_i^{t,J} \right)$
12:    $\theta^t \leftarrow \theta^{t-1} - \eta u^t$
13:    $\tilde{m}_\tau^{t+1} \leftarrow \frac{1}{\tau J} \left( \theta^{t-\tau} - \theta^t \right)$
14: **end for**

---

---

**Algorithm 3:** GHBM (THEORY VERSION)

---

**Require:** initial model $\theta^0$, $K$ clients, $C$ participation ratio, $T$ number of total round, $B$ batch size, $\eta$ and $\eta_l$ learning rates.

1: **for** $t = 1$ to $T$ **do**
2:     $\mathcal{S}^t \leftarrow$ subset of clients $\sim \mathcal{U}(\mathcal{S}, \max(1, K \cdot C))$
3:     **for** $i \in \mathcal{S}^t$ **in parallel do**
4:        $\theta_i^{t,0} \leftarrow \theta^{t-1}$
5:        **for** $j = 1$ to $J$ **do**
6:           sample a mini-batch $d_{i,j}$ from $\mathcal{D}_i$
7:           $u_i^{t,j} \leftarrow \beta \nabla f_i(\theta_i^{t,j-1}, d_{i,j}) + (1 - \beta)\tilde{m}_\tau^t$
8:           $\theta_i^{t,j} \leftarrow \theta_i^{t,j-1} - \eta_l u_i^{t,j}$
9:        **end for**
10:    **end for**
11:    $u^t \leftarrow \frac{1}{\eta_l |\mathcal{S}^t| J} \sum_{i=1}^{|\mathcal{S}^t|} \left( \theta^{t-1} - \theta_i^{t,J} \right)$
12:    $\bar{\theta}^t \leftarrow \theta^{t-1} - u^t + (1 - \beta)\tilde{m}_\tau^t$
13:    $\tilde{m}_\tau^{t+1} \leftarrow (1 - \beta)\tilde{m}_\tau^t + \frac{1}{\tau}\left( \bar{\theta}^{t-\tau} - \bar{\theta}^t \right)$
14:    $\theta^t \leftarrow \theta^{t-1} - \eta \tilde{m}_\tau^{t+1}$
15: **end for**

---

In the following, we list the differences between the two:

1. Explicit use of $\tau$-averaged gradients when updating the momentum term (line 13). This can be implemented by keeping server-side an auxiliary sequence of models $\bar{\theta}^t$, in which the momentum added client side is subtracted server-side (line 12), such that taking the difference of two models gives the sum of pseudo-grads.

2. Use of convex sum in local updates (line 7). This is done to align with the formulation of momentum methods in Cheng et al. (2024), and more in general with the formulation of momentum commonly analyzed in literature. There is no theoretical difference between the two versions, as they only differ by a constant scaling (Liu et al., 2020).

3. Use of gradients averaged over local steps (line 11). This is done to align with the analysis of Cheng et al. (2024); Xu et al. (2021), and it is equivalent to coupling server and client learning rates (*i.e.* setting $\eta = \gamma J \eta_l$ in Algorithm 3, where $\gamma$ is the server learning rate we would use in Algorithm 2).

The two algorithms have similar performances, which are reported in Fig. 4

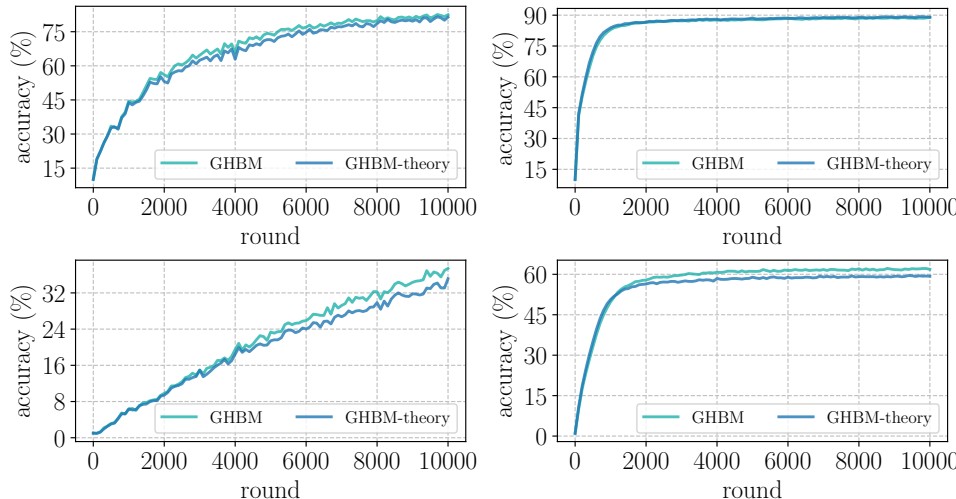

Figure 4: Comparing the GHBM implementation analyzed in theory (Algorithm 3) with the one proposed in the main paper (Algorithm 2). The plots show the convergence rate on CIFAR-10 (top) and CIFAR-100 (bottom), in non-iid (left) and iid (right) scenarios with RESNET-20 architecture.

PRELIMINARIES

Our convergence proof for GHBM is based on the recent work of Cheng et al. (2024), which offers new proof techniques for momentum-based FL algorithms. Throughout the proofs we use the following auxiliary variables to facilitate the presentation:

$$\mathcal{U}_t := \frac{1}{|\mathcal{S}|J} \sum_{j=1}^{J} \sum_{i=1}^{|\mathcal{S}|} \mathbb{E}\left[ \left\| \theta_i^{t,j} - \theta^{t-1} \right\|^2 \right] \tag{11}$$

$$\mathcal{E}_t := \mathbb{E}\left[ \left\| \nabla f(\theta^{t-1}) - \tilde{m}_\tau^{t+1} \right\|^2 \right] \tag{12}$$

$$\zeta_i^{t,j} := \mathbb{E}\left[ \theta_i^{t,j+1} - \theta_i^{t,j} \right] \tag{13}$$

$$\Xi_t := \frac{1}{|\mathcal{S}|} \sum_{i=1}^{|\mathcal{S}|} \mathbb{E}\left[ \left\| \zeta_i^{t,0} \right\|^2 \right]$$

$$\Lambda_t := \mathbb{E}\left[ \left\| \left( \frac{1}{\tau} \sum_{k=t-\tau+1}^{t} \frac{1}{|\mathcal{S}^k|J} \sum_{i=1}^{|\mathcal{S}^k|} \sum_{j=1}^{J} \tilde{g}_i^{k,j}(\theta_i^{k,j-1}) \right) - g^{t_\tau} \right\|^2 \right] \tag{14}$$

$$\gamma_t := \mathbb{E}\left[ \left\| g^{t_\tau} - \nabla f(\theta^{t-1}) \right\|^2 \right] \tag{15}$$

Additionally, here we report the *bounded gradient heterogeneity* assumption. It is used to quantify the heterogeneity reduction effect of GHBM varying its $\tau$ hyperparameter. Notice that our main claim does not depend on this assumption, as for the optimal value of $\tau = 1/C$ the assumption is not needed (see lemma 4.4).

**Assumption B.1** (Bounded gradient dissimilarity). There exist a constant $G \geq 0$ such that, $\forall i, \theta$:

$$\frac{1}{|\mathcal{S}|} \sum_{i=1}^{|\mathcal{S}|} \| g_i(\theta) - g(\theta) \|^2 \leq G^2$$

### B.1 MOMENTUM EXPRESSIONS

In this section we report the derivation of the momentum expressions in eq. (2) and eq. (6) from the main paper.

**Lemma B.2** (Heavy-Ball formulation of classical momentum). *Let us consider the following classical formulation of momentum:*

$$\tilde{m}^t = \beta \tilde{m}^{t-1} + \tilde{g}^t(\theta^{t-1}) \tag{16}$$

$$\theta^t = \theta^{t-1} - \eta \tilde{m}^t \tag{17}$$

*The same update rule can be equivalently expressed with the following, known as heavy-ball formulation:*

$$\theta^t = \theta^{t-1} + \beta(\theta^{t-1} - \theta^{t-2}) - \eta \tilde{g}(\theta^{t-1}) \tag{18}$$

*Proof.* First derive the expression of $\tilde{m}^t$ from eq. (17), both for time $t$ and $t-1$:

$$\tilde{m}^t = \frac{(\theta^{t-1} - \theta^t)}{\eta}$$

$$\tilde{m}^{t-1} = \frac{(\theta^{t-2} - \theta^{t-1})}{\eta}$$

Now plug these expressions into equation (16) to obtain (18):

$$\frac{(\theta^{t-1} - \theta^t)}{\eta} = \beta \frac{(\theta^{t-2} - \theta^{t-1})}{\eta} + \tilde{g}^t(\theta^{t-1})$$

$$(\theta^t - \theta^{t-1}) = \beta \left( \theta^{t-1} - \theta^{t-2} \right) - \eta \tilde{g}^t(\theta^{t-1})$$

$$\theta^t = \theta^{t-1} + \beta \left( \theta^{t-1} - \theta^{t-2} \right) - \eta \tilde{g}^t(\theta^{t-1})$$

$\square$

**Lemma B.3** (Heavy-Ball formulation of generalized momentum). *Let us consider the following generalized formulation of momentum:*

$$\tilde{m}_\tau^t = \frac{1}{\tau} \sum_{k=1}^{\tau} \beta \tilde{m}_\tau^{t-k} + \tilde{g}^t(\theta^{t-1}) \tag{19}$$

$$\theta^t = \theta^{t-1} - \eta \tilde{m}_\tau^t \tag{20}$$

*The same update rule can be equivalently expressed in an heavy ball form, which we call as Generalized Heavy-Ball momentum (GHB):*

$$\theta^t = \theta^{t-1} + \frac{\beta}{\tau}(\theta^{t-1} - \theta^{t-\tau-1}) - \eta \tilde{g}(\theta^{t-1}) \tag{21}$$

*Proof.* First derive the expression of $\tilde{m}_\tau^t$ from eq. (20), both for time $t$ and $t-1$:

$$\tilde{m}_\tau^t = \frac{\left(\theta^{t-1} - \theta^t\right)}{\eta}$$

$$\tilde{m}_\tau^{t-1} = \frac{\left(\theta^{t-2} - \theta^{t-1}\right)}{\eta}$$

Now plug these expressions into equation (19):

$$\frac{\left(\theta^{t-1} - \theta^t\right)}{\eta} = \frac{\beta}{\tau} \sum_{k=1}^{\tau} \frac{\left(\theta^{t-k-1} - \theta^{t-k}\right)}{\eta} + \tilde{g}^t(\theta^{t-1})$$

$$\left(\theta^t - \theta^{t-1}\right) = \frac{\beta}{\tau} \sum_{k=1}^{\tau} \left(\theta^{t-k} - \theta^{t-k-1}\right) - \eta \tilde{g}^t(\theta^{t-1})$$

$$\theta^t = \theta^{t-1} + \frac{\beta}{\tau} \sum_{k=1}^{\tau} \left(\theta^{t-k} - \theta^{t-k-1}\right) - \eta \tilde{g}^t(\theta^{t-1})$$

$$\theta^t = \theta^{t-1} + \frac{\beta}{\tau}(\theta^{t-1} - \theta^{t-\tau-1}) - \eta \tilde{g}^t(\theta^{t-1})$$

Where the last equality (21) comes from telescoping the summation on the rhs. $\square$

### B.2 TECHNICAL LEMMAS

Now we cover some technical lemmas which are useful for computations later on. These are known results that are reported here for the convenience of the reader.

**Lemma B.4** (relaxed triangle inequality). *Let $\{\boldsymbol{v}_1, \ldots, \boldsymbol{v}_n\}$ be $n$ vectors in $\mathbb{R}^d$. Then, the following is true:*

$$\left\| \sum_{i=1}^{n} \boldsymbol{v}_i \right\|^2 \leq n \sum_{i=1}^{n} \|\boldsymbol{v}_i\|^2$$

*Proof.* By Jensen's inequality, given a convex function $\phi$, a series of $n$ vectors $\{\boldsymbol{v}_1, \ldots, \boldsymbol{v}_n\}$ and a series of non-negative coefficients $\lambda_i$ with $\sum_{i=1}^{n} \lambda_i = 1$, it results that

$$\phi\left(\sum_{i=1}^{n} \lambda_i \boldsymbol{v}_i\right) \leq \sum_{i=1}^{n} \lambda_i \phi(\boldsymbol{v}_i)$$

Since the function $\boldsymbol{v} \to \|\boldsymbol{v}\|^2$ is convex, we can use this inequality with coefficients $\lambda_1 = \ldots = \lambda_n = 1/n$, with $\sum_{i=1}^{n} \lambda_i = 1$, and obtain that

$$\left\| \frac{1}{n} \sum_{i=1}^{n} \boldsymbol{v}_i \right\|^2 = \frac{1}{n^2} \left\| \sum_{i=1}^{n} \boldsymbol{v}_i \right\|^2 \leq \frac{1}{n} \sum_{i=1}^{n} \|\boldsymbol{v}_i\|^2$$

$\square$

### B.3 PROOFS OF MAIN LEMMAS

In this section we provide the proofs of the main theoretical results presented in the main paper.

**Proof of Lemma 4.4**    (Deviation of $\tau$-averaged gradient from true gradient)

Let define $\mathcal{S}_d := \mathcal{S} - \mathcal{S}_\tau^t$ and $\mathcal{S}_i := \mathcal{S} \cap \mathcal{S}_\tau^t$. Let us note that when all clients participate, *i.e.* $\mathcal{S}_d = \emptyset$, the claim is trivially true. For $\mathcal{S}_d \neq \emptyset$, we can expand the terms at the left-hand side using their definitions as follows:

$$\gamma_t = \mathbb{E}\left[\left\|\frac{1}{|\mathcal{S}_\tau^t|}\sum_{i=1}^{|\mathcal{S}_\tau^t|} g_i^t - \frac{1}{|\mathcal{S}|}\sum_{i=1}^{|\mathcal{S}|} g_i^t\right\|^2\right] \tag{22}$$

$$= \mathbb{E}\left[\left\|\sum_{i\in\mathcal{S}_i}\left(\frac{1}{|\mathcal{S}_\tau^t|} - \frac{1}{|\mathcal{S}|}\right)g_i^t - \sum_{k\in\mathcal{S}_d}\frac{1}{|\mathcal{S}|}g_k^t\right\|^2\right] \tag{23}$$

$$\overset{\text{lemma B.4}}{\leq} 2\left(\underbrace{\mathbb{E}\left[\left\|\sum_{i\in\mathcal{S}_i}\left(\frac{1}{|\mathcal{S}_\tau^t|} - \frac{1}{|\mathcal{S}|}\right)g_i^t\right\|^2\right]}_{\mathcal{T}_3} + \underbrace{\mathbb{E}\left[\left\|\sum_{k\in\mathcal{S}_d}\frac{1}{|\mathcal{S}|}g_k^t\right\|^2\right]}_{\mathcal{T}_4}\right) \tag{24}$$

Let us consider first $\mathcal{T}_3$. We have:

$$\mathcal{T}_3 = \mathbb{E}\left[\left\|\sum_{i\in\mathcal{S}_i}\left(\frac{1}{|\mathcal{S}_\tau^t|} - \frac{1}{|\mathcal{S}|}\right)g_i^t\right\|^2\right] = \mathbb{E}\left[\left(\frac{1}{|\mathcal{S}_\tau^t|} - \frac{1}{|\mathcal{S}|}\right)^2\left\|\sum_{i\in\mathcal{S}_i} g_i^t\right\|^2\right] \tag{25}$$

$$\overset{\text{lemma B.4}}{\leq} \mathbb{E}\left[\left(\frac{1}{|\mathcal{S}_\tau^t|} - \frac{1}{|\mathcal{S}|}\right)^2 |\mathcal{S}_i|\sum_{i\in\mathcal{S}_i}\left\|g_i^t\right\|^2\right] \tag{26}$$

$$= \mathbb{E}\left[\left(\frac{1}{|\mathcal{S}_\tau^t|} - \frac{1}{|\mathcal{S}|}\right)^2 |\mathcal{S}_i|\sum_{i\in\mathcal{S}_i}\left\|g_i^t - \nabla f(\theta^{t-1}) + \nabla f(\theta^{t-1})\right\|^2\right] \tag{27}$$

$$\overset{\text{lemma B.4}}{\leq} 2\mathbb{E}\left[\left(\frac{1}{|\mathcal{S}_\tau^t|} - \frac{1}{|\mathcal{S}|}\right)^2 |\mathcal{S}_i|\sum_{i\in\mathcal{S}_i}\left(\left\|g_i^t - \nabla f(\theta^{t-1})\right\|^2 + \left\|\nabla f(\theta^{t-1})\right\|^2\right)\right] \tag{28}$$

$$\overset{\text{assumption B.1}}{\leq} 2\mathbb{E}\left[\left(\frac{1}{|\mathcal{S}_\tau^t|} - \frac{1}{|\mathcal{S}|}\right)^2 |\mathcal{S}_i|\left(|\mathcal{S}_i|G^2 + \sum_{i\in\mathcal{S}_i}\left\|\nabla f(\theta^{t-1})\right\|^2\right)\right] \tag{29}$$

Since the term $\nabla f(\theta^{t-1})$ does not depend on the index $i$, we get

$$2\mathbb{E}\left[\left(\frac{1}{|\mathcal{S}_\tau^t|} - \frac{1}{|\mathcal{S}|}\right)^2 |\mathcal{S}_i|\left(|\mathcal{S}_i|G^2 + \sum_{i\in\mathcal{S}_i}\left\|\nabla f(\theta^{t-1})\right\|^2\right)\right] \tag{30}$$

$$= 2\mathbb{E}\left[\left(\frac{1}{|\mathcal{S}_\tau^t|} - \frac{1}{|\mathcal{S}|}\right)^2 |\mathcal{S}_i|\left(|\mathcal{S}_i|G^2 + |\mathcal{S}_i|\left\|\nabla f(\theta^{t-1})\right\|^2\right)\right] \tag{31}$$

$$= 2\mathbb{E}\left[\left(\frac{1}{|\mathcal{S}_\tau^t|} - \frac{1}{|\mathcal{S}|}\right)^2 |\mathcal{S}_i|^2\right]\left(G^2 + \left\|\nabla f(\theta^{t-1})\right\|^2\right) \tag{32}$$

Now, note that $\mathcal{S}_\tau^t \subseteq \mathcal{S} \implies |\mathcal{S}_i| = |\mathcal{S}_\tau^t|$. Therefore,

$$\mathcal{T}_3 \le 2\mathbb{E}\left[\left(\frac{1}{|\mathcal{S}_\tau^t|} - \frac{1}{|\mathcal{S}|}\right)^2 |\mathcal{S}_i|^2\right]\left(G^2 + \|\nabla f(\theta^{t-1})\|^2\right) \tag{33}$$

$$= 2\mathbb{E}\left[\left(\frac{|\mathcal{S}| - |\mathcal{S}_\tau^t|}{|\mathcal{S}|}\right)^2\right]\left(G^2 + \|\nabla f(\theta^{t-1})\|^2\right) \tag{34}$$

Moving now to $\mathcal{T}_4$, we have:

$$\mathcal{T}_4 = \mathbb{E}\left[\left\|\sum_{k\in\mathcal{S}_d}\frac{1}{|\mathcal{S}|}g_k^t\right\|^2\right] \le \mathbb{E}\left[\left(\frac{1}{|\mathcal{S}|}\right)^2\left\|\sum_{k\in\mathcal{S}_d}g_k^t\right\|^2\right] \tag{35}$$

$$\overset{\text{lemma B.4}}{\le} \mathbb{E}\left[\left(\frac{1}{|\mathcal{S}|}\right)^2 |\mathcal{S}_d|\sum_{k\in\mathcal{S}_d}\|g_k^t\|^2\right] \tag{36}$$

$$= \mathbb{E}\left[\left(\frac{1}{|\mathcal{S}|}\right)^2 |\mathcal{S}_d|\sum_{k\in\mathcal{S}_d}\|g_k^t - \nabla f(\theta^{t-1}) + \nabla f(\theta^{t-1})\|^2\right] \tag{37}$$

$$\overset{\text{lemma B.4}}{\le} 2\mathbb{E}\left[\left(\frac{1}{|\mathcal{S}|}\right)^2 |\mathcal{S}_d|\sum_{k\in\mathcal{S}_d}\left(\|g_k^t - \nabla f(\theta^{t-1})\|^2 + \|\nabla f(\theta^{t-1})\|^2\right)\right] \tag{38}$$

$$\overset{\text{assumption B.1}}{\le} 2\mathbb{E}\left[\left(\frac{1}{|\mathcal{S}|}\right)^2 |\mathcal{S}_d|\left(|\mathcal{S}_d|G^2 + \sum_{k\in\mathcal{S}_d}\|\nabla f(\theta^{t-1})\|^2\right)\right] \tag{39}$$

$$= 2\mathbb{E}\left[\left(\frac{1}{|\mathcal{S}|}\right)^2 |\mathcal{S}_d|\left(|\mathcal{S}_d|G^2 + |\mathcal{S}_d|\|\nabla f(\theta^{t-1})\|^2\right)\right] \tag{40}$$

$$= 2\mathbb{E}\left[\left(\frac{|\mathcal{S}_d|}{|\mathcal{S}|}\right)^2\right]\left(G^2 + \|\nabla f(\theta^{t-1})\|^2\right) \tag{41}$$

$$\tag{42}$$

Observing that $|\mathcal{S}_d| = |\mathcal{S}| - |\mathcal{S}_\tau^t|$ we obtain:

$$\mathcal{T}_4 \le 2\mathbb{E}\left[\left(\frac{|\mathcal{S}_d|}{|\mathcal{S}|}\right)^2\right]\left(G^2 + \|\nabla f(\theta^{t-1})\|^2\right) = \mathbb{E}\left[\left(\frac{|\mathcal{S}| - |\mathcal{S}_\tau^t|}{|\mathcal{S}|}\right)^2\right]\left(G^2 + \|\nabla f(\theta^{t-1})\|^2\right) \tag{43}$$

Finally, by plugging (33) and (43) in (24) we obtain

$$\mathbb{E}_{\mathcal{S}^t\sim\mathcal{U}(\mathcal{S})}\left[\left\|g^{(t)\tau}(\theta) - \nabla f(\theta)\right\|^2\right] \le 8\mathbb{E}_{\mathcal{S}^t\sim\mathcal{U}(\mathcal{S})}\left[\left(\frac{|\mathcal{S}| - |\mathcal{S}_\tau^t|}{|\mathcal{S}|}\right)^2\right]\left(G^2 + \|\nabla f(\theta)\|^2\right)$$

which concludes the proof.

$\square$

**Proof of Corollary 4.5**  This corollary follows from Lemma 4.4, which states that

$$\mathbb{E}_{\mathcal{S}^t\sim\mathcal{U}(\mathcal{S})}\left[\left\|g^{(t)\tau}(\theta) - \nabla f(\theta)\right\|^2\right] \le 8\mathbb{E}_{\mathcal{S}^t\sim\mathcal{U}(\mathcal{S})}\left[\left(\frac{|\mathcal{S}| - |\mathcal{S}_\tau^t|}{|\mathcal{S}|}\right)^2\right]\left(G^2 + \|\nabla f(\theta)\|^2\right)$$

To prove the results, we use (i) assumption 4.3, (ii) the fact that $|\mathcal{S}^t| = |\mathcal{S}|C \ \forall t$ and (iii) $\mathcal{S}_\tau^t$ is union of $\tau$ disjoint $\mathcal{S}^t$ sets. Using points (i)-(iii), and assuming $\tau \in [0, \frac{1}{C}]$, it follows that:

$$\left\|g^{(t)\tau}(\theta) - \nabla f(\theta)\right\|^2 \le 8\left(1 - \tau C\right)^2\left(G^2 + \|\nabla f(\theta)\|^2\right)$$

$\square$

**Proof of Lemma 4.6** (Bounded error of delayed gradients)

Note that, by assumption 4.3, $|\mathcal{S}^t| = |\mathcal{S}|C \,\forall t$, and that $|\mathcal{S}|C\tau = |\mathcal{S}^t_\tau|$:

$$
\Lambda_t = \mathbb{E}\left[\left\|\frac{1}{\tau}\sum_{k=t-\tau+1}^{t}\frac{1}{|\mathcal{S}^k|J}\sum_{i=1}^{|\mathcal{S}^k|}\sum_{j=1}^{J}\tilde{g}_i^{k,j}(\theta_i^{k,j-1}) - g^{t_\tau}\right\|^2\right] \tag{44}
$$

$$
= \mathbb{E}\left[\left\|\frac{1}{\tau}\sum_{k=t-\tau+1}^{t}\frac{1}{|\mathcal{S}^k|J}\sum_{i=1}^{|\mathcal{S}^k|}\sum_{j=1}^{J}\left(\tilde{g}_i^{k,j}(\theta_i^{k,j-1}) - g_i(\theta^{t-1})\right)\right\|^2\right] \tag{45}
$$

$$
= \mathbb{E}\left[\left\|\frac{1}{\tau}\sum_{k=t-\tau+1}^{t}\frac{1}{|\mathcal{S}^k|J}\sum_{i=1}^{|\mathcal{S}^k|}\sum_{j=1}^{J}\left(\tilde{g}_i^{k,j}(\theta_i^{k,j-1}) - g_i(\theta_i^{k,j-1}) + g_i(\theta_i^{k,j-1}) - g_i(\theta^{k-1}) + g_i(\theta^{k-1}) - g_i(\theta^{t-1})\right)\right\|^2\right] \tag{46}
$$

$$
\leq 3\left(\mathcal{T}_1 + \mathcal{T}_2 + \mathcal{T}_3\right) \tag{47}
$$

$$
\mathcal{T}_1 = \mathbb{E}\left[\left\|\frac{1}{\tau}\sum_{k=t-\tau+1}^{t}\frac{1}{|\mathcal{S}^k|J}\sum_{i=1}^{|\mathcal{S}^k|}\sum_{j=1}^{J}\left(\tilde{g}_i^{k,j}(\theta_i^{k,j-1}) - g_i(\theta_i^{k,j-1})\right)\right\|^2\right] \tag{48}
$$

$$
\leq \frac{1}{\tau}\frac{\sigma^2}{|\mathcal{S}^t|J} = \frac{\sigma^2}{|\mathcal{S}^t_\tau|J} \tag{49}
$$

$$
\mathcal{T}_2 = \mathbb{E}\left[\left\|\frac{1}{\tau}\sum_{k=t-\tau+1}^{t}\frac{1}{|\mathcal{S}^k|J}\sum_{i=1}^{|\mathcal{S}^k|}\sum_{j=1}^{J}\left(g_i(\theta_i^{k,j-1}) - g_i(\theta^{k-1})\right)\right\|^2\right] \tag{50}
$$

$$
\leq \frac{L^2}{|\mathcal{S}|J\tau}\sum_{k=t-\tau+1}^{t}\sum_{i=1}^{|\mathcal{S}|}\sum_{j=1}^{J}\mathbb{E}\left[\left\|\theta^{k,j-1} - \theta^{k-1}\right\|^2\right] \tag{51}
$$

$$
= \frac{L^2}{\tau}\sum_{k=t-\tau+1}^{t}\mathcal{U}_k \tag{52}
$$

$$
\mathcal{T}_3 = \mathbb{E}\left[\left\|\frac{1}{\tau}\sum_{k=t-\tau+1}^{t}\frac{1}{|\mathcal{S}^k|J}\sum_{i=1}^{|\mathcal{S}^k|}\sum_{j=1}^{J}\left(g_i(\theta^{k-1}) - g_i(\theta^{t-1})\right)\right\|^2\right] \tag{53}
$$

$$
\leq \frac{L^2}{|\mathcal{S}|\tau}\sum_{k=t-\tau+1}^{t}\sum_{i=1}^{|\mathcal{S}|}\mathbb{E}\left[\left\|\theta^{k-1} - \theta^{t-1}\right\|^2\right] \tag{54}
$$

$$
\leq \frac{L^2}{\tau}\sum_{k=t-\tau+1}^{t}\mathbb{E}\left[\left\|\theta^{k-1} - \theta^{t-1}\right\|^2\right] \tag{55}
$$

$$
= \frac{L^2}{\tau}\sum_{k=t-\tau+1}^{t}(t-k)\,\mathbb{E}\left[\left\|\theta^{k} - \theta^{k-1}\right\|^2\right] \tag{56}
$$

$$
\leq 2L^2\eta^2\sum_{k=t-\tau+1}^{t-1}\left(\mathbb{E}\left[\left\|\nabla f(\theta^{k-1})\right\|^2\right] + \mathcal{E}_k\right) \tag{57}
$$

So, combining with lemma B.6 and lemma B.7 we have:

$$\sum_{t=1}^{T} \Lambda_t \le 3 \left( \frac{T\sigma^2}{|\mathcal{S}_\tau^t|J} + L^2 \sum_{t=1}^{T} \mathcal{U}_t + 2L^2\eta^2(\tau-1) \sum_{t=1}^{T-1} \left( \mathbb{E}\left[ \left\| \nabla f(\theta^{t-1}) \right\|^2 \right] + \mathcal{E}_t \right) \right) \tag{58}$$

$$\overset{\text{lemma B.6}}{=} 3 \left( \frac{T\sigma^2}{|\mathcal{S}_\tau^t|J} + 2L^2\eta^2(\tau-1) \sum_{t=1}^{T-1} \left( \mathbb{E}\left[ \left\| \nabla f(\theta^{t-1}) \right\|^2 \right] + \mathcal{E}_t \right) \right. \tag{59}$$

$$\left. + \underbrace{L^2 T J \eta_l^2 \beta^2 \sigma^2 \left( 1 + 2J^3 \eta_l^2 \beta^2 L^2 \right)}_{\mathcal{T}_4} + 2J^2 L^2 e^2 \sum_{t=1}^{T} \Xi_t \right) \right)$$

$$\overset{\text{lemma B.7}}{=} 3 \left( \frac{T\sigma^2}{|\mathcal{S}_\tau^t|J} + 2L^2\eta^2(\tau-1) \sum_{t=1}^{T-1} \left( \mathbb{E}\left[ \left\| \nabla f(\theta^{t-1}) \right\|^2 \right] + \mathcal{E}_t \right) \right. \tag{60}$$

$$\left. + \mathcal{T}_4 + \underbrace{2J^2 L^2 e^2 \left( 4\eta_l^2 \left( (1-\beta)^2 + e(\beta\eta LT)^2 \right) \right)}_{\alpha_1} \sum_{t=0}^{T-1} \left( \mathcal{E}_t + \mathbb{E}\left[ \left\| \nabla f(\theta^{t-1}) \right\|^2 \right] \right) \right.$$

$$\left. + \underbrace{2e^2 J^2 L^2 (2e\eta_l^2 \beta\tau T G_\tau)}_{\mathcal{T}_5} \right)$$

$$= 3 \left( \frac{T\sigma^2}{|\mathcal{S}_\tau^t|J} + \mathcal{T}_4 + \underbrace{(\alpha_1 + 2L^2\eta_l^2(\tau-1))}_{\alpha_2} \sum_{t=1}^{T-1} \left( \mathbb{E}\left[ \left\| \nabla f(\theta^{t-1}) \right\|^2 \right] + \mathcal{E}_t \right) + \mathcal{T}_5 \right) \tag{61}$$

$$\square$$

## B.4 Convergence proof

**Lemma B.5** (Bounded variance of server updates). *Under assumptions 4.1-4.2, it holds that:*

$$\sum_{t=1}^{T} \mathcal{E}_t \le \frac{8}{5\beta} \mathcal{E}_0 + \frac{3}{5} \sum_{t=0}^{T-1} \mathbb{E}\left[ \left\| \nabla f(\theta^{t-1}) \right\|^2 \right] + 21\beta \frac{\sigma^2}{|\mathcal{S}_\tau^t|J} T + \tag{62}$$

$$+ \frac{448}{5} (\eta_l J L)^2 (e^3 \tau T) G_\tau + 6\beta \sum_{t=1}^{T} \gamma_t$$

*Proof.*

$$\mathcal{E}_t := \mathbb{E}\left[ \left\| \nabla f(\theta^{t-1}) - \tilde{m}_\tau^{t+1} \right\|^2 \right] \tag{63}$$

$$= \mathbb{E}\left[ \left\| (1-\beta)(\nabla f(\theta^{t-1}) - \tilde{m}_\tau^t) + \beta(\nabla f(\theta^{t-1}) - \tilde{g}^{t_\tau}) \right\|^2 \right] \tag{64}$$

$$= \mathbb{E}\left[ \left\| (1-\beta)(\nabla f(\theta^{t-1}) - \tilde{m}_\tau^t) \right\|^2 \right] + \beta^2 \mathbb{E}\left[ \left\| (\nabla f(\theta^{t-1}) - \tilde{g}^{t_\tau}) \right\|^2 \right] \tag{65}$$

$$+ 2\beta \mathbb{E}\left[ \left\langle (1-\beta)(\nabla f(\theta^{t-1}) - \tilde{m}_\tau^t), \nabla f(\theta^{t-1}) - \frac{1}{\tau} \sum_{k=t-\tau+1}^{t} \frac{1}{|\mathcal{S}^k|J} \sum_{i=1}^{|\mathcal{S}^k|} \sum_{j=1}^{J} g_i(\theta_i^{k,j-1}) \right\rangle \right] \tag{66}$$

Using the AM-GM inequality and lemma B.4:

$$\leq \left(1 + \frac{\beta}{2}\right) \mathbb{E}\left[\left\|(1-\beta)(\nabla f(\theta^{t-1}) - \tilde{m}_\tau^t)\right\|^2\right] + 2\beta^2\left(\gamma_t + \Lambda_t\right) +$$

$$+ 4\beta\gamma_t + 8\beta \left(\frac{L^2}{\tau} \sum_{k=t-\tau+1}^{t} \mathcal{U}_k + 2L^2\eta^2 \sum_{k=t-\tau+1}^{t-1}\left(\mathbb{E}\left[\left\|\nabla f(\theta^{k-1})\right\|^2\right] + \mathcal{E}_k\right)\right) \quad (67)$$

$$\overset{\text{lemma 4.6}}{\leq} \left(1 + \frac{\beta}{2}\right) \mathbb{E}\left[\left\|(1-\beta)(\nabla f(\theta^{t-1}) - \tilde{m}_\tau^t)\right\|^2\right] + \left(2\beta^2 + 4\beta\right)\gamma_t + 6\beta^2\frac{\sigma^2}{|\mathcal{S}_\tau^t|J} + \quad (68)$$

$$+ \left(6\beta^2 + 8\beta\right)\underbrace{\left(\frac{L^2}{\tau} \sum_{k=t-\tau+1}^{t} \mathcal{U}_k + 2L^2\eta^2 \sum_{k=t-\tau+1}^{t-1}\left(\mathbb{E}\left[\left\|\nabla f(\theta^{k-1})\right\|^2\right] + \mathcal{E}_k\right)\right)}_{\mathcal{T}_1}$$

$$\leq (1-\beta)^2 \left(1 + \frac{\beta}{2}\right) \mathbb{E}\left[\left\|\nabla f(\theta^{t-2}) - \tilde{m}_\tau^t + \nabla f(\theta^{t-1}) - \nabla f(\theta^{t-2})\right\|^2\right] + \quad (69)$$

$$+ 6\beta^2\frac{\sigma^2}{|\mathcal{S}_\tau^t|J} + 6\beta\gamma_t + 14\beta\mathcal{T}_1$$

Applying the AM-GM inequality again:

$$\leq (1-\beta)^2 \left(1 + \frac{\beta}{2}\right) \left[\left(1 + \frac{\beta}{4}\right) \mathbb{E}\left[\left\|\nabla f(\theta^{t-2}) - \tilde{m}_\tau^t\right\|^2\right] + \right. \quad (70)$$

$$\left. + \left(1 + \frac{1}{\beta}\right) \mathbb{E}\left[\left\|\nabla f(\theta^{t-1}) - \nabla f(\theta^{t-2})\right\|^2\right]\right] + 6\beta^2\frac{\sigma^2}{|\mathcal{S}_\tau^t|J} + 6\beta\gamma_t + 14\beta\mathcal{T}_1$$

$$\overset{\text{assumption 4.2}}{\leq} (1-\beta)^2 \left(1 + \frac{\beta}{2}\right) \left[\left(1 + \frac{\beta}{4}\right) \mathcal{E}_{t-1} + \right. \quad (71)$$

$$\left. + \left(1 + \frac{1}{\beta}\right) L^2 \mathbb{E}\left[\left\|\theta^{t-1} - \theta^{t-2}\right\|^2\right]\right] + 6\beta^2\frac{\sigma^2}{|\mathcal{S}_\tau^t|J} + 6\beta\gamma_t + 14\beta\mathcal{T}_1$$

$$\leq (1-\beta)^2 \left(1 + \frac{\beta}{2}\right) \left[\left(1 + \frac{\beta}{4}\right) \mathcal{E}_{t-1} + \right. \quad (72)$$

$$\left. + 2\left(1 + \frac{1}{\beta}\right) L^2\eta^2 \left(\mathbb{E}\left[\left\|\nabla f(\theta^{t-2})\right\|^2\right] + \mathcal{E}_{t-1}\right)\right] + 6\beta^2\frac{\sigma^2}{|\mathcal{S}_\tau^t|J} + 6\beta\gamma_t + 14\beta\mathcal{T}_1$$

Where in the last inequality we used the fact that:

$$\left\|\theta^{t-1} - \theta^{t-2}\right\|^2 \leq 2\eta^2 \left(\left\|\nabla f(\theta^{t-2})\right\|^2 + \left\|\nabla f(\theta^{t-2}) - \tilde{m}_\tau^t\right\|^2\right).$$

Now notice that $(1-\beta)^2 \left(1 + \frac{\beta}{2}\right)\left(1 + \frac{\beta}{4}\right) \leq (1-\beta)$ and that $2(1-\beta)^2 \left(1 + \frac{\beta}{2}\right)\left(1 + \frac{1}{\beta}\right) \leq \frac{2}{\beta}$:

$$\mathcal{E}_t \leq (1-\beta)\mathcal{E}_{t-1} + \frac{2}{\beta}L^2\eta^2 \left(\mathbb{E}\left[\left\|\nabla f(\theta^{t-2})\right\|^2\right] + \mathcal{E}_{t-1}\right) + 6\beta^2\frac{\sigma^2}{|\mathcal{S}_\tau^t|J} + 6\beta\gamma_t + 14\beta\mathcal{T}_1 \quad (73)$$

$$= \left(1 - \beta + \frac{2}{\beta}L^2\eta^2\right)\mathcal{E}_{t-1} + \frac{2}{\beta}L^2\eta^2 \mathbb{E}\left[\left\|\nabla f(\theta^{t-2})\right\|^2\right] + 6\beta^2\frac{\sigma^2}{|\mathcal{S}_\tau^t|J} + 6\beta\gamma_t + 14\beta\mathcal{T}_1 \quad (74)$$

Define:

- $\mathcal{T}_2 := L^2 T J \eta_l^2 \beta^2 \sigma^2 \left(1 + 2J^3\eta_l^2\beta^2 L^2\right)$

- $\mathcal{T}_3 := 2e^2 J^2 L^2 (2e\eta_l^2\beta\tau T G_\tau)$

- $\alpha_1 := 2J^2 L^2 e^2 \left(4\eta_l^2\left((1-\beta)^2 + e(\beta\eta LT)^2\right)\right) + 2L^2\eta_l^2(\tau - 1)$

Summing up over $T$ and substituting into $\mathcal{T}_1$ the expression for $\mathcal{U}_t$:

$$\sum_{t=1}^{T} \mathcal{E}_t \le \underbrace{\left(1 - \beta + \frac{2}{\beta}L^2\eta^2 + 14\beta\alpha_1\right)}_{\alpha_2} \sum_{t=0}^{T-1} \mathcal{E}_t + \tag{75}$$

$$+ \underbrace{\left(\frac{2}{\beta}L^2\eta^2 + 14\beta\alpha_1\right)}_{\alpha_3} \sum_{t=0}^{T-1} \mathbb{E}\left[\left\|\nabla f(\theta^{t-1})\right\|^2\right] +$$

$$+ 14\beta\left(\mathcal{T}_2 + \mathcal{T}_3\right)T + 6\beta^2 \frac{\sigma^2}{|\mathcal{S}_\tau^t|J}T + 6\beta \sum_{t=1}^{T} \gamma_t$$

We now have that:

$$\alpha_2 := \left(1 - \beta + \frac{2}{\beta}L^2\eta^2 + 14\beta\left[2J^2L^2e^2\left(4\eta_l^2\left((1-\beta)^2 + e(\beta\eta LT)^2\right)\right) + 2L^2\eta_l^2(\tau-1)\right]\right) \tag{76}$$

$$= \left(1 - \beta + \frac{2}{\beta}L^2\eta^2 + 14\beta\left[8J^2L^2e^2\eta_l^2\left((1-\beta)^2 + e(\beta\eta LT)^2\right) + 2L^2\eta_l^2(\tau-1)\right]\right) \tag{77}$$

$$\le \left(1 - \beta + \frac{2}{\beta}L^2\eta^2 + 112\beta e^2(\eta_l JL)^2\left[(1-\beta)^2 + (\beta\eta LT)^2 + (\tau-1)\right]\right) \tag{78}$$

$$\tag{79}$$

Now impose $(\eta_l JL) \le \left(37\sqrt{\tau}\beta\eta LTe\right)^{-1}$ and $\eta \le \frac{\beta}{\sqrt{8}L}$. We have that:

$$\alpha_2 \le \left(1 - \beta + \frac{2\beta}{8} + \frac{\beta}{8}\right) = \left(1 - \frac{5\beta}{8}\right) \tag{80}$$

$$\alpha_3 \le \frac{3\beta}{8} \tag{81}$$

$$14\beta\mathcal{T}_2 = 14\beta L^2 TJ\eta_l^2\beta^2\sigma^2\left(1 + 2J^3\eta_l^2\beta^2L^2\right) \tag{82}$$

$$= 14\beta^3(\eta_l JL)^2\left(\frac{1}{J} + 2(\eta_l JL\beta)^2\right)\sigma^2 T \tag{83}$$

$$\le 7\beta^2 \frac{\sigma^2}{|\mathcal{S}_\tau^t|J}T \tag{84}$$

Where in the last inequality we apply:

$$2\beta(\eta_l JL)^2\left(\frac{1}{J} + 2(\eta_l JL\beta)^2\right) \le \frac{1}{|\mathcal{S}_\tau^t|J}$$

Plugging all the terms together we have:

$$\sum_{t=1}^{T} \mathcal{E}_t \le \left(1 - \frac{5}{8\beta}\right) \sum_{t=0}^{T-1} \mathcal{E}_t + \frac{3\beta}{8} \sum_{t=0}^{T-1} \mathbb{E}\left[\left\|\nabla f(\theta^{t-1})\right\|^2\right] + 13\beta^2 \frac{\sigma^2}{|\mathcal{S}_\tau^t|J}T + \tag{85}$$

$$+ 56\beta(\eta_l JL)^2(e^3\tau T)G_\tau + 6\beta \sum_{t=1}^{T} \gamma_t$$

Rearranging the terms completes the proof. $\qquad\square$

**Lemma B.6.** *Under assumptions 4.1-4.2, for definition 11 it holds that:*

$$\mathcal{U}_t \le 2J^2e^2\Xi_t + J\eta_l^2\beta^2\sigma^2(1 + 2J^3\eta_l^2L^2\beta^2) \tag{86}$$

$$\sum_{t=1}^{T} \mathcal{U}_t \le TJ\eta_l^2\beta^2\sigma^2(1 + 2J^3\eta_l^2\beta^2L^2) + 2J^2e^2 \sum_{t=1}^{T} \Xi_t \tag{87}$$

*Proof.*

$$\mathbb{E}\left[\left\|\theta_i^{t,j} - \theta^{t-1}\right\|^2\right] \leq 2\mathbb{E}\left[\left\|\sum_{k=0}^{j-1}\zeta_i^{t,k}\right\|^2\right] + 2j\eta_l^2\beta^2\sigma^2 \tag{88}$$

$$\stackrel{\text{lemma B.4}}{\leq} 2j\sum_{k=0}^{j-1}\mathbb{E}\left[\left\|\zeta_i^{t,k}\right\|^2\right] + 2j\eta_l^2\beta^2\sigma^2 \tag{89}$$

For any $1 \leq k \leq j - 1 \leq J - 2$, using $\eta L \leq \frac{1}{\beta J} \leq \frac{1}{\beta(j+1)}$, we have:

$$\mathbb{E}\left[\left\|\zeta_i^{t,k}\right\|^2\right] \leq \left(1 + \frac{1}{j}\right)\mathbb{E}\left[\left\|\zeta_i^{t,k-1}\right\|^2\right] + (1+j)\mathbb{E}\left[\left\|\zeta_i^{t,k} - \zeta_i^{t,k-1}\right\|^2\right] \tag{90}$$

$$\leq \left(1 + \frac{1}{j}\right)\mathbb{E}\left[\left\|\zeta_i^{t,k-1}\right\|^2\right] + (1+j)\eta_l^2\beta^2 L^2\left(\eta_l^2\beta^2\sigma^2 + \mathbb{E}\left[\left\|\zeta_i^{t,k-1}\right\|^2\right]\right) \tag{91}$$

$$\leq \left(1 + \frac{1}{j}\right)\mathbb{E}\left[\left\|\zeta_i^{t,k-1}\right\|^2\right] + (1+j)\eta_l^4\beta^4 L^2\sigma^2 + \frac{1}{1+j}\mathbb{E}\left[\left\|\zeta_i^{t,k} - \zeta_i^{t,k-1}\right\|^2\right] \tag{92}$$

$$\leq \left(1 + \frac{2}{j}\right)\mathbb{E}\left[\left\|\zeta_i^{t,k-1}\right\|^2\right] + (1+j)\eta_l^4\beta^4 L^2\sigma^2 \tag{93}$$

$$\stackrel{(1+\frac{2}{j})^j \leq e^2}{\leq} e^2\mathbb{E}\left[\left\|\zeta_i^{t,0}\right\|^2\right] + 4j^2\eta_l^4\beta^4 L^2\sigma^2 \tag{94}$$

So it holds that:

$$\mathbb{E}\left[\left\|\theta_i^{t,j} - \theta^{t-1}\right\|^2\right] \leq 2j^2\left(e^2\mathbb{E}\left[\left\|\zeta_i^{t,0}\right\|^2\right] + 4j^2\eta_l^4 L^2\sigma^2\right) + 2j\eta_l^2\sigma^2 \tag{95}$$

$$= 2e^2 j^2\mathbb{E}\left[\left\|\zeta_i^{t,0}\right\|^2\right] + 2j\eta_l^2\sigma^2\beta^2(1 + 4j^3\eta_l^2 L^2\beta^2) \tag{96}$$

So, summing up over $i$ and $j$:

$$\mathcal{U}_t \leq \frac{1}{|\mathcal{S}|J}\sum_{i=1}^{|\mathcal{S}|}\sum_{j=1}^{J} 2e^2 j^2\mathbb{E}\left[\left\|\zeta_i^{t,0}\right\|^2\right] + 2j\eta_l^2\sigma^2\beta^2(1 + 4j^3\eta_l^2 L^2\beta^2) \tag{97}$$

$$\leq 2J^2 e^2\Xi_t + J\eta_l^2\beta^2\sigma^2(1 + 2J^3\eta_l^2 L^2\beta^2) \tag{98}$$

Finally, summing up over $T$:

$$\sum_{t=1}^{T}\mathcal{U}_t \leq \underbrace{TJ\eta_l^2\beta^2\sigma^2(1 + 2J^3\eta_l^2\beta^2 L^2)}_{\mathcal{T}_1} + 2J^2 e^2\sum_{t=1}^{T}\Xi_t \tag{99}$$

$$\leq \mathcal{T}_1 + 2J^2 e^2\left(4\eta^2\left((1-\beta)^2 + e(\beta\eta LT)^2\right)\sum_{t=1}^{T-1}\left(\mathcal{E}_t + \mathbb{E}\left[\left\|\nabla f(\theta^{t-1})\right\|^2\right]\right) + \underbrace{2e\eta^2\beta^2\tau TG_\tau}_{\mathcal{T}_2}\right) \tag{100}$$

$$\leq \mathcal{T}_1 + \alpha_1\sum_{t=1}^{T-1}\left(\mathcal{E}_t + \mathbb{E}\left[\left\|\nabla f(\theta^{t-1})\right\|^2\right]\right) + \alpha_2\mathcal{T}_2 \tag{101}$$

$\square$

**Lemma B.7.** *Under assumptions 4.1-4.2-4.3, if* $224e(\eta_l JL)^2\left((1-\beta)^2 + e(\beta\eta LT)^2\right) \leq 1$, *for definition 13 it holds for* $t \geq 0$ *that:*

$$\Xi_t \leq \frac{1}{56eJ^2 L^2}\sum_{t=0}^{T-1}\left(\mathcal{E}_t + \mathbb{E}\left[\left\|\nabla f(\theta^{t-1})\right\|^2\right]\right) + 2e\eta_l^2\beta^2\tau TG_\tau \tag{102}$$

*Proof.* Note that $\zeta_i^{t,0} = -\eta_l\left((1-\beta)\tilde{m}_\tau^t + \beta g_i(\theta^{t-1})\right)$,

$$\frac{1}{|\mathcal{S}|}\sum_{i=1}^{|\mathcal{S}|}\left\|\zeta_i^{t,0}\right\|^2 \leq 2\eta_l^2\left((1-\beta)^2\left\|\tilde{m}_\tau^t\right\|^2 + \frac{\beta^2}{|\mathcal{S}|}\sum_{i=1}^{|\mathcal{S}|}\left\|g_i(\theta^{t-1})\right\|^2\right) \tag{103}$$

For any $a > 0$, considering each client participates to the train every $\tau = \frac{1}{C}$ rounds:

$$\mathbb{E}\left[\left\|g_i(\theta^{t-1})\right\|^2\right] = \mathbb{E}\left[\left\|g_i(\theta^{t-1}) - g_i(\theta^{t-\tau-1}) + g_i(\theta^{t-\tau-1})\right\|^2\right] \tag{104}$$

$$\overset{\text{lemma B.4}}{\leq} (1+a)\mathbb{E}\left[\left\|g_i(\theta^{t-\tau-1})\right\|^2\right] + \tag{105}$$

$$+ \left(1 + \frac{1}{a}\right)\mathbb{E}\left[\left\|g_i(\theta^{t-1}) - g_i(\theta^{t-\tau-1})\right\|^2\right]$$

$$\leq (1+a)\mathbb{E}\left[\left\|g_i(\theta^{t-\tau-1})\right\|^2\right] + \tag{106}$$

$$+ \left(1 + \frac{1}{a}\right)L^2\mathbb{E}\left[\left\|\theta^{t-1} - \theta^{t-\tau-1}\right\|^2\right] \tag{107}$$

$$\leq (1+a)\mathbb{E}\left[\left\|g_i(\theta^{t-\tau-1})\right\|^2\right] + \tag{108}$$

$$+ 2\left(1 + \frac{1}{a}\right)L^2\eta^2\tau\sum_{k=1}^{\tau}\left(\mathcal{E}_{t-k} + \mathbb{E}\left[\left\|\nabla f(\theta^{t-k-1})\right\|^2\right]\right) \tag{109}$$

$$\leq (1+a)^{\frac{t}{\tau}}\mathbb{E}\left[\left\|g_i(\theta^{t_i-1})\right\|^2\right] + \tag{110}$$

$$+ 2\left(1 + \frac{1}{a}\right)L^2\eta^2\tau\sum_{s=1}^{\frac{t}{\tau}}\sum_{k=1}^{\tau}\left(\mathcal{E}_{s\tau-k} + \mathbb{E}\left[\left\|\nabla f(\theta^{s\tau-k})\right\|^2\right]\right)(1+a)^{\frac{t}{\tau}-s}$$

$$\leq (1+a)^{\frac{t}{\tau}}\mathbb{E}\left[\left\|g_i(\theta^{t_i-1})\right\|^2\right] + \tag{111}$$

$$+ 2\left(1 + \frac{1}{a}\right)L^2\eta^2\tau\sum_{k=1}^{t-1}\left(\mathcal{E}_k + \mathbb{E}\left[\left\|\nabla f(\theta^{k-1})\right\|^2\right]\right)(1+a)^{\frac{t}{\tau}}$$

Where $t_i := \min_{t\in[T]}(t \text{ s.t. } i \in \mathcal{S}^t)$. Now take $a = \frac{\tau}{t}$:

$$\mathbb{E}\left[\left\|g_i(\theta^{t-1})\right\|^2\right] \leq e\mathbb{E}\left[\left\|g_i(\theta^{t_i-1})\right\|^2\right] + \tag{112}$$

$$+ 2e\eta^2L^2\tau\left(\frac{t}{\tau}+1\right)\sum_{k=1}^{t-1}\left(\mathcal{E}_k + \mathbb{E}\left[\left\|\nabla f(\theta^{k-1})\right\|^2\right]\right)$$

So:

$$\sum_{t=1}^{T}\Xi_t \leq \sum_{t=1}^{T}2\eta_l^2\left(2(1-\beta)^2\left(\mathcal{E}_{t-1} + \mathbb{E}\left[\left\|\nabla f(\theta^{t-2}\right\|^2\right]\right) + \frac{\beta^2}{|\mathcal{S}|}\sum_{i=1}^{|\mathcal{S}|}\mathbb{E}\left[\left\|g_i(\theta^{t-1})\right\|^2\right]\right) \tag{113}$$

$$\leq \sum_{t=1}^{T}4\eta_l^2(1-\beta)^2\left(\mathcal{E}_{t-1} + \mathbb{E}\left[\left\|\nabla f(\theta^{t-2})\right\|^2\right]\right) + \tag{114}$$

$$+ 2\eta_l^2\beta^2\sum_{t=1}^{T}\left(\frac{e}{|\mathcal{S}|}\sum_{i=1}^{|\mathcal{S}|}\mathbb{E}\left[\left\|g_i(\theta^{t_i-1})\right\|^2\right] + 2e\eta_l^2L^2\tau\left(\frac{t}{\tau}+1\right)\sum_{k=1}^{t-1}\left(\mathcal{E}_k + \mathbb{E}\left[\left\|\nabla f(\theta^{t-1}\right\|^2\right]\right)\right)$$

$$\leq 4\eta_l^2(1-\beta)^2\sum_{t=1}^{T}\left(\mathcal{E}_{t-1} + \mathbb{E}\left[\left\|\nabla f(\theta^{t-2})\right\|^2\right]\right) + \tag{115}$$

$$+ 2\eta_l^2\beta^2\left(eT\sum_{t=1}^{\tau}G_t + 2e(\eta LT)^2\sum_{t=1}^{T-1}\left(\mathcal{E}_t + \mathbb{E}\left[\left\|\nabla f(\theta^{t-1})\right\|^2\right]\right)\right)$$

Let us define $G_\tau := \max_{t \in [1,\tau]} G_t$, with $G_t := \frac{1}{|\mathcal{S}^t|} \sum_{i=1}^{|\mathcal{S}^t|} \mathbb{E}\left[\left\|g_i(\theta^{t-1})\right\|^2\right]$. We have that:

$$\sum_{t=1}^{T} \Xi_t \leq 4\eta_l^2 \left((1-\beta)^2 + e(\beta\eta LT)^2\right) \sum_{t=0}^{T-1} \left(\mathcal{E}_t + \mathbb{E}\left[\left\|\nabla f(\theta^{t-1})\right\|^2\right]\right) + 2e\eta_l^2 \beta^2 \tau T G_\tau \quad (116)$$

Applying the upper bound of $\eta_l$ completes the proof. $\qquad \square$

**Lemma B.8** (Cheng et al. (2024)). *Under assumption 4.2, if $\eta L \leq \frac{1}{24}$, the following holds for all $t \geq 0$:*

$$\mathbb{E}\left[f(\theta^t)\right] \leq \mathbb{E}\left[f(\theta^{t-1})\right] - \frac{11\eta}{24}\mathbb{E}\left[\left\|\nabla f(\theta^{t-1})\right\|^2\right] + \frac{13\eta}{24}\mathcal{E}_t \quad (117)$$

*Proof.* Since f is $L$-smooth, we have:

$$f(\theta^t) \leq f(\theta^{t-1}) + \left\langle \nabla f(\theta^{t-1}), \theta^t - \theta^{t-1} \right\rangle + \frac{L}{2}\left\|\theta^t - \theta^{t-1}\right\|^2 \quad (118)$$

$$= f(\theta^{t-1}) - \eta\left\|\nabla f(\theta^{t-1})\right\|^2 + \eta\left\langle \nabla f(\theta^{t-1}), \nabla f(\theta^{t-1}) - \tilde{m}_\tau^{t+1}\right\rangle + \frac{L\eta^2}{2}\left\|\tilde{m}_\tau^{t+1}\right\|^2 \quad (119)$$

Since $\theta^t = \theta^{t-1} - \eta\tilde{m}_\tau^{t+1}$, using Young's inequality and imposing $\eta L \leq \frac{1}{24}$, we further have:

$$f(\theta^t) \leq f(\theta^{t-1}) - \frac{\eta}{2}\left\|\nabla f(\theta^{t-1})\right\|^2 + \frac{\eta}{2}\left\|\nabla f(\theta^{t-1}) - \tilde{m}_\tau^{t+1}\right\|^2 + \quad (120)$$

$$+ L\eta^2\left(\left\|\nabla f(\theta^{t-1})\right\|^2 + \left\|\nabla f(\theta^{t-1}) - \tilde{m}_\tau^{t+1}\right\|^2\right)$$

$$\leq f(\theta^{t-1}) - \frac{11\eta}{24}\left\|\nabla f(\theta^{t-1})\right\|^2 + \frac{13\eta}{24}\left\|\nabla f(\theta^{t-1}) - \tilde{m}_\tau^{t+1}\right\|^2 \quad (121)$$

$$\square$$

**Proof of Theorem 4.7** (Convergence rate of GHBM for non-convex functions)

*Under assumptions 4.1-4.2-4.3, if we take:*

$$\tilde{m}_\tau^0 = 0, \qquad \beta = \min\left\{1, \sqrt{\frac{|\mathcal{S}|JL\Delta}{\sigma^2 T}}\right\}, \qquad \eta = \min\left\{\frac{1}{24L}, \frac{\beta}{\sqrt{8}L}\right\} \quad (122)$$

$$\eta_l JL \lesssim \min\left\{1, \frac{1}{\beta\eta L\sqrt{\tau}T}, \sqrt{\frac{L\Delta}{\beta^3\tau G_\tau T}}, \frac{1}{\sqrt{\beta|\mathcal{S}|}}, \left(\frac{1}{\beta^3|\mathcal{S}|J}\right)^{\frac{1}{4}}\right\} \quad (123)$$

*then GHBM with optimal $\tau = \frac{1}{C}$ converges as:*

$$\frac{1}{T}\sum_{t=1}^{T}\mathbb{E}\left[\left\|\nabla f(\theta^{t-1})\right\|^2\right] \lesssim \frac{L\Delta}{T} + \sqrt{\frac{L\Delta\sigma^2}{|\mathcal{S}|JT}} \quad (123)$$

*Proof.* Combining the results of lemma B.5 and lemma B.8, we have that:

$$\sum_{t=1}^{T}\left(\mathbb{E}\left[f(\theta^t)\right] - \mathbb{E}\left[f(\theta^{t-1})\right]\right) \leq -\frac{11\eta}{24}\sum_{t=1}^{T}\mathbb{E}\left[\left\|\nabla f(\theta^{t-1})\right\|^2\right] + \frac{13\eta}{24}\sum_{t=1}^{T}\mathcal{E}_t \quad (124)$$

$$\frac{1}{\eta}\mathbb{E}\left[f(\theta^{t-1}) - f(\theta^0)\right] \leq \frac{26}{30\beta}\mathcal{E}_0 - \frac{1}{15}\sum_{t=1}^{T}\mathbb{E}\left[\left\|\nabla f(\theta^{t-1})\right\|^2\right] + 32\beta\frac{\sigma^2}{|\mathcal{S}_\tau^t|J}T + \quad (125)$$

$$+ \frac{448}{5}(\eta_l JL)^2(e^3\tau T)G_\tau + 6\beta\sum_{t=1}^{T}\gamma_t \quad (126)$$

Imposing $\tau = \frac{1}{C}$, by corollary 4.5 we have that $\gamma_t = 0$ and $\mathcal{S}_\tau^t = \mathcal{S} \ \forall t$. Also, noticing that $\tilde{m}_\tau^0 = 0$ implies $\mathcal{E}_0 \leq 2L\left(f(\theta^0) - f^*\right) = 2L\Delta$, we have that:

$$\frac{1}{T}\sum_{t=1}^{T}\mathbb{E}\left[\left\|\nabla f(\theta^{t-1})\right\|^2\right] \lesssim \frac{L\Delta}{\eta LT} + \frac{\mathcal{E}_0}{\beta T} + (\eta_l JL\beta)^2\tau G_\tau + \beta\frac{\sigma^2}{|\mathcal{S}|J} \tag{127}$$

$$\lesssim \frac{L\Delta}{T} + \frac{2L\Delta}{\beta T} + (\eta_l JL\beta)^2\tau G_\tau + \beta\frac{\sigma^2}{|\mathcal{S}|J} \tag{128}$$

$$\lesssim \frac{L\Delta}{T} + \frac{L\Delta}{\beta T} + \beta\frac{\sigma^2}{|\mathcal{S}|J} \tag{129}$$

$$\lesssim \frac{L\Delta}{T} + \sqrt{\frac{L\Delta\sigma^2}{|\mathcal{S}|JT}} \tag{130}$$

$\square$

## C  Experimental Setting

### C.1  Datasets and Models

**Cifar-10/100**  We consider Cifar-10 and Cifar-100 to experiment with image classification tasks, each one respectively having 10 and 100 classes. For all methods, training images are pre-processed by applying random crops, followed by random horizontal flips. Both training and test images are finally normalized according to their mean and standard deviation. As the main model for experimentation, we used a model similar to LeNet-5 as proposed in (Hsu et al., 2020). To further validate our findings, we also employed a ResNet-20 as described in (He et al., 2015), following the implementation provided in (Idelbayev, 2021). Since batch normalization Ioffe & Szegedy (2015) layers have been shown to hamper performance in learning from decentralized data with skewed label distribution (Hsieh et al., 2020), we replaced them with group normalization (Wu & He, 2018), using two groups in each layer. For a fair comparison, we used the same modified network also in centralized training. We report the result of centralized training for reference in Table 5: as per the hyperparameters, we use 64 for the batch size, 0.01 and 0.1 for the learning rate respectively for the LeNet and the ResNet-20 and 0.9 for momentum. We trained both models on both datasets for 150 epochs using a cosine annealing learning rate scheduler.

**Shakespeare**  The Shakespeare language modeling dataset is created by collating the collective works of William Shakespeare and originally comprises 715 clients, with each client denoting a speaking role. However, for this study, a different approach was used, adopting the LEAF (Caldas et al., 2019) framework to split the dataset among 100 devices and restrict the number of data points per device to 2000. The non-IID dataset is formed by assigning each device to a specific role, and the local dataset for each device contains the sentences from that role. Conversely, the IID dataset is created by randomly distributing sentences from all roles across the devices.

Table 5: Test accuracy (%) of centralized training over datasets and models used. Results are reported in term of mean top-1 accuracy over the last 10 epochs, averaged over 5 independent runs.

| Dataset | Acc. Centralized (%) |
|---|---|
| Cifar-10 w/ LeNet | $86.48 \pm 0.22$ |
| Cifar-10 w/ ResNet-20 | $89.05 \pm 0.44$ |
| Cifar-100 w/ LeNet | $57.00 \pm 0.09$ |
| Cifar-100 w/ ResNet-20 | $62.21 \pm 0.85$ |
| Shakespeare | $52.00 \pm 0.16$ |
| StackOverflow | $28.50 \pm 0.25$ |
| GLDv2 | $74.03 \pm 0.15$ |

For this task, we have employed a two-layer Long Short-Term Memory (LSTM) classifier, consisting of 100 hidden units and an 8-dimensional embedding layer. Our objective is to predict the next character in a sequence, where there are a total of 80 possible character classes. The model takes in a sequence of 80 characters as input, and for each character, it learns an 8-dimensional representation. The final output of the model is a single character prediction for each training example, achieved through the use of 2 LSTM layers and a densely-connected layer followed by a softmax. This model architecture is the same used by (Li et al., 2020; Acar et al., 2021).

We report the result of centralized training for reference in Table 5: we train for 75 epochs with constant learning rate, using as hyperparameters 100 for the batch size, 1 for the learning rate, 0.0001 for the weight decay and no momentum.

**StackOverflow** The Stack Overflow dataset is a language modeling corpus that comprises questions and answers from the popular Q&A website, StackOverflow. Initially, the dataset consists of 342477 unique users but for, practical reasons, we limit our analysis to a subset of $40k$ users. Our goal is to perform the next-word prediction on these text sequences. To achieve this, we utilize a Recurrent Neural Network (RNN) that first learns a 96-dimensional representation for each word in a sentence and then processes them through a single LSTM layer with a hidden dimension of 670. Finally, the model generates predictions using a densely connected softmax output layer. The model and the preprocessing steps are the same as in (Reddi et al., 2021).

We report the result of centralized training for reference in Table 5: as per the hyperparameters, we use 16 for the batch size, $10^{-1/2}$ for the learning rate and no momentum or weight decay. We train for 50 epochs with a constant learning rate.

Given the size of the test dataset, testing on STACKOVERFLOW is conducted on a subset of them made by 10000 randomly chosen test examples, selected at the beginning of training.

**Large-scale real-world datasets** As large-scale real-world datasets for our experimentation, we follow Hsu et al. (2020). GLDv2 is composed of $\approx 164k$ images belonging to $\approx 2000$ classes, realistically split among 1262 clients. INATURALIST is composed of $\approx 120k$ images belonging to $\approx 1200$ classes, split among 9275 clients. These datasets are challenging to train not only because of their inherent complexity (size of images, number of classes) but also because usually at each round a very small portion of clients is selected. In particular, for GLDv2 we sample 10 clients per round, while for INATURALIST we experiment with different participation rates, sampling 10, 50, or 100 clients per round. In the main paper, we choose to report the participation rate instead of the number of sampled clients to better highlight that the tested scenarios are closer to a cross-device setting, which is the most challenging for algorithms based on client participation, like SCAFFOLD and ours. As per the model, for both datasets, we use a MobileNetV2 pretrained on ImageNet.

Table 6: Details about datasets' split used for our experiments

|  | CIFAR-10 | CIFAR-100 | SHAKESPEARE | STACKOVERFLOW | GLDv2 | INATURALIST |
|---|---|---|---|---|---|---|
| Clients | 100 | 100 | 100 | 40.000 | 1262 | 9275 |
| Number of clients per round | 10 | 10 | 10 | 50 | 10 | $\{10, 50, 100\}$ |
| Number of classes | 10 | 100 | 80 | 10004 | 2028 | 1203 |
| Avg. examples per client | 500 | 500 | 2000 | 428 | 130 | 13 |
| Number of local steps | 8 | 8 | 20 | 27 | 13 | 2 |
| Average participation (round no.) | 1k | 1k | 25 | 1.5 | 40 | $\{5, 27, 54\}$ |

## C.2 SIMULATING HETEROGENEITY

For CIFAR-10/100 we simulate arbitrary heterogeneity by splitting the total datasets according to a Dirichlet distribution with concentration parameter $\alpha$, following Hsu et al. (2020). In practice, we draw a multinomial $q_i \sim \mathbf{Dir}(\alpha p)$ from a Dirichlet distribution, where $p$ describes a prior class distribution over $N$ classes, and $\alpha$ controls the heterogeneity among all clients: the greater $\alpha$ the more homogeneous the clients' data distributions will be. After drawing the class distributions $q_i$, for every client $i$, we sample training examples for each class according to $q_i$ without replacement.

In the main paper, we considered only two levels of heterogeneity: the first uses $\alpha = 0$ and is used to simulate a pathological non-iid scenario, while the second uses $\alpha = 10k$ and corresponds to having homogeneous local datasets. To further investigate the impact of heterogeneity, we provide the results for different values of $\alpha$ in section C.6 of this supplementary.

## C.3 EVALUATING COMMUNICATION AND COMPUTATIONAL COST

In the main paper we showed a comparison in communication and computational cost of state-of-art FL algorithms compared to our solutions GHBM and FEDHBM: in this section we detail how those results in table Tab. 4 have been obtained. We follow a three-step procedure:

1. For each algorithm $a$, we calculate the minimum number of rounds $r_a$ to reach the performance of FEDAVG, the total amount of bytes exchanged $b_a$ in the whole training budget

(number of rounds, as described in Appendix C.5) and the measure the corresponding total training time $t_a$. In this way, the different requirements in communication and computation of each algorithm are taken into account for the next steps.

2. We calculate the actual communication and computational requirements as $(tb_a = b_a \cdot s_a, tt_a = t_a \cdot s_a)$, where $s_a = \frac{r_a}{T}$ is the speedup of the algorithm w.r.t. FEDAVG. For those competitor algorithms that did not reach the target performance (*e.g.*MIMEMOM) in the training budget $T$, we conservatively consider $r_a = T$. In this way, the convergence speed of each algorithm is taken into account for determining the actual amount of computation needed.

3. We complement the above information with with a reduction/increase factor w.r.t. FEDAVG, calculated as $rtb_a = \left(1 - \frac{tb_a}{tb_{\text{FEDAVG}}}\right)$ and $rtt_a = \left(1 - \frac{tt_a}{tt_{\text{FEDAVG}}}\right)$ and expressed as a percentage. A cost reduction (*i.e.* $rtb_a > 0$ or $rtt_a > 0$) is indicated with ↓, while a cost increase (*i.e.* $rtb_a < 0$ or $rtt_a < 0$) is indicated with ↑. This gives a practical indication of how much communication/computation have been saved in choosing the algorithm at hand as an alternative for FEDAVG.

## C.4 HYPERPARAMETERS

For ease of consultation, we report the hyper-parameters grids as well as the chosen values in Table 7. For GLDv2 and INATURALIST we only test the best SOTA algorithms: FEDAVG and FEDAVGM as baselines, SCAFFOLD and MIMEMOM.

**MOBILENETV2** For all algorithms we perform $E = 5$ local epochs, and searched $\eta \in \{0.1, 1\}$ and $\eta_l \in \{0.01, 0.1\}$, and found $\eta = 0.1, \eta_l = 0.1$ works best for FEDAVGM, while $\eta = 1, \eta_l = 0.1$ works best for the others. For INATURALIST, we had to enlarge the grid for SCAFFOLD and MIMEMOM: for both we searched $\eta \in \{10^{-3/2}, 10^{-1}, 10^{-1/2}, 1\}$ and $\eta_l \in \{10^{-2}, 10^{-3/2}, 10^{-1}, 10^{-1/2}\}$.

**VIT-B\16** For all algorithms we perform $E = 5$ local epochs, and searched $\eta \in \{0.1, 1\}$ and $\eta_l \in \{0.03, 0.01\}$ following (Steiner et al., 2022), and found $\eta = 0.1, \eta_l = 0.03$ works best for FEDAVGM, while $\eta = 1, \eta_l = 0.03$ works best for the others.

## C.5 IMPLEMENTATION DETAILS

We implemented all the tested algorithms and training procedures in a single codebase, using PYTORCH 1.10 framework, compiled with CUDA 10.2. The federated learning setup is simulated by using a single node equipped with 11 Intel(R) Core(TM) i7-6850K CPUS and 4 NVIDIA GeForce GTX 1070 GPUS. For the large-scale experiments we used the computing capabilities offered by LEONARDO cluster of CINECA-HPC, employing nodes equipped with 1 CPU Intel(R) Xeon 8358 32 core, 2,6 GHz CPUS and 4 NVIDIA A100 SXM6 64GB (VRAM) GPUS. The simulation always runs in a sequential manner (on a single GPU) the parallel client training and the following aggregation by the central server.

**Practicality of experiments** Under the above conditions, a single FEDAVG experiment on CIFAR-100 takes $\approx$ 02:05 hours (CNN, with $T = 20.000$) and $\approx$ 03:36 hours (RESNET-20, with $T = 10.000$). For SCAFFOLD we always use the `"option II"` of their algorithm (Karimireddy et al., 2020) to calculate the client controls, incurring almost no overhead in our simulations. We found that using `"option I"` usually degrades both final model quality and requires almost double the training time, due to the additional forward+backward passes. Conversely, all MIME's methods incur a significant overhead due to the additional round needed to calculate the full-batch gradients, taking $\approx$ 10:40 hours for CIFAR-100 with RESNET-20. On SHAKESPEARE and STACKOVERFLOW, FEDAVG takes $\approx$ 22 minutes and $\approx$ 3.5 hours to run respectively $T = 250$ and $T = 1500$ rounds.

## C.6 ADDITIONAL EXPERIMENTS

**Experiments on CIFAR-10** Table 8 reports the results of experiments analogous to the ones presented in Tab. 2. For the main paper, we report experiments on CIFAR-100, as it is a more complex dataset and often a more reliable testing ground for FL algorithms. Indeed, sometimes algorithms perform well on CIFAR-10 but worse on CIFAR-100 (as for the already discussed case of FEDDYN). Results in Tab. 8 confirm the findings of the main paper: under extreme heterogeneity, some algorithms behave inconsistently across CNN and RESNET-20 (notice that FEDDYN and

Table 7: Hyper-parameter search grid for each combination of method and dataset (for $\alpha = 0$). The best values are indicated in **bold**.

| METHOD | HPARAM | CIFAR-10/100 | | SHAKESPEARE | STACKOVERFLOW |
|---|---|---|---|---|---|
| | | LENET | RESNET-20 | | |
| ALL FL | wd | [**0.001**, 0.0008, 0.0004] | [0.0001, **0.00001**] | [**0**, 0.0001, 0.00001] | [**0**, 0.0001, 0.00001] |
| | $B$ | 64 | 64 | 100 | 16 |
| FEDAVG | $\eta$ | [**1**, 0.5, 0.1] | [**1**, 0.1] | [**1**, 0.5, 0.1] | [**1**, 0.5, 0.1] |
| | $\eta_l$ | [0.1, 0.05, **0.01**] | [**0.1**, 0.01] | [**1**, 0.5, 0.1] | [1, 0.5, **0.3**, 0.1] |
| FEDPROX | $\eta$ | [**1**, 0.5, 0.1] | [**1**, 0.1] | [**1**, 0.5, 0.1] | [**1**, 0.5, 0.1] |
| | $\eta_l$ | [0.1, 0.05, **0.01**] | [**0.1**, 0.01] | [**1**, 0.5, 0.1] | [1, 0.5, **0.3**, 0.1] |
| | $\mu$ | [**0.1**, 0.01, 0.001] | [**0.1**, 0.01, 0.001] | [0.1, 0.01, 0.001, **0.0001**] | [0.1, **0.01**, 0.001, 0.0001] |
| SCAFFOLD | $\eta$ | [**1**, 0.5, 0.1] | [**1**, 0.1] | [**1**, 0.5, 0.1] | [**1**, 0.5, 0.1] |
| | $\eta_l$ | [0.1, 0.05, **0.01**] | [**0.1**, 0.01] | [**1**, 0.5, 0.1] | [1, 0.5, **0.3**, 0.1] |
| FEDDYN | $\eta$ | [**1**, 0.5, 0.1] | [**1**, 0.1] | [**1**, 0.5, 0.1] | [**1**, 0.5, 0.1] |
| | $\eta_l$ | [0.1, 0.05, **0.01**] | [0.1, **0.01**] | [**1**, 0.5, 0.1] | [1, 0.5, **0.3**, 0.1] |
| | $\alpha$ | [0.1, 0.01, **0.001**] | [0.1, 0.01, **0.001**] | [0.1, **0.009**, 0.001] | [**0.1**, 0.009, 0.001] |
| ADABEST | $\eta$ | [**1**, 0.5, 0.1] | [**1**, 0.5, 0.1] | [**1**, 0.5, 0.1] | [**1**, 0.5, 0.1] |
| | $\eta_l$ | [0.1, 0.05, **0.01**] | [0.1, 0.05, **0.01**] | [**1**, 0.5, 0.1] | [1, 0.5, **0.3**, 0.1] |
| | $\alpha$ | [0.1, 0.01, **0.001**] | [0.1, 0.01, **0.001**] | [0.1, **0.009**, 0.001] | [**0.1**, 0.009, 0.001] |
| MIME | $\eta$ | [**1**, 0.5, 0.1] | [**1**, 0.1] | [**1**, 0.5, 0.1] | [**1**, 0.5, 0.1] |
| | $\eta_l$ | [0.1, 0.05, **0.01**] | [0.1, **0.01**] | [**1**, 0.5, 0.1] | [1, 0.5, **0.3**, 0.1] |
| FEDAVGM | $\eta$ | [1, 0.5, **0.1**] | [1, **0.1**] | [1, 0.5, **0.1**] | [**1**, 0.5, 0.1] |
| | $\eta_l$ | [0.1, 0.05, **0.01**] | [**0.1**, 0.01] | [**1**, 0.5, 0.1] | [1, 0.5, **0.3**, 0.1] |
| | $\beta$ | [0.99, **0.9**] | [0.99, **0.9**] | [0.99, **0.9**] | [0.99, **0.9**] |
| MIMEMOM | $\eta$ | [1, 0.5, **0.1**] | [1, 0.5, 0.3, **0.1**] | [1, 0.5, **0.1**] | [**1**, 0.5, 0.1] |
| | $\eta_l$ | [0.1, 0.05, **0.01**] | [**0.1**, 0.05, 0.03, 0.01] | [**1**, 0.5, 0.1] | [1, 0.5, 0.3, **0.1**] |
| | $\beta$ | [0.99, 0.95, **0.9**] | [0.99, 0.95, **0.9**] | [0.99, **0.9**] | [0.99, **0.9**] |
| MIMELITEMOM | $\eta$ | [1, 0.5, **0.1**] | [1, 0.5, 0.3, **0.1**] | [1, 0.5, **0.1**] | [**1**, 0.5, 0.1] |
| | $\eta_l$ | [0.1, 0.05, **0.01**] | [**0.1**, 0.05, 0.03, 0.01] | [**1**, 0.5, 0.1] | [1, 0.5, 0.3, **0.1**] |
| | $\beta$ | [0.99, **0.9**] | [0.99, 0.95, **0.9**] | [0.99, **0.9**] | [0.99, **0.9**] |
| FEDCM | $\eta$ | [1, 0.5, **0.1**] | [**1**, 0.5, 0.1] | [**1**, 0.5, 0.1] | - |
| | $\eta_l$ | [1, 0.5, **0.1**] | [1, 0.5, **0.1**] | [**1**, 0.5, 0.1] | - |
| | $\alpha$ | [0.05, **0.1**, 0.5] | [0.05, **0.1**, 0.5] | [0.05, **0.1**, 0.5] | - |
| **GHBM (ours)** | $\eta$ | [**1**, 0.5, 0.1] | [**1**, 0.1] | [**1**, 0.5, 0.1] | [**1**, 0.5, 0.1] |
| | $\eta_l$ | [**1**, 0.5, 0.1] | [1, 0.5, **0.1**] | [**1**, 0.5, 0.1] | [1, 0.5, **0.3**, 0.1] |
| | $\beta$ | [**0.9**] | [**0.9**] | [**0.9**] | [**0.9**] |
| | $\tau$ | [5, **10**, 20, 40] | [5, **10**, 20, 40] | [5, **10**, 20, 40] | [5, 10, **20**, 40] |
| **FEDHBM(ours)** | $\eta$ | [**1**, 0.5, 0.1] | [**1**, 0.1] | [**1**, 0.5, 0.1] | [**1**, 0.5, 0.1] |
| | $\eta_l$ | [0.1, 0.05, **0.01**] | [0.1, **0.01**] | [**1**, 0.5, 0.1] | [1, 0.5, **0.3**, 0.1] |
| | $\beta$ | [**1**, 0.99, 0.9] | [**1**, 0.99, 0.9] | [**1**, 0.99, 0.9] | [**1**, 0.99, 0.9] |

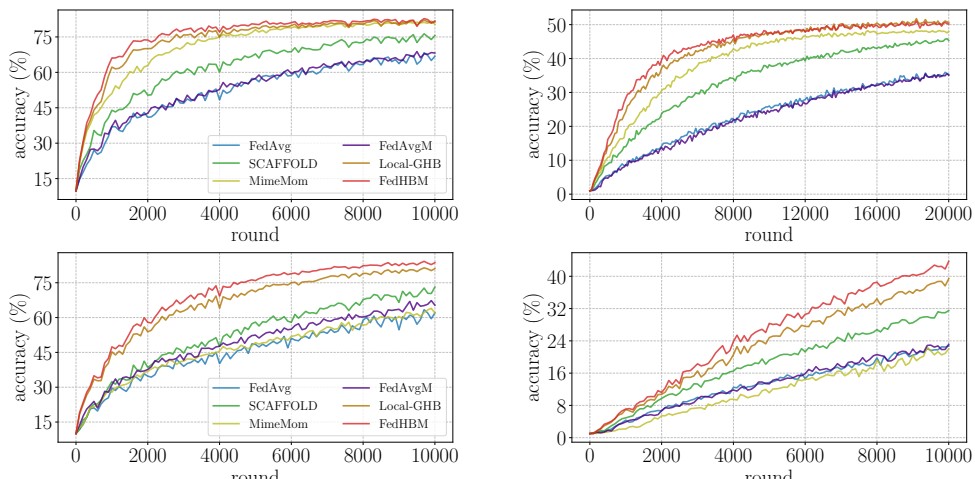

Figure 5: Accuracy plot of the best performing algorithms on CIFAR-10 (left, Tab. 8) and CIFAR-100 (right, Tab. 2) on CNN (top) and RESNET-20 (bottom), on our most heterogeneous setting ($\alpha = 0$).

MIMELITEMOM only with CNN improve FEDAVG. Conversely, LOCALGHBM and FEDHBM both consistently improve the state-of-art by a large margin.

Figure 6: Ablation study on the effect of several degrees of heterogeneity on performance of SOTA algorithms and FEDHBM on CIFAR-10 and CNN. The left figure shows the final accuracy reached by algorithms, while the right figure shows the number of rounds needed to reach 70% of absolute accuracy. The tables show the values depicted in the respective picture above. The best results are in **bold**, second best are in underlined.

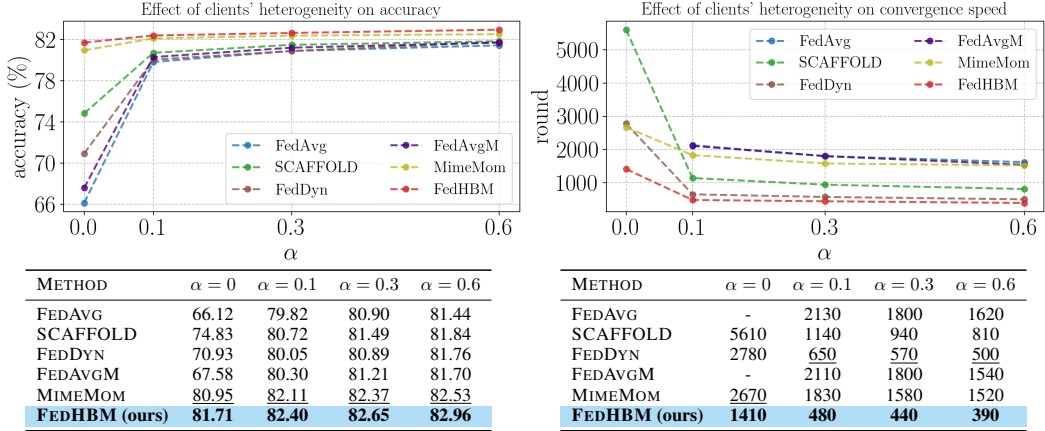

| METHOD | $\alpha = 0$ | $\alpha = 0.1$ | $\alpha = 0.3$ | $\alpha = 0.6$ | METHOD | $\alpha = 0$ | $\alpha = 0.1$ | $\alpha = 0.3$ | $\alpha = 0.6$ |
|---|---|---|---|---|---|---|---|---|---|
| FEDAVG | 66.12 | 79.82 | 80.90 | 81.44 | FEDAVG | - | 2130 | 1800 | 1620 |
| SCAFFOLD | 74.83 | 80.72 | 81.49 | 81.84 | SCAFFOLD | 5610 | 1140 | 940 | 810 |
| FEDDYN | 70.93 | 80.05 | 80.89 | 81.76 | FEDDYN | 2780 | 650 | 570 | 500 |
| FEDAVGM | 67.58 | 80.30 | 81.21 | 81.70 | FEDAVGM | - | 2110 | 1800 | 1540 |
| MIMEMOM | 80.95 | 82.11 | 82.37 | 82.53 | MIMEMOM | 2670 | 1830 | 1580 | 1520 |
| **FEDHBM (ours)** | **81.71** | **82.40** | **82.65** | **82.96** | **FEDHBM (ours)** | **1410** | **480** | **440** | **390** |

**Effect of different levels of heterogeneity** Figure 6 presents an analysis of the effect of heterogeneity on (i) final model quality (left) and (ii) convergence speed (right). The experimental results, while confirming that it is crucial to perform some form of drift control during local optimization, show that momentum methods handle extreme heterogeneity scenarios better than methods that rely on stochastic variance reduction, such as SCAFFOLD. Let us notice that the considered algorithms are robust w.r.t. non-extreme heterogeneity: this underlines the need for algorithms that do not sacrifice communication efficiency for robustness to heterogeneity. The right part of the figure shows that heterogeneity has a strong effect also on convergence speed. In line with the results on the left graph, MIMEMOM and FEDHBM are the fastest when facing the pathological case of $\alpha = 0$. Surprisingly, MIMEMOM is not significantly faster than FEDAVG and FEDAVGM in non-extremely heterogeneous scenarios; indeed it is slower if taking into account the communication overhead. In all cases FEDHBM performs best, demonstrating high robustness to heterogeneity from both the considered perspectives.

## C.7 ABOUT THE USE OF LEARNING RATE SCHEDULERS

For simplicity, in all our FL experiments we did not use any learning rate scheduler. In fact, while using strategies to change the learning rate as training proceeds is in general beneficial, this would result in a difficult tuning of hyper-parameters associated with the scheduler, since the algorithms present very different convergence rates.

Let us also point out that many well-established works in FL do not use learning rate schedules (McMahan et al., 2017; Li et al., 2020; Hsu et al., 2019; Karimireddy et al., 2020; 2021), while some others do (Acar et al., 2021). Figure 7 shows the accuracy curves of the best FL algorithms from Tab. 2, using a learning rate decay with de-

Table 8: Test accuracy (%) comparison of SOTA FL algorithms in a controlled setting. Best result is in **bold**, second best is underlined.

| METHOD | CIFAR-10 (RESNET-20) | | CIFAR-10 (CNN) | |
|---|---|---|---|---|
| | NON-IID | IID | NON-IID | IID |
| FEDAVG | $61.0_{\pm1.0}$ | $86.4_{\pm0.2}$ | $66.1_{\pm0.3}$ | $83.1_{\pm0.3}$ |
| FEDPROX | $61.0_{\pm1.8}$ | $86.7_{\pm0.2}$ | $66.1_{\pm0.3}$ | $83.1_{\pm0.3}$ |
| SCAFFOLD | $71.8_{\pm1.7}$ | $86.8_{\pm0.3}$ | $74.8_{\pm0.2}$ | $82.9_{\pm0.2}$ |
| FEDDYN | $60.2_{\pm3.0}$ | $87.0_{\pm0.3}$ | $70.9_{\pm0.2}$ | $83.5_{\pm0.1}$ |
| ADABEST | $73.6_{\pm3.0}$ | $86.7_{\pm0.5}$ | $66.1_{\pm0.3}$ | $83.1_{\pm0.4}$ |
| MIME | $53.7_{\pm2.9}$ | $86.7_{\pm0.1}$ | $75.1_{\pm0.5}$ | $83.1_{\pm0.2}$ |
| FEDAVGM | $66.0_{\pm2.2}$ | $87.7_{\pm0.3}$ | $67.6_{\pm0.3}$ | $83.6_{\pm0.3}$ |
| FEDCM (GHBM $\tau=1$) | $65.2_{\pm3.2}$ | $87.1_{\pm0.3}$ | $69.0_{\pm0.3}$ | $83.4_{\pm0.3}$ |
| FEDADC (GHBM $\tau=1$) | $65.7_{\pm3.0}$ | $87.1_{\pm0.2}$ | $66.1_{\pm0.3}$ | $83.4_{\pm0.3}$ |
| MIMEMOM | $69.2_{\pm3.6}$ | $88.0_{\pm0.1}$ | $80.9_{\pm0.4}$ | $83.1_{\pm0.2}$ |
| MIMELITEMOM | $57.0_{\pm0.9}$ | $88.0_{\pm0.4}$ | $78.8_{\pm0.4}$ | $83.2_{\pm0.3}$ |
| **LOCALGHBM (ours)** | $80.6_{\pm0.3}$ | $88.8_{\pm0.1}$ | $81.1_{\pm0.3}$ | $83.7_{\pm0.1}$ |
| **FEDHBM (ours)** | **$83.4_{\pm0.3}$** | **$89.2_{\pm0.1}$** | **$81.7_{\pm0.1}$** | **$83.8_{\pm0.1}$** |

cay coefficient fine-tuned for each algorithm, searched in the range {0.999, 0.9992, 0.9995, 0.9999}. For all the algorithms, the best learning rate decay turned out to be 0.9999. Comparing with performances without learning rate decay, it is possible to notice that: (i) the use of learning rate decay, in general, does not change the relative performance of the algorithms; (ii) in these settings, the use of

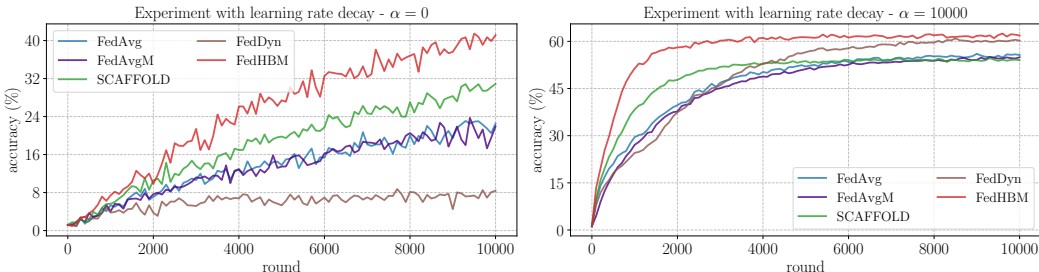

Figure 7: Experiments with learning rate decay of SOTA algorithms and FEDHBM on CIFAR-100 with RESNET-20. The decay coefficient has been searched in the range $\{0.999, 0.9992, 0.9995, 0.9999\}$ separately for each algorithm.

learning rate decay does not help convergence. This is particularly true in non-iid scenarios, where the performances are degraded w.r.t. not applying any schedule. This is motivated by the fact that a large number of rounds is needed to achieve convergence, and probably the simple decay strategy adopted from Acar et al. (2021) is not optimal to practically give an advantage. Other learning rate schedules may be more appropriate, but this largely expands the needed hyperparameter search, considering that it must be searched separately for each algorithm.

## C.8 GHBM AS SERVER-SIDE MOMENTUM

In this section, we show a comparison between GHBM applied at the client side, as proposed in this work, and its use as a server-side optimizer. In the latter case, clients simply run vanilla SGD locally, their gradients are then aggregated and the resulting server pseudo-gradient is used to update the momentum term and the global model. The momentum update rule itself remains unchanged, i.e. follows eq. (7), the only difference is that momentum is applied server-side, similarly to FEDAVGM (Hsu et al., 2019) and accordingly to the FEDOPT framework (Reddi et al., 2021). As shown in Fig. 8, server-side GHBM shows surprisingly good results, very close to client-side GHBM, but only with one of the architectures adopted in this study. This underscores that, if clients have not irremediably drifted during the local update, then server-side GHBM can be highly effective, even in the most heterogeneous settings. However, these findings are not consistent across architectures; indeed server-side GHBM does not show the same improvement when the experiment is carried out on a simpler CNN. In summary, these results confirm the findings in previous work about the necessity of integrating drift correction locally at clients (Karimireddy et al., 2020; 2021).

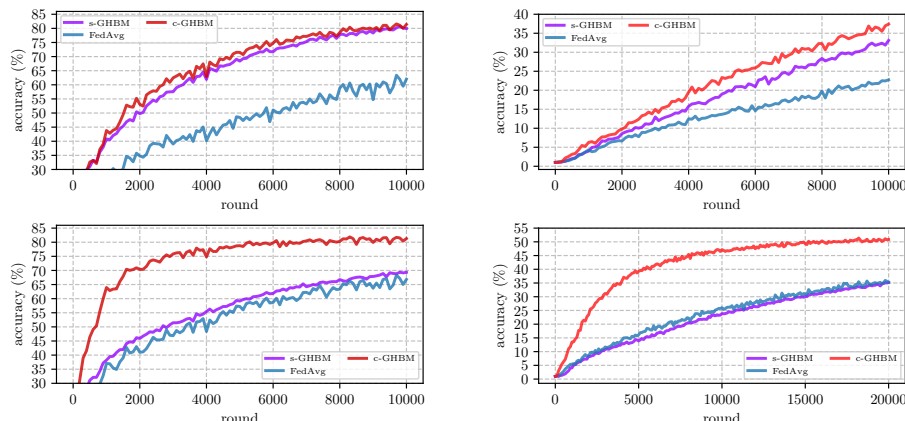

Figure 8: Comparison between server-side (s-*) and client-side (c-*) GHBM, on CIFAR-10 (left) and CIFAR-100 (right), with RESNET-20 (top) and a simpler CNN (bottom) on non-iid setting ($\alpha = 0$). Results show that client-side GHBM always performs better, confirming previous work on the necessity of integrating drift correction locally at clients (Karimireddy et al., 2020; 2021).

## C.9 THEORETICAL EXPERIMENTS ON THE RELATIONSHIP BETWEEN FEDCM AND GHBM

In this section, we conduct an experimental verification that FEDCM in partial participation, even when enforcing cyclic participation (see assumption 4.3), does not behave like GHBM, *i.e.* the algorithms are not equivalent.

We propose to analyze a theoretical setting where a federation of $K = 10$ clients with non-iid local datasets, collaboratively learns a quadratic function. In the experiment, $K \cdot C = 2$ clients are cyclically selected at each round. We compare, FEDCM (equivalent to GHBM with $\tau = 1$) and GHBM using the theoretical optimum $\tau = 1/C = 5$. As it is possible to notice from the training losses in Fig. 9, FEDCM in cyclic partial participation (in blue) does not converge as GHBM (in red). Indeed, if the two algorithms were equivalent, the two curves should be always overlapping.

This experiment also validates an important point of our theoretical analysis: GHBM with optimal $\tau$ approximates in (cyclic) partial participation the same momentum update rule that FEDCM applies in full participation. This is shown by comparing the lines of GHBM in partial participation (*i.e.* $C = 0.2$) and FEDCM in full participation (*i.e.* $C = 1$) (in orange). The red curve approaches the orange one, with a slight slowdown introduced by (i) the initial $\tau$ rounds in which GHBM is still building up the momentum buffer and (ii) the errors analyzed in lemma 4.6 and shown in Fig. 1. This finding also holds in deep learning experiments: in Tab. 9 we compare GHBM and FEDCM in

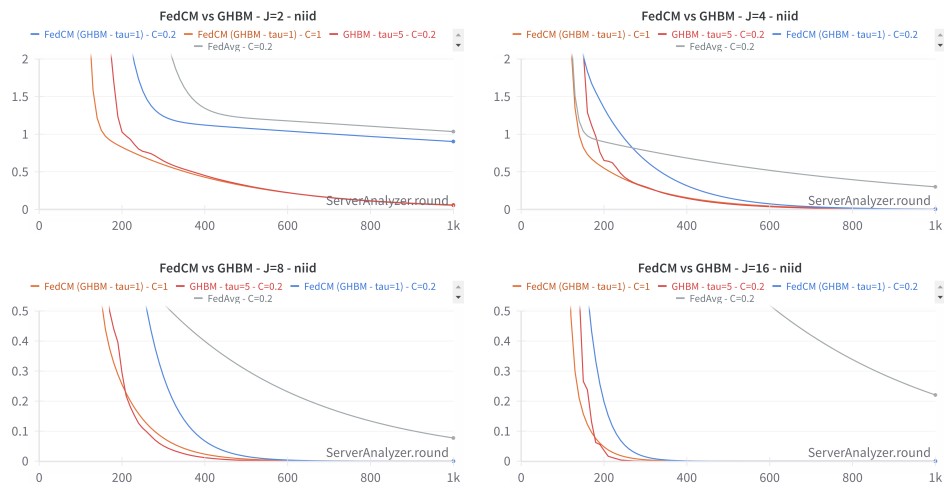

Figure 9: Comparison between FEDCM and GHBM (train loss) in cyclic partial participation on a theoretical, non-iid setting, across different numbers of local steps $J$. The plots show that: (i) FEDCM cannot achieve the results of GHBM (even enforcing cyclic participation) and (ii) GHBM in cyclic partial participation has nearly the same performance as FEDCM in full participation.

cyclic partial participation, on RESNET-20 with CIFAR-100, $\alpha = 0$ and $C = 0.1$. This experiment confirms that even when enforcing cyclic participation, the gap between the two algorithms remains the same. Furthermore, it shows that the performance of GHBM under random and uniform client sampling is similar to the ones in cyclic participation, corroborating the fact that, from an experimental perspective, strictly enforcing cyclic participation is not necessary for obtaining good performance.

| Method | Participation scenario | Final Test Accuracy (%) |
|--------|------------------------|-------------------------|
| FEDCM | random, uniform sampling | $22.2_{\pm 1.0}$ |
| GHBM | random, uniform sampling | $\mathbf{38.5}_{\pm 1.0}$ |
| FEDCM | cyclic participation | $22.5_{\pm 0.8}$ |
| GHBM | cyclic participation | $\mathbf{39.0}_{\pm 0.7}$ |

Table 9: Comparison between FEDCM and GHBM in cyclic and uniform partial participation. FEDCM is cannot attain the results of GHBM, even when enforcing cyclic participation.

## C.10 ADDITIONAL PLOTS OVER FEWER ROUNDS

Fig. 10 shows some results using fewer rounds than our experimental setup described in Sec. 5. We remind that the number of rounds $T = 10.000$ had been chosen because algorithms present very different convergence speeds depending on heterogeneity. Restricting a low number of rounds would cause evaluating the algorithms when they are too far from their convergence point in our most difficult setting ($\alpha = 0$). Indeed, as it is possible to notice from Fig. 10, none of the algorithms have reached its convergence point, so evaluating the algorithms under this restricted training budget can be misleading.

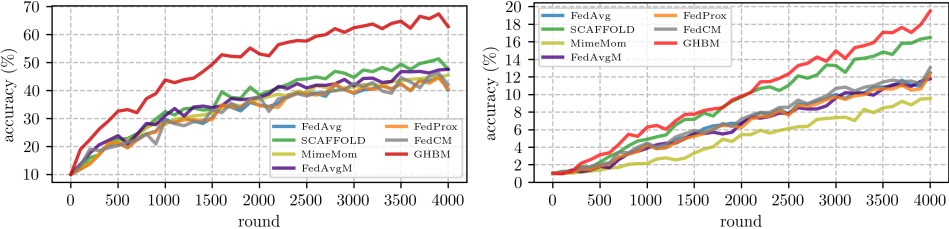

Figure 10: Accuracy plot of algorithms on CIFAR-10 (left, Tab. 8) and CIFAR-100 (right, Tab. 2) on RESNET-20, on our most heterogeneous setting ($\alpha = 0$).

