# OpenReview forum: "Communication-Efficient Heterogeneous Federated Learning with Generalized Heavy-Ball Momentum"
_ICLR.cc/2025/Conference — Submitted to ICLR 2025_

### Official Review · Reviewer_KZ3z · 2024-10-21

**Soundness:** 2
**Presentation:** 3
**Contribution:** 2
**Rating:** 6
**Confidence:** 4

**Summary:**

This paper proposes a multistep inertial momentum method in federated learning to alleviate the biases of the partial participation. It provides the theoretical improvements that the proposed method can achieve the full-participation rate under partial-participation. Experiments also shows its efficiency on different setups.

**Strengths:**

1. The writing is well-done and easy to understand the motivation and design in this paper.

2. The motivation of reducing the gaps caused by partial participation is also a good perspective, which is also a essential academic issue in FL.

**Weaknesses:**

1. The proposed approach is incremental. The use of momentum averaged over $\tau$ rounds, rather than a single round, was introduced based on experimental intuition. Although a correction scheme was applied both locally and globally, this appears to be a simple extension of the FedCM and FedADC approach [1].

[1] Ozfatura, Emre, Kerem Ozfatura, and Deniz Gündüz. "Fedadc: Accelerated federated learning with drift control." 2021 IEEE International Symposium on Information Theory (ISIT). IEEE, 2021.

2. The assumption 4.3 is too strong. Periodic participation is feasible, but the disconnect between the experiments and the theory prevents these two parts from supporting each other. Periodic participation implies that the sampling distribution is time-varying and uneven, which simplifies the impact of heterogeneity. The difference between the two is analogous to the theoretical analysis of SGD versus incremental GD. In fact, incremental GD has been proven under certain conditions to surpass the convergence speed of SGD. I believe this is one of the main weaknesses of the paper. The authors mention that under optimal participation rate design, the heterogeneity assumption is unnecessary. In fact, this is one of the theories behind incremental GD eliminating the stochastic biases. This proof does not accurately demonstrate that such properties can be maintained under random selection.

3. Some of the assumptions in the proofs seem to differ. Lemma 4.4 and Corollary 4.5 adopts the heterogeneity assumption but the authors do not claim it clearly. See question (3).

4. Some of the conclusions seem to lack precision. Line 381 comments "since increasing $\tau$ does not increase the overall error due to this term." Does the term $\frac{1}{\tau}\sum_{k=t-\tau+1}^{t}U_{k}$ is proven a constant order? If I missed this proof, please let me know. Currently, there does not seem to be sufficient evidence to indicate the specific nature of this term.

5. The comparisons in Table 1 lack important information. Although Table 1 shows that the proposed method can achieve global participation convergence speed under partial participation, this seems to strictly rely on Assumption 3 and the cyclical setting. I believe it is necessary to clearly state the conditions of this conclusion, rather than allowing readers to mistakenly think that it has surpassed the upper bound of partial participation as it currently stands.  In fact, after reading the entire paper, I believe that with the help of Assumption 3, the theoretical conclusions of FedCM could also reach the same conclusions presented in this paper. Based on the current results, I do not think this theoretical advancement is due to improvements in the algorithm itself.

**Questions:**

1. What is the meaning of $\alpha$ in Figure.1? And, why is the variacne of the gaps larger on iid than non-iid? In principle, the raw gradients of identically distributed datasets should remain highly similar between local clients, resulting in smaller variance. The author could further elaborate on the experimental process and explain why this conclusion leads to counterintuitive results.

2. I have a question about the form of Equation 3. Why is this considered the desired momentum update? Especially for the term $\widetilde{g}^{k}(\theta^{t-1})$, if the state $\theta^{t-1}$ is fixed, i.e. it is irrelated to $k$, then it seems to be only related to $S^k$. Do authors mean by that under a large $\tau$, the union of $S^k\rightarrow S$? I believe the explanation here is overly simplistic, making the approach appear unreasonable.

3. Eq (8) and (9) adopt the bounded heterogeneity assumption, since there is $G$ terms to show the deminished errors. However, the absence of the $G$ term's influence in the subsequent conclusions. Additionally, are the inferences from Equations (8) and (9) still valid for the later conclusions?

4. Can authors provide the general result of Theorem 4.7? For instance, under any selection on $\tau$ without fixing it as $1/C$. Also, how to recover the $\tau =1$ case? And how does the selection $\tau$ impact the final results?

5. I notice the general rounds are set as $10,000$. What is number of local steps $J$? This seems to be a very exaggerated setup, as the referenced baseline algorithms typically do not exceed 1,000 training rounds on the same task. Though FedCM provides results after 4,000 rounds. Additionally, why Table 3 does not contain the results of FedCM and FedADC, and other methods in Table 2?

6. The selection of $\beta$ seems to be related to $T$ and $J$. So, this term is a parameter that needs to be precisely designed in this algorithm rather than being a constant? Does it imply that different total training lengths $T$ will affect the choice of $\beta$?

Some typos:
(1) Line 142, $\theta^{t,J_i}$ should be corrected as $\theta_i^{t,J_i}$.

---------------------------------- After rebuttal -----------------------------------------------

I hope the author can include a discussion section in the main text to draw an analogy between the discussion of IGD comparisons and heterogeneity in FL. This will help enhance the paper's impact, as it truly highlights the core source of this technique. Secondly, regarding the author's mention of research on lower bounds, I encourage them to pursue related studies. Research on lower bounds has always been very challenging yet valuable, but it does not need to be included as a contribution in this paper. Finally, I suggest further revising the narrative in the paper to explicitly state the conditions and assumptions for any claimed theoretical advancements to prevent readers from being misled. Regarding the author's statement that other methods cannot achieve certain results due to the lack of guarantees in existing proofs, I understand this writing style—it is sometimes valid. However, in the context of this paper, it is clearly debatable. After all, new assumptions can also be applied to many previous algorithms to derive entirely new conclusions. I have raised my score to 6.

---

> ### Author Response · Authors · 2024-11-23
> **Response 1/3**
>
> We thank the reviewer for the time spent reviewing our manuscript. In the following, we respond pointwise to the reported weaknesses (W) and questions (Q).
>
> **W1: On the novelty and relationship with prior momentum-based approaches**
> The idea of averaging the momentum over $\tau$ rounds is built from a theoretical and simple intuition on why classical momentum in FL may not work well under high heterogeneity and partial participation. In particular, the update rule in eq. (4) is derived from the modification that is needed in theory to extend the guarantees on heterogeneity proved by [1] from full to partial participation (eq. (3)).
> In practice, even if the algorithmic modification is simple, this stems from fundamental theoretical considerations, and then it is shown to be exactly what is needed to achieve in partial participation a similar update of the momentum term as FedCM in full participation (see Figure 1), unlocking superior performance both in theory and in simulations.
> **The above factors account for the significant contributions we bring with this work, which indeed is not a mere extension of FedCM and FedADC**.
>
> **W5-W2: Why cyclic participation is a legitimate assumption for GHBM**
> Thanks for pointing out this concern, we will add a discussion on why the cyclic participation assumption is needed for our conclusions, and clarify that our result is non-trivial and follows from the algorithmic design of GHBM rather than from the additional assumption itself.
>
> __The FedCM cannot reach the same conclusion as GHBM__
> To this end, the first point we want to highlight is that, while it is needed for simplifying our analysis, **cyclic participation is not sufficient by itself to prove the same result for FedCM.**
>
> To clearly see this fact, we propose a theoretical experiment where a federation of $K=10$ clients, $K\cdot C=2$ cyclically selected at each round, collaboratively learns a quadratic function, with non-iid local datasets. If the two algorithms are equivalent under the assumption of cyclic participation, they should converge in the same way We report the experiment results in Fig. 9 of our revised manuscript (section C.9), showing that **FedCM in cyclic partial participation does not have the same behavior as GHBM**. Indeed, the plots also show that **GHBM with optimal $\tau$ approximates in (cyclic) partial participation the same momentum update rule of FedCM in the full participation setting**, confirming the validity of our analysis in section 4.2.
>
> To further dispel any doubts, we additionally show that this finding also holds in deep learning experiments: in the table below we compare GHBM and FedCM in cyclic partial participation, on ResNet20 with CIFAR-100, $\alpha=0$ and $C=0.1$.
> It shows that **the performance of GHBM under random and uniform client sampling is similar to the ones in cyclic participation** and that, even enforcing cyclic participation, **the gap between GHBM and FedCM does not decrease**.
>
> This confirms **that the advantage of GHBM does not come as a consequence of the additional cyclic participation assumption but from the algorithm itself.**
>
> | Method | Participation scenario   | Final Test Accuracy (%) |
> |--------|--------------------------|-------------------------|
> | FedCM  | random, uniform sampling | $22.2_{\pm 1.0}$    	|
> | GHBM   | random, uniform sampling | $38.5_{\pm 1.0}$    	|
> | FedCM  | cyclic participation 	| $22.5_{\pm 0.8}$    	|
> | GHBM   | cyclic participation 	| $39.0_{\pm 0.7}$    	|
>
> __The advantages of cyclic participation in literature__
> The best-known analysis of FedAvg under cyclic participation is provided by [2], while it proves that in certain situations (e.g. clients run GD instead of SGD) there can be an asymptotic advantage in the case we prospect with assumption 4.3, **their results still rely on forms of bounded heterogeneity, and with this respect the results presented in this work are novel and advance the state of the art**.
>
> Based on the above premises, cyclic participation is clearly a legitimate assumption for GHBM and does not affect the validity of the results we present.
>
> **W3-Q3: Clarifications about lemma 4.4 and corollary 4.5**
> Lemma 4.5 and corollary 4.6 show how the error induced by heterogeneity decreases as $\tau$ increases, and for general $\tau$ it does use the bounded heterogeneity assumption. However, for the special case of $\tau=\frac{1}{C}$ the assumption is not needed, since the term in the LHS of the equations is zero by definition. This is already specified in lines 356-358 of the main paper and as the first step in the proof of lemma 4.5, lines 1024-1025.
>
> > __Additionally, are the inferences from Equations (8) and (9) still valid for the later conclusions?__
>
> Consequently, the inferences of equations (8)-(9) are valid for the rest of the proofs, as theorem 4.7 explicitly imposes $\tau=\frac{1}{C}$.

---

> ### Author Response · Authors · 2024-11-23
> **Response 2/3**
>
> **W4: On the (b) term of lemma 4.6**
> The term $\mathcal{U}_k \forall k \in [T]$ is constant w.r.t. $\tau$. To see this fact it is necessary to combine lemma B.6 with lemmas B.7-B.5 and apply the upper bound on $\eta_l$ in eq. (122). Then, since the (b) term in lemma 4.6 is an average of the last $\tau$ of $\mathcal{U}_k$ terms, increasing $\tau$ does not linearly increase this term.
> We realize that inferring such dependence is not trivial, even for the attentive reader. We will be more detailed in the explanation of the lemma.
>
> **Q1: Analysis of gradients over heterogeneous scenarios**
> This reviewer’s concern is caused by a typo in the caption of figure 1, which should be “... in **non-iid** ($\alpha=0$, left) and **iid** ($\alpha=10k$, right) …”, i.e. $\alpha=0$ corresponds to non-iid and $\alpha=10k$ to iid.  $\alpha$ is the parameter of the Dirichlet distribution controlling the heterogeneity in splitting the dataset among clients, the procedure is the standard one in FL works and it is detailed in section C.2. Low values of $\alpha$ indicate more heterogeneous splits, while higher values indicate lower heterogeneity. The reviewer’s intuition is correct - there is higher variance in the non-iid scenario than iid - and indeed it does align with the presented results. Apologies for the confusion, we fixed the typo.
>
> **Q2: Explanation of why eq. (3) is the desired momentum update rule**
>
> The reason why eq. (3) is the ideal momentum update rule is indeed linked to parallelism the reviewer presented in W2. Essentially, the more clients we use to calculate a single gradient over the whole local dataset, the more the resulting gradient is the true gradient (by linearity of differentiation), i.e the momentum is updated as if we do GD instead of SGD. In heterogeneous FL this has a big impact, because the gradient over a small subset of clients (e.g. the ones selected at the current round) can be very different from the true gradient, leading to a biased update of the momentum.
> By using cyclic participation, it is possible to exactly determine how many clients we considered in averaging the gradients of the last $\tau$ rounds, hence the optimal $\tau$.
>
> This idea, while it is simple and builds on fundamentals, is exactly what is missing from FedCM to offer strong guarantees against heterogeneity even in partial participation. Indeed, a careful analysis of the best proof of FedCM presented in [1] reveals that, in order to reach convergence under arbitrary heterogeneity, the momentum term must be updated with the average gradient over all the clients. FedCM can achieve this only by enforcing full participation, as in other cases there will always be a residual heterogeneity.
> GHBM instead changes the design of the momentum rule, and attains similar results by using past gradients (i.e. uses eq. (4) instead of (3)).
>
> **Q4: General form of theorem 4.7**
> When $\tau$ is not properly set the algorithm will still depend on bounded heterogeneity, since an additional term on the order of $\mathcal{O}(\frac{G^2}{\sqrt{T}})$ appears in the rhs of the convergence rate. We will add this result to the revision later, due to the short time of the rebuttal process.
> However, convergence guarantees should be stated for the optimal choice of parameters, and for this reason our Theorem 4.7 is stated for $\tau=\frac{1}{C}$.

---

> ### Author Response · Authors · 2024-11-23
> **Response 3/3**
>
> **Q5: On the experimental setting**
> _Local steps_
> Regarding local steps, for the controlled scenarios we fix to one local epoch, that equals $J=8$ for CIFAR-10/100, $J=20$ for Shakespeare and $J=27$ for StackOverflow. Similarly, for large scale vision datasets, following [1] we run the algorithms for five local epochs (all these details are reported in Table 6)
>
> __Number of rounds__
> We added a new section C.10 with plots over fewer rounds ($T=4000$) in our revision. However, we point out that we reported the results over $T=10.000$ rounds because algorithms present very different convergence speed depending on heterogeneity. Restricting a low number of rounds, e.g. $1000$, would cause evaluating the algorithms when they are too far from their convergence point in our most difficult setting ($\alpha=0$).
>
> Other papers only evaluate under milder heterogeneity (e.g. $\alpha \in \{0.1, 0.3, 0.6\}$) and the difference with $\alpha=0$ in convergence speed of FL algorithms is quite substantial (e.g. see our Figure 6-right).
>
> Let us also note that evaluating over more rounds is an advantage for competitor algorithms: since our algorithm is by far the fastest (see Table 4), our practice favors them to reach their best performance (of course there always is a maximum time/resource budget for the experiments).
>
> __Methods in Table 3__
> In Table 3 we extend the experimentation only for the best performing methods, that is we excluded methods that present failure cases or did not improve FedAvg on the controlled scenario. In particular, with reference to settings in Table 3, FedDyn and AdaBest did not converge in any of them, FedProx has lower performance than FedAvg and Mime/MimeLiteMom perform worse than the reported MimeMom (which is expected to be better than both).
> FedCM instead has been implicitly considered in the grid search for GHBM, since it is equivalent to GHBM with $\tau=1$. For transparency, we report below the results of GHBM-$\tau=1$ compared to the best tuned GHBM:
>
> | Method            	| MobileNetV2  	| ViT-B\16     	|
> |-----------------------|------------------|------------------|
> | FedCM {\small (GHBM-$\tau=1$)}     	| $61.6_{\pm 0.3}$ | $70.1_{\pm 0.5}$ |
> | GHBM-$\tau=40$ (best) | $65.1_{\pm 0.1}$ | $74.3_{\pm 0.5}$ |
>
>
> **Q6: Selection of $\beta$**
> Good observation, indeed the optimal value for $\beta$ depends on $J, T$, similarly to the current best known proof for momentum-based FL algorithms [1], which ours is based on. Indeed, our bound on $\beta$ is very similar to theirs (see their eq. (5)).
>
> [1] Cheng et al., Momentum Benefits Non-IID Federated Learning Simply and Provably, ICLR 2024
>
> [2] Cho et al., On the Convergence of Federated Averaging with Cyclic Client Participation, ICML 2023

---

> > ### Comment · Reviewer_KZ3z · 2024-11-23
> >
> > Thanks for the author's rebuttal. After reading the reply, I believe there are yet some issues unresolved well.
> >
> > 1. Originality and novelty: I have reviewed some FL research based on internal momentum, and this work appears to be similar to the multi-step momentum design in [1]. Moreover, this design itself is merely a straightforward extension of FedCM, I believe the technical contribution of the method in this paper is not as "substantial" as the authors have claimed. Based on my current understanding, the purpose of introducing multi-step components in the theoretical contribution is to construct a stochastic error elimination term similar to the incremental gradient descent (GD) method. In the proof of incremental GD, when a complete epoch is finished—meaning the entire dataset has been traversed—the stochastics is perfectly eliminated. In the context of FL, the authors extend this idea to address stochastics introduced by partial participation.
> >
> > [1] Enhance local consistency in federated learning: A multi-step inertial momentum approach
> >
> > 2. In FL, the concept of partial client participation in experimental research is not solely aimed at reducing communication overhead. A critical reason lies in the fundamental assumption of FL: many edge devices are unreliable and may drop out during training. This unreliability makes it essential to account for node failures as part of the system design. The randomization introduced by partial participation is significant because it reflects the reality that we cannot predict which nodes will drop out. Thus, dropout events are treated as random occurrences. This approach not only ensures the robustness of the FL system under unpredictable conditions but also aligns with real-world scenarios where device availability fluctuates due to factors like connectivity issues or user behavior. Here, I must emphasize that while using periodic participation as a training method is feasible, but the current theoretical benefits rely on the assumption that all participants remain consistently connected without any dropouts, as the choice of $\tau$ is fixed by $C$. This imposes extremely high stability requirements on FL system, which are challenging to achieve in practice.
> >
> > 3. FedCM cannot take advantage of this theoretical advancement: [2] proves that local optimizer is selected as GD, SGD, or SSGD, they all achieves the rate as full participation case under the cyclic client participation. I am unclear about the theoretical basis for the author's claim that FedCM cannot share this property. Could the author provide a proof showing that under reasonable hyperparameter design, FedCM's convergence lower bound is greater than the full-participation scenario? My basis for this statement is that the convergence of FedCM currently aligns with that of FedAvg, meaning that the momentum-enhanced local optimization retains the same properties and rate as SGD. Since SGD can achieve this property through cyclic participation, why can't FedCM? Could authors provide more evidence to support this claim?
> >
> > [2] On the Convergence of Federated Averaging with Cyclic Client Participation
> >
> > The author's responses to other experimental questions are good, and I think there are no points worth discussing. Given the time limitation for responses, I will further discuss the aforementioned issues 1 and 2 above with other reviewers and the AC in the next phase. If the authors have any additional insights into the contributions of this paper, they can summarize and emphasize them again. I hope the authors can address the issue 3, which is the theoretical justification for why cyclic participation cannot help other methods, such as FedCM, achieve acceleration. This description seems to slightly conflict with the conclusions in [2].

---

> > > ### Author Response · Authors · 2024-12-02
> > > **R2: Cyclic training doesn’t need to be imposed on a FL system in practice to get good performance out of GHBM**
> > >
> > > **R2: Cyclic training doesn’t need to be imposed on a FL system in practice to get good performance out of GHBM**
> > >
> > > It's important to distinguish between the role of cyclic participation in our theoretical analysis and its relevance to practical applications. As explained in our response, while our theoretical convergence rates are derived under the assumption of cyclic participation, this is primarily a tool for analysis and does not limit the applicability of GHBM to more realistic settings.
> > >
> > > On the practical implications for real-world application, we share the reviewer’s concern about the limitations of this assumption. **This is precisely why our experiments are not conducted under cyclic participation** but in the more common setting of random uniform client sampling. In this sense, we do appreciate the reviewer’s attention to the implications of theoretical aspects in real-world FL systems, as **in our work we took particular care in considering realistic and large-scale settings** with a large number of clients (on the order of $10^4$-$10^5$) and very low participation (e.g. $C<1\\%$). This simulates the challenges of real-world FL systems more accurately than many existing studies.
> > >
> > > **Our results clearly demonstrate that GHBM significantly improves final accuracy (Table 3) and convergence speed (Table 4), even in these challenging settings**. This directly addresses the reviewer's concern about the applicability of GHBM in the real world.

---

> > > ### Author Response · Authors · 2024-12-02
> > > **R3-1: [2] does not claim that FedAvg under cyclic participation has the same rate as full participation**
> > >
> > > **R3-1: [2] does not claim that FedAvg under cyclic participation has the same rate as full participation**
> > >
> > > > _[2] proves that local optimizer is selected as GD, SGD, or SSGD, they all achieves the rate as full participation case under the cyclic client participation._
> > >
> > > We appreciate the inquisitiveness of the reviewer’s observations, but **[2] does not prove that under cyclic participation SGD achieves the same rate as in full participation**.
> > >
> > > In fact, there is no mention of such a claim throughout the entire paper. Indeed, plugging the constants corresponding to full and cyclic partial participation in their theorem 2, we can show below that the rates in (cyclic) partial and full participation are not the same.
> > >
> > > _Rate in theorem 2 of [2] in cyclic participation (our assumption 4.3)_
> > >
> > > Let us use the notation in [2], and then translate the rate with our notation for clarity. Notice that our cyclic participation corresponds to $\bar{K}=M/N$ where $M$ is the total number of clients and $N$ is the number of clients selected in any round, and that $T=K\bar{K}$. Theorem 2 states that:
> > >
> > > $\mathbb{E}[F(w^{(K,0)}] - F^* \leq \frac{F(w^{(0,0)}) -F^*}{MK} + \tilde{\mathcal{O}}\left(\frac{\kappa^2 (\bar{K} - 1)\alpha^2}{\mu K}\right)  + \tilde{\mathcal{O}}\left(\frac{\kappa \sigma^2}{\mu \tau NK}\right) + \tilde{\mathcal{O}}\left(\frac{\kappa^2 (\tau - 1) \nu^2}{\mu T M^2 K^2}\right)$
> > >
> > > Use the fact that $K=T/\bar{K}=T\frac{N}{M}$ to obtain that:
> > >
> > > $\mathbb{E}[F(w^{(K,0)}] - F^* \leq \frac{M(F(w^{(0,0)}) -F^*)}{(TN)^2} + \tilde{\mathcal{O}}\left(\frac{\kappa^2 \bar{K}(\bar{K} - 1)\alpha^2}{\mu T^2}\right)  + \tilde{\mathcal{O}}\left(\frac{\kappa M \sigma^2}{\mu \tau N^2 T}\right) + \tilde{\mathcal{O}}\left(\frac{\kappa^2 (\tau - 1) \nu^2}{\mu T^2 N^2}\right)$
> > >
> > > Translating to our notation, $M=|\mathcal{S}|$, $N=|\mathcal{S}|C$, $\tau=J$, we further have that the LHS is:
> > >
> > > $\leq \frac{\Delta}{T|\mathcal{S}|C^2} + \tilde{\mathcal{O}}\left(\frac{\frac{\kappa^2}{C} (\frac{1}{C} - 1)\alpha^2}{\mu T^2}\right)  + \tilde{\mathcal{O}}\left(\frac{\kappa \sigma^2}{\mu J T |\mathcal{S}|C^2}\right) + \tilde{\mathcal{O}}\left(\frac{\kappa^2 (J - 1) G^2}{\mu JT^2 {(|\mathcal{S}|C)}^2}\right)$
> > >
> > > where the symbols $\kappa=\frac{L}{\mu}$ and $\alpha$ have been maintained as in [2] because they do not have a corresponding symbol in our notation.
> > > **As the above rate depends on the participation ratio $C$, the rate for partial participation ($C<1$) is not the same as full participation($C=1$).**
> > >
> > >
> > > _What authors of [2] actually claim_
> > >
> > > The authors of [2] compare the rates of FedAvg between (cyclic) and random uniform participation **under the same participation ratio**, not between (cyclic) partial participation and full participation.
> > > In particular, they prove that, **given the same participation ratio**, under suitable conditions, cyclic participation can improve the convergence rate of FedAvg w.r.t. random uniform sampling from $\tilde{\mathcal{O}}(\frac{1}{T})$ to $\tilde{\mathcal{O}}(\frac{1}{T^2})$ (for PL objectives).
> > >
> > > > My basis for this statement is that the convergence of FedCM currently aligns with that of FedAvg, [...]. Since SGD can achieve this property through cyclic participation, why can't FedCM?
> > >
> > > As such, **this fact contradicts the basis for the reviewer’s statement about SGD achieving the same rate as in full participation through cyclic participation**, and consequently the implications for FedCM. Indeed, below, we formally prove that FedAvg lower bound does not improve under cyclic participation.

---

> > > ### Author Response · Authors · 2024-12-02
> > > **R3-2: [Proof] FedAvg lower bound does not improve under cyclic participation**
> > >
> > > **R3-2: [Proof] FedAvg lower bound does not improve under cyclic participation**
> > >
> > > We extend theorem II of [4] to (cyclic) partial participation, proving that a similar lower bound for the minimizer found by FedAvg also holds in this setting.
> > >
> > > More precisely, we assume FedAvg is run for $T=R\bar{R}$ rounds, where $R$ can be referred to as a \textit{cycle-epoch}, and $\bar{R}$ is the number of rounds needed to see all clients once. We further call their respective indexes $(r, \bar{r})$.
> > > %
> > > We assume clients run the algorithm with $J > 1$, and arbitrary possibly adaptive positive step-sizes $\{\eta_{l}^{(1)}, ..., \eta_l^{(R)}\}$ are used with $\eta_l^{(r)} \leq 1/\mu$ and fixed within a cycle-epoch for all clients. The server step size is fixed to $\eta=1$.
> > >
> > > At each round we sample $|\mathcal{S}^t| = |\mathcal{S}|C$, where $C$ is the participation ratio and hence $\bar{R}=\frac{1}{C}$.
> > >
> > > **Theorem B.9.** _For any positive constants $G, \mu$ there exist $\mu$-strongly convex functions satisfying assumption B.1 for which that the output of FedAvg satisfying the above conditions has the following error for any $t \ge 1$:_
> > > \begin{equation*}
> > > 	f(\theta^t) - f(\theta^*) \ge \Omega \left( \min \left(f(\theta^0) - f(\theta^*), \frac{\bar{R}^2 G^2}{\mu T^2} \right)\right)
> > > \end{equation*}
> > >
> > > _Proof._
> > > We assume each client is assigned one of the two following simple one-dimensional functions for any given $\mu$ and $G$:
> > >
> > > $f_1(\theta) := \mu\theta^2 + G\theta, \text{ and } f_2(\theta):=-G\theta$
> > >
> > > We assume that there is a set $\mathcal{S}$ of clients, with $|\mathcal{S}|$ even, and that $|\mathcal{S}|/2$ clients optimize $f_i(\theta):=f_1(\theta)$ and the rest $f_i(\theta):=f_2(\theta)$.
> > > It is clear that $f(\theta)=\frac{1}{|\S{}|}\sum_{i=1}^{|\S{}|}f_i(\theta)=\frac{\mu}{2}\theta^2$, which has global minimizer at $\theta=0$.
> > > We construct the client sampling process such that clients optimizing the same function are always sampled for $\frac{\bar{R}}{2}$ rounds in a cyclic order.
> > > Running FedAvg from $\theta^0 > 0$, running $J$ local steps the update after $\bar{R}/2$ rounds is:
> > >
> > > \begin{equation}
> > >     	\theta_1^{r\bar{R}} = \theta^{(r-1/2)\bar{R}}\left(1-2\mu \eta_l^{(r)} \right)^{J\bar{R}/2} - \eta_l^{(r)} G \sum_{j=0}^{J\bar{R}/2-1}\left( 1-2\mu\eta_l^{(r)}\right)^j
> > > 	\end{equation}
> > > 	when sampling clients optimizing $f_1$ and
> > > 	\begin{equation}
> > >     	\theta_2^{r\bar{R}} = \theta^{(r-1/2)\bar{R}} + \eta_l^{(r)}GJ\bar{R}/2
> > > 	\end{equation}
> > > 	when sampling clients optimizing $f_2$.
> > > 	Assume the training starts sampling clients optimizing $f_1$. It follows from the above-described cyclic participation pattern that:
> > >
> > > \begin{align}
> > >     	\theta^{r\bar{R}} &= \theta^{(r-1)\bar{R}}\left(1-2\mu \eta_l^{(r)} \right)^{J\bar{R}/2} + \eta_l^{(r)} G \sum_{j=0}^{J\bar{R}/2-1}\left(1-(1-2\mu\eta_l^{(r)})^j\right) \\
> > > & \ge \theta^{(r-1)\bar{R}}\left(1-2\mu \eta_l^{(r)} \right)^{J\bar{R}/2} + \frac{\eta_l^{(r)} G}{2} \sum_{j=0}^{J\bar{R}/2-1}\left(1-(1-2\mu\eta_l^{(r)})^j\right)
> > > 	\end{align}
> > >
> > > We can apply lemma 9 of [4] for $R=\frac{T}{\bar{R}}$ rounds, which proves that, for any $r>1$:
> > >
> > > $\begin{equation}
> > >     	\theta^r \ge c \min(\theta^0, \frac{G}{\mu R})
> > > 	\end{equation}$
> > >
> > > Finally, the result follows from noting that $f(\theta^r) = \frac{\mu}{2}(\theta^r)^2$.

---

> > > ### Author Response · Authors · 2024-12-02
> > > **R3-3: On the reviewer’s claim that FedCM with cyclic participation could converge at the full participation rate under unbounded heterogeneity**
> > >
> > > **R3-3: On the reviewer’s claim that FedCM with cyclic participation could converge at the full participation rate under unbounded heterogeneity**
> > >
> > > It's important to note that the literature doesn't support the reviewer's claim that "FedCM with cyclic participation converges at the full participation rate under unbounded heterogeneity", and it shouldn't be expected. Indeed, the classical momentum used in FedCM is inevitably biased towards the last update, as heterogeneous FL with partial participation is biased towards the most recently selected clients. The fact that the state-of-the-art study on momentum in FL [1] couldn’t extend the same guarantees of FedCM from full to partial participation is a strong indication that expecting it is not sound.
> > >
> > > In contrast, GHBM is specifically designed to address this issue. Its momentum term incorporates the average of the last $\tau$ rounds, ensuring that all clients' gradients are included when $\tau$ is appropriately chosen. This leads to the desired convergence property, recovering the full participation rate, which is both theoretically proven and extensively validated in our experiments.
> > >
> > > We believe we have addressed all the reviewer's concerns and demonstrated the value of our contributions. If any questions should remain, we are available for further clarification.

---

> > > ### Author Response · Authors · 2024-12-02
> > > **R1: On originality, novelty and significance**
> > >
> > > **R1: On originality, novelty and significance**
> > > > Moreover, this design itself is merely a straightforward extension of FedCM. [...] If the authors have any additional insights into the contributions of this paper, they can summarize and emphasize them again
> > >
> > > We would like to highlight the precise motivations that support our claim that the contributions of this work are both _significant_ and _novel_.
> > >
> > > **Algorithmic design.**
> > > The design of our GHBM is built from careful consideration of the theoretical challenges posed by heterogeneity in partial participation, as **we demonstrate both in theory and in practice why the momentum update rule should be modified as we propose for obtaining our strong theoretical result** (i.e. why we should choose $\tau=\frac{1}{C}$ and not just $\tau=1$ or any other fixed constant). As long as a proposed method is grounded and proven to be effective, simplicity is a further advantage.
> > >
> > > With respect to algorithmic design novelty, in the additionally cited arXiv work [3] there is no mention of the motivation and resulting design choices behind GHBM we clearly explain in section 3.3, nor the result of completely removing the dependence on gradient dissimilarity in (cyclic) partial partial participation by properly modifying the classical momentum update rule. In fact, their convergence rate does not show any improvement regarding heterogeneity. As the cited arxiv work does not present such key contributions, it cannot be regarded as “similar” to ours.
> > >
> > > **Theoretical results.**
> > > In our work we prove that the proposed GHBM converges under arbitrary heterogeneity in (cyclic) partial participation, advancing over [1], which proves this result for FedCM in full participation only. **Removing the restriction of full participation, which is made possible by the algorithmic structure of GHBM, is indeed a significant result**, as the reviewer discusses the practical importance of partial participation for realistic deployment.
> > >
> > > **Practical implications.**
> > > In our work, we directly demonstrate the effectiveness of GHBM in large-scale real-world scenarios, by conducting extensive experimentation on challenging settings characterized by a large number of clients (on the order of $10^4$-$10^5$) and very low participation (e.g. $C<1\\%$). As the reviewer mentioned that our algorithm is simple, we want to emphasize that **the simplicity of our algorithm is a significant advantage**. Indeed, a simple, principled approach with strong theoretical guarantees and solid empirical performance, that can efficiently scale to realistic settings, is undeniably preferable to more complex alternatives.
> > >
> > > **Motivated by the above points, we believe the originality, novelty, and significance of our work are well supported.** We are confident that our work addresses a significant gap in the existing literature and offers practical benefits for FL applications.  We hope this explanation resolves the reviewer’s concerns and are happy to provide further clarifications if needed.
> > >
> > > [3] Enhance local consistency in federated learning: A multi-step inertial momentum approach, arxiv
> > >
> > > [4] Karimireddy et al., SCAFFOLD: Stochastic Controlled Averaging for Federated Learning, ICML 2019

---

> > > > ### Comment · Reviewer_KZ3z · 2024-12-02
> > > > **Response for the rate questions**
> > > >
> > > > The authors' response of [2] that it shows "PL+CyCP" can improve the rate from $1/T$ to $1/T^2$. While this is a correct statement, it does not fully capture the true capability of the cyclic assumption. Let us revisit their conclusion: with a precise selection of the cyclic frequency $\overline{K}$, cyclic participation can eliminate the impact of the $\frac{M/\overline{K}-N}{M/\overline{K}-1}$ term. The core point here is not that this term is $1/T$, but rather that it is $1/N$. Cyclic participation essentially eliminates the influence of the $1/N$ term.
> > > >
> > > > Additionally, I have consistently encouraged the author to compare the methods of Incremental GD and SGD and to gain new insights through this comparison. This is because Incremental GD does not strictly require a complete pass through the dataset in one round, yet it can still achieve the elimination of randomness. Currently, authors claim only $\tau=1/C$ it can achieve acceleration. I understand this is because the author followed the proof in [2] to arrive at this conclusion. However, this conclusion is incomplete. The current conclusion expects that during **1 cyclic traversal**, after completing one pass through all clients, the local correction term $\Delta$ similarly contains the historical information of all clients. However, this property can be **extended periodically**. When the author chooses to traverse all clients over two communication rounds, a similar conclusion can also be drawn. In the proof, the author should not only consider the smoothness inequality for a single step but rather for aligning them on multiple steps. Then the selection of $\overline{K}$ can also be extended larger. This is precisely the conclusion already proven by Incremental GD vs SGD. We do not need to confine our analysis to a single epoch; instead, it can be extended to $c$ epochs. As long as $c$ satisfies certain conditions, it is always possible to find a suitable interval to eliminate randomness.
> > > >
> > > > The conclusion of this paper regarding the extension within a single cycle is correct, but the evaluation that methods like FedCM cannot benefit remains uncertain. I believe that, analogous to the conclusions of Incremental GD, when the proof is extended to multiple communication rounds, similar conclusions may emerge. Authors' response citing the lower bound for strongly convex functions is irrelevant. If authors can demonstrate that under the CyCP assumption, FedCM achieves a lower bound larger than the rate of full participation, I would agree with the overall conclusion that FedCM cannot benefit from this assumption. However, the current conclusion does not support this claim. I understand that the author aims to use the current proof to explain the behavior of all algorithms. Within a single traversal cycle, FedCM cannot exhibit this property. However, this does not imply that it remains absent when observed over a longer interval. The author's proof fails to provide accurate conclusions for observations over extended intervals, leaving the theoretical explanation of this property uncertain. I would like to remind the author that demonstrating FedCM's inability to achieve this property is not based on its upper bound proof. The author cannot use the upper bound in the current conclusions for comparison, as the tightness of this upper bound has not been verified.
> > > >
> > > > Currently, I believe the authors should remove the conclusion that methods like FedCM cannot achieve certain benefits, as the paper does not provide accurate theoretical proofs to support the correctness of this claim. It would be sufficient for the authors to focus on emphasizing what the proposed GHBM method can accomplish, rather than making over-claims about what other methods cannot achieve. I understand that this is a writing approach, but stating conclusions without strict proof is not appropriate. As I mentioned earlier, there is a requirement for a solid theoretical basis for this conclusion, specifically regarding the properties of the lower bound under FedCM. After the revision of these over-stated conclusions that may be incorrect, I will increase my score.

---

> > > > > ### Author Response · Authors · 2024-12-02
> > > > > **Response 2/2**
> > > > >
> > > > > **On the needed value of $\tau$**
> > > > > > Currently, authors claim only $\tau=\frac{1}{C}$ can achieve acceleration. The current conclusion expects that during 1 cyclic traversal, after completing one pass through all clients, the local correction term $\Delta$ similarly contains the historical information of all clients. However, this property can be extended periodically. When the author chooses to traverse all clients over two communication rounds, a similar conclusion can also be drawn.
> > > > >
> > > > > Thanks for pointing it out, this is indeed correct, under cyclic participation any $\tau=\frac{k}{C}$ with $k \ge 1$ will lead to similar conclusion, even though considering larger interval only increases the error due to using old gradients, so in practice we would like to choose $\tau$ as the minimum that allows convergence under unbounded heterogeneity. Since the comparison is drawn to classical momentum, our focus was on showing that it should be chosen $\tau>1$, implicitly suggesting that it should not be unnecessarily larger. **We will be more precise in specifying this when discussing the role of $\tau$ in section 4.2**

---

> ### Author Response · Authors · 2024-12-02
> **Response 1/2**
>
> We thank the reviewer for the high responsiveness, we would like to address the remaining concerns.
>
> **On the claim that FedCM cannot achieve the benefits**
> > Currently, I believe the authors should remove the conclusion that methods like FedCM cannot achieve certain benefits, as the paper does not provide accurate theoretical proofs to support the correctness of this claim. [...] I understand that this is a writing approach, but stating conclusions without strict proof is not appropriate.
>
> We recognize that, at this stage, our paper suggests this imprecisely, and that the suggestion of the reviewer is the correct thing to do. We want to assure that it is an imprecision due to writing and not intended to overstate claims. When talking about theoretical guarantees, we always mean that the available guarantees for other methods are worse than the ones we provide for GHBM.
>
> **We will correct lines 408-409, and will state that _“the current best convergence guarantees of FedCM do not prove an advantage of momentum in (cyclic) partial participation”_**, which is accurate and reflects the importance of solving this open question the reviewer highlighted. We will make sure that from the paper it is clear that we leave that interesting theoretical question open.
> Finally, we believe we can extend theorem B.9 to prove a lower bound for FedCM (not just for FedAvg) and we are working on it.
>
> **Connection with Incremental GD.**
> We appreciate the suggestion of the reviewer regarding drawing connection with existing literature in optimization outside FL. To the best of the author's knowledge, the term _incremental gradient method_ is used to refer to a broad family of algorithms in the context of finite-sum optimization of $n$ functions (similar to the FL setting we are in, where functions are clients). They aim at matching or surpassing the worst case performance of GD without requiring to evaluate the gradient at all the $n$ functions at each iteration, but only one for iteration.
> Incremental Gradient Descent (IGD) evaluates each of the $n$ functions cyclically and uses only the last gradient for updating the model. In practice, the difference with SGD is the deterministic order the $n$ functions are traversed.
>
> Similarly to IGD, Incremental Aggregated Gradient (IAG) uses the cyclic order when traversing the $n$ functions and evaluates only one gradient at each iteration, **but also maintains the average of the last $n$ gradients.**
>
> In this panorama, our GHBM is based on the similar intuition of IAG (i.e. approximating the full gradient via old gradients of each function), and connects to it in that, under proper choice of $\tau$, the momentum is updated considering all the client's gradients. The scenario is however fundamentally different, as we use this intuition to converge under unbounded heterogeneity in FL, where the analysis must take into consideration additional errors (e.g. the client drift).
> The use of established optimization concepts in centralized optimization to build powerful FL algorithms is not uncommon, as two of the classical FL algorithms, SCAFFOLD and FedDyn, respectively build on SAGA and ADMM.
>
> > This is because Incremental GD does not strictly require a complete pass through the dataset in one round, yet it can still achieve the elimination of randomness.
>
> IGD eliminates the randomness due to sampling the functions, but there is still an error due to the fact that the gradient used at each iteration has “incomplete” information w.r.t to the true gradient. **Proofs of IGD assume this error can be bounded by a constant, and such assumption exactly corresponds to the bounded heterogeneity assumption commonly used in FL.** As is, just enforcing a client sampling order, while removing the stochasticity due to sampling, does not remove the error that is due to having (highly) dissimilar functions (see section 3.2.1 of [5], in particular assumption 3.4).
>
> This motivates the need of a scheme like IAG, where we explicitly build an estimate of the global gradient and use it as opposed to using only the last one, when it is unreasonable to assume that the dissimilarity among the functions is not upper bounded.
>
> We hope this provides further insight of why just enforcing cyclic partial participation in FL is likely to be not enough to remove the need for the unbounded gradient dissimilarity assumption.
>
> If there are other approaches we should consider, please provide us some reference, we will be happy to add a discussion in the final revision of our manuscript.
>
> [5] Gurbuzbalaban Et Al., Convergence Rate Of Incremental Gradient And Incremental Newton Methods, SIAM Journal on Optimization 2022

---

### Official Review · Reviewer_GUBZ · 2024-10-30

**Soundness:** 3
**Presentation:** 3
**Contribution:** 3
**Rating:** 6
**Confidence:** 1

**Summary:**

This paper proposes propose a Generalized Heavy-Ball Momentum (GHBM), proving that it enjoys an improved theoretical convergence rate w.r.t. existing FL methods based on classical momentum in partial participation, without relying on bounded data
heterogeneity. This paper also presents FEDHBM as an adaptive, communication-efficient by-design instance of GHBM. Extensive experimentation on vision and language tasks, in both controlled and realistic large-scale scenarios, confirms our theoretical findings, showing that GHBM substantially improves the state of the art, especially in large scale scenarios with high data heterogeneity and low client participation.

**Strengths:**

1. The authors present a novel formulation of momentum called Generalized Heavy-Ball (GHBM) momentum, which extends the classical heavy-ball (Polyak, 1964), and propose variants that are robust to heterogeneity and communication-efficient by design.
2. The theoretical convergence rate of the proposed GHBM is given for non-convex functions, which extend the previous result of Cheng et al. (2024) of classical momentum, showing that GHBM converges under arbitrary heterogeneity even (and most notably) in partial participation.
3. The authors empirically show that existing FL algorithms suffer severe limitations in extreme non-iid scenarios and real-world settings. In contrast, FEDHBM is extremely robust and achieves higher model quality with significantly faster convergence speeds than other client-drift correction methods.

**Weaknesses:**

Although this paper is written very clearly, the Assumption 4.3 (Cyclic Participation) is important and should be further discussed in this paper.

**Questions:**

Although this paper is written very clearly, the Assumption 4.3 (Cyclic Participation) is important and should be further discussed in this paper.

---

> ### Author Response · Authors · 2024-11-21
>
> We thank the reviewer for the time spent reviewing our manuscript. Although we acknowledge that the reviewer admitted this submission may be outside their area of expertise, we hope our answer can provide the reviewer with a better understanding of our work.
>
> **On the use cyclic partial participation assumption**
> Assumption 4.3 simply states that there is a fixed order in which clients appear across rounds in the training, i.e. each client is sampled every $k=\frac{1}{C}$ rounds.
> Previous works have shown that if client availability is correlated with geographical location (e.g. mobile phones charging at night), then client availability follows a cyclic pattern [3,4].
> The best-known analysis of FedAvg under cyclic participation is provided by [1], which proves that in certain situations (e.g. clients run GD instead of SGD) there can be an asymptotic advantage in the case we prospect with assumption 4.3. However, it is important to notice that all the results presented in [1] rely on forms of bounded heterogeneity, and with this respect, the results presented in this work are novel and advance the state of the art.
>
> We hope our answer clarifies the reviewer's concern; if not, please let us know, and we are available for further discussion.
>
> [1] Cho et al., On the Convergence of Federated Averaging with Cyclic Client Participation, ICML 2023
>
> [2] Cheng et al., Momentum Benefits Non-IID Federated Learning Simply and Provably, ICLR 2024
>
> [3] Zhu et al., Diurnal Or Nocturnal? Federated Learning Of Multi-Branch Networks From Periodically Shifting Distributions, ICLR 2022
>
> [4] Crawshaw et al., Federated Learning under Periodic Client Participation and Heterogeneous Data: A New Communication-Efficient Algorithm and Analysis, NeurIPS 2024

---

> > ### Comment · Reviewer_GUBZ · 2024-12-02
> >
> > Thank you for the detailed response. I keep my score.

---

### Official Review · Reviewer_Zb5t · 2024-11-03

**Soundness:** 2
**Presentation:** 3
**Contribution:** 2
**Rating:** 3
**Confidence:** 3

**Summary:**

The paper proposes new federated learning (FL) algorithm variants—GHBM, LocalGHBM, and FedHBM—which generalize previous momentum-based methods by allowing the momentum term to be computed using multiple past iterations, rather than just the most recent one. The authors theoretically prove that the iterates of GHBM, with access to a stochastic gradient oracle, converge to a first-order stationary point in a mean-squared sense for non-convex functions, even with partial client participation and arbitrary client dissimilarity. Experimentally, they demonstrate that for large neural network training, their methods outperform existing FL approaches, especially in scenarios with high client dissimilarity and low client participation.

**Strengths:**

The paper advances previous theoretical results that relied on assumptions of bounded client heterogeneity or full client participation. The authors’ method generalizes the Fed-Avg-M approach from Cheng (2024), achieving the same convergence rate without requiring full client participation. Experimental results are compelling, with the proposed methods showing significant improvements over existing methods used for comparison.

**Weaknesses:**

The main weakness of the paper’s contribution is the similarity to (Cheng 2024). The theoretical novelty is limited as the authors admit the analysis is based on (Cheng 2024). Further, this paper compares against Fed-Avg-M from (Cheng 2024), but makes no mention of the SCAFFOLD-M algorithm from the same paper which has convergence guarantees in the desired setting of unbounded client heterogeneity and partial client participation. SCAFFOLD-M incurs a slightly worse convergence rate but does not require cyclic client participation. The line in the introduction section that reads“existing momentum-based FL methods still theoretically rely on bounded heterogeneity in partial participation…” is thus misleading. The absence of discussion of SCAFFOLD-M in terms of theoretical guarantees and experimental performance is a major weakness, especially since (Cheng 2024) showed that SCAFFOLD-M performed much stronger than Fed-Avg-M under severe client heterogeneity.

Although the experimental results are strong relative to the other methods chosen for comparison, the discussion on why the proposed methods significantly outperform existing methods is unsatisfying. It is mentioned throughout the paper that existing methods perform poorly in the “large-scale” setting and the proposed methods improve in this area, but this difference does not seem to relate to the theoretical convergence results established in the paper.

**Questions:**

Can you compare your theoretical and experimental results to SCAFFOLD-M? It seems to be the most natural algorithmic competitor to the proposed methods. Also, can you offer an explanation as to why the proposed methods outperform existing methods in the “large-scale” setting especially? This point is made many times but does not seem to be supported by the theoretical discussion. Are the proposed methods just converging faster than other methods, or is there something about using the generalized momentum method that biases the networks to learn better quality solutions?

---

> ### Author Response · Authors · 2024-11-21
> **Response 1/2**
>
> We thank the reviewer for the time spent reviewing our manuscript. In the following, we respond pointwise to the reported weaknesses (W) and questions (Q).
>
> **W1: On the novelty and relationship with [1]**
> The work of [1] proposes a new theoretical analysis of existing momentum-based approaches (FedCM) and their integration with SCAFFOLD. Their core contribution is proving that FedCM (or as called in [1], FedAvg-M), under full participation, converges under arbitrary heterogeneity; their work itself does not propose a new algorithm, but the integration of momentum with SCAFFOLD and other forms of variance reduction.
> Conversely, our work proposes the novel GHBM formulation, whose motivations exactly lie in the limitations of FedCM, namely that the classical momentum itself is not enough to converge under arbitrary heterogeneity in partial participation. The core difference with classical momentum is explained from the ground up, precisely tackling the theoretical reason why FedCM does not achieve the same result in partial participation, as its momentum term is updated with a biased estimate of the global direction (section 3.3). We then provide a theoretical analysis that builds upon the result of [1] and extends it by considering our momentum formulation instead of the classical one.
> With these premises, the fact that we build on existing knowledge and extend previous theories does not affect the novelty of our work.
> On the contrary, by employing similar theoretical tools as [1], we clearly demonstrate the advancements of our new algorithm and highlight the specific improvements achieved, by enabling a result that is unattainable by FedCM.
>
> **W2-Q1: On the comparison with SCAFFOLD-M**
>
> Our algorithm is the first that can use only momentum to converge under arbitrary heterogeneity in partial participation. Indeed, proving that such a result is possible by using only momentum is the main point of our theory. We did not compare with SCAFFOLD-M because, as SCAFFOLD, SCAFFOLD-M is based on variance reduction to overcome the dependence on bounded heterogeneity, i.e. SCAFFOLD-M converges under arbitrary heterogeneity thanks to SCAFFOLD’s variance reduction, not because it uses momentum.
> As is, our claim that “existing momentum-based FL algorithms still rely on bounded heterogeneity in partial participation…” refers to momentum-only FL algorithms. We acknowledge that we can be more precise on that point, so we added this discussion to our revision.
>
> From the practical perspective, there are several reasons why GHBM is preferable to SCAFFOLD-M, especially in settings with very low participation. The first one is the algorithmic structure: SCAFFOLD-M requires clients to keep local client states across rounds, a limitation that is known for making the algorithm not well-suited for cross-device FL [2]. The second one is related to actual empirical performance: there is extensive evidence that, despite the strong theoretical properties, variance reduction does not perform well in deep learning [3]. Indeed, performance issues of SCAFFOLD have been noticed in previous work [2] as well as in ours. In particular, it does not work in deep learning settings with both high heterogeneity and low participation (see Table 3). As is, building on the effectiveness of momentum in deep learning, our work significantly improves on the use of momentum in FL, showing that it can both give similar guarantees to SCAFFOLD(-M) and finally improve by a large margin the performance on realistic scenarios.
>
> However, we do agree that an experimental comparison with SCAFFOLD-M could be useful to the reviewer to evaluate the improvement we bring with this work. In the following, we simulate the same setting of Table 2 for CIFAR-100 w/ Resnet20, $C=10\\%$ and $\alpha \in \\{0, 0.1, 0.3\\}$. As it is possible to notice, SCAFFOLD-M does not significantly improve SCAFFOLD, and most notably has much lower performance than GHBM.
>
> | Method | Final Test acc. (%) - $\alpha=0$ | Final Test acc. (%) - $\alpha=0.1$ | Final Test acc. (%) - $\alpha=0.3$ |
> |---|---|---|---|
> | SCAFFOLD | $30.7_{\pm 1.3}$ | $51.0_{\pm 0.4}$ | $52.8_{\pm 0.4}$ |
> | SCAFFOLD-M | $30.9_{\pm 0.7}$ | $51.5_{\pm 0.3}$ | $54.0_{\pm 0.2}$ |
> | GHBM | $38.5_{\pm 1.0}$ | $54.1_{\pm 0.2}$ | $56.9_{\pm 0.1}$ |

---

> ### Author Response · Authors · 2024-11-21
> **Response 2/2**
>
> **W3-Q2: On why our GHBM largely outperforms the state of the art**
> We have explained in section 3.3 that the key intuition behind GHBM is updating the momentum term, at each round, as it would be updated in full participation, that is with the average gradient over all clients. To do so in a way compliant with partial participation, GHBM uses past server pseudo-gradients (eq. 4) instead of (unavailable) gradients at current parameters (eq. 3). This is why we are able to obtain the same theoretical result [1] gets for FedCM under full participation, as GHBM approximates the same kind of update. Indeed, our analysis in Figure 1 demonstrates that this strategy is very effective since it drastically reduces the estimation error for the update of the momentum term.
> Our theory keeps track of the error induced by using past gradients in the momentum updates, and this aspect is different and expands upon the one proposed in [1].
>
> The empirical improvements we get in large-scale scenarios relate to the theoretical convergence rate established because those considered settings are characterized by (i) high heterogeneity and (ii) very low client participation (indicated directly in Table 3), which are exactly the aspects the algorithm is expected to improve upon by looking at the theory.
>
> We agree that understanding where our improvement comes from is crucial to correctly evaluate the contribution of our work, so please let us know if there are any more concerns on this matter.
>
> **Q2: Is GHBM just faster or does it also learn better solutions?**
> This is a very interesting question. In all our experiments we noticed particularly faster convergence over other methods, not only in non-iid settings where the effect is most noticeable but also in iid settings. The theory we propose does explain that speedup, even in the iid case, where we expect the (local) stochastic noise $\sigma$ to affect the optimization. In all our experiments we also notice that the model trained with GHBM reaches at convergence higher accuracy w.r.t other methods, e.g. LocalGHBM (as GHBM) reaches 62.0% accuracy in Table 2 (ResNet20, iid) instead of the 60.5% of MimeMom (best competitor) despite both having long reached their convergence point. Based on these results, we conjecture that it is possible that GBHM promotes learning better solutions, but we do not currently have any theoretical indications of why it should happen, so we leave this question open for future work.
>
> [1] Cheng et al., Momentum Benefits Non-IID Federated Learning Simply and Provably, ICLR 2024
>
> [2] Reddi et al., Federated Adaptive Optimization, ICLR 2021
>
> [3] Defazio et al., On the Ineffectiveness of Variance Reduced Optimization for Deep Learning, NeurIPS 2019

---

> > ### Comment · Reviewer_Zb5t · 2024-11-25
> >
> > **W1: On the novelty and relationship with [1]**
> >
> > Thank you for clarifying the difference between your contribution and that of [1]. As I understand, the main technical contribution of your work is showing that the proposed algorithm which looks back $tau$ steps when calculating momentum rather than just 1 step as in FedCM can overcome unbounded client heterogeneity under partial participation, whereas FedCM cannot handle partial participation.
> >
> > **W2-Q1: On the comparison with SCAFFOLD-M**
> >
> > Thank you for explaining the rationale behind not comparing with SCAFFOLD. However, I am still unsure of your statement “SCAFFOLD-M converges under arbitrary heterogeneity thanks to SCAFFOLD’s variance reduction, not because it uses momentum”. In [1], they mention that only after introducing momentum to SCAFFOLD does it converge under arbitrary heterogeneity (Section 1.1), so it seems like momentum is a key characteristic for handling client dissimilarity. Can you point out where in the paper you have revised your claim “existing momentum-based FL algorithms still rely on bounded heterogeneity in partial participation…?”
> >
> > Although you mention practical and empirical reasons for why SCAFFOLD-based methods may be undesirable, the direct comparison should be discussed when presenting your theoretical results. The difference in memory requirements between your proposed method and SCAFFOLD-M can be stated as you point out SCAFFOLD-M requires extra client storage, but this does not warrant completely omitting SCAFFOLD-M completely when discussing existing methods and their limitations in Table 1 and Section 3.2, especially given your central claim that existing methods are unsatisfactory.
> >
> > Thank you for providing additional experimental results. These are quite surprising, especially the lack of improvement between SCAFFOLD to SCAFFOLD-M, as Figure 2(b) in [1] shows that in the partial participation and severe client heterogeneity regime that SCAFFOLD is much worse than SCAFFOLD-M. Can you explain this discrepancy? In any case, these additional results should be added to Table 2.
> >
> > **W3-Q2: On why our GHBM outperforms the state of the art**
> >
> > Thank you for this explanation. As I understand, since the methods in comparison do not guarantee convergence under the partial participation and severe client heterogeneity regime, the proposed method outperforms them. The only other method that theoretically handles this regime is SCAFFOLD-M which based on your provided results still performs poorly. I think the failure of SCAFFOLD-M relative to your proposed method warrants much more explanation. Given the similarity of GHBM to FedCM (GHBM is a generalization as you mention in Section 3.3) and your explanation that GHBM recovers the same theoretical result of FedCM but with partial participation,, it would be useful in Table 2 to include a baseline result of FedCM under full participation to see if GHBM can recover similar performance by only sampling clients at each round. This would make the theoretical connections to the experiments much stronger and would make your contribution clearer.
> >
> > **Q2: Is GHBM just faster or does it learn better solutions?**
> >
> > Thank you for touching on this point. It is fair to leave this for future work.
> >
> >
> > Overall: There are still open questions, so I will leave the scores as is for now and wait for your responses.

---

> > > ### Author Response · Authors · 2024-11-28
> > > **Response 1/2**
> > >
> > > **W1: we are glad our novelty and contribution is now clear.**
> > > The reviewer is right in identifying the core theoretical contribution of our work. We are glad that the novelty and the contribution have been fairly recognized.
> > >
> > > **W2: additional notes on the comparison with SCAFFOLD-M**
> > >
> > > > _SCAFFOLD-M converges under arbitrary heterogeneity thanks to SCAFFOLD’s variance reduction, not because it uses momentum_
> > >
> > > We confirm that this statement is correct, SCAFFOLD-M inherits this property from SCAFFOLD, which itself already converges in partial participation under arbitrary heterogeneity [4], and does not need momentum to achieve this result. Indeed, in Table 2 of Cheng et al. [1], both SCAFFOLD and SCAFFOLD-M do not additionally require the bounded heterogeneity assumption. The advantage of using momentum is instead related to accelerated convergence rate and not to heterogeneity reduction effect, as discussed in the paragraph _Comparison with SCAFFOLD_ in section 4.1 of [1].
> > >
> > > > Can you point out where in the paper you have revised your claim “existing momentum-based FL algorithms still rely on bounded heterogeneity in partial participation…?”
> > >
> > > In the first revision, we changed lines 53-54 in the main paper and added a paragraph with a more detailed explanation in the new section A.1, lines 769-776 (all changes are colored in violet).
> > >
> > > > the direct comparison [with SCAFFOLD-M] should be discussed when presenting your theoretical results.
> > >
> > > Thanks for the suggestions, in the last revision we added SCAFFOLD-M to Table 1 as well as a discussion in section 4.3, right after the comparison with FedCM.
> > >
> > > > Thank you for providing additional experimental results. These are quite surprising, especially the lack of improvement between SCAFFOLD to SCAFFOLD-M, as Figure 2(b) in [1] shows that in the partial participation and severe client heterogeneity regime that SCAFFOLD is much worse than SCAFFOLD-M. Can you explain this discrepancy?
> > >
> > > Thanks for the question. The reason most likely lies in the different experimental settings of [1] w.r.t. ours. The experiments in Figure 2-b in [1] are conducted by splitting the dataset into $K=10$ clients, and selecting $C=20\\%$ of them at each round, while our setting involves $K=100$ clients, with a participation ratio of $C=10\\%$. Moreover, we provided results on CIFAR-100, which is much harder to improve upon than CIFAR-10, especially in heterogeneous scenarios.
> > >
> > > From the algorithmic perspective, a higher participation ratio helps avoid SCAFFOLD’s control variates becoming stale, as they get updated more frequently. We expect any setting with a lower number of clients and higher participation to benefit SCAFFOLD-based methods. Indeed, the staleness of control variates has been recognized as a limitation for the empirical performance of SCAFFOLD [2,5], as well as for the general ineffectiveness of variance reduction in deep learning [3].
> > >
> > > > In any case, these additional results [SCAFFOLD-M] should be added to Table 2.
> > >
> > > Thanks for pointing this out, we added full results for SCAFFOLD-M in Table 2 in the last revision.

---

> > > ### Author Response · Authors · 2024-11-28
> > > **Response 2/2**
> > >
> > > **W3-Q2: On why our GHBM outperforms the state of the art**
> > >
> > > > I think the failure of SCAFFOLD-M relative to your proposed method warrants much more explanation.
> > >
> > > This is a very interesting observation. An explanation is given above for the provided additional experiments: client participation has a big impact on the staleness of SCAFFOLD(-M) control variates, which in turn reflects on performance [2,5].
> > > Besides this empirical observation, we shall see that the matter is much more profound and relates to the underlying theory used to handle heterogeneity: variance-reduction for SCAFFOLD(-M) and (Generalized) momentum for GHBM. As such, this concern really touches upon a more fundamental question in deep learning: why does momentum work better than variance reduction? The answer to this (open) question is certainly of much broader scope than FL, and for this reason, it is left out of this work.
> > >
> > > > it would be useful in Table 2 to include a baseline result of FedCM under full participation to see if GHBM can recover similar performance by only sampling clients at each round.
> > >
> > > Thanks for pointing this out, that is great advice. We added complete results for FedCM in full participation in our last revision. The difference in final model quality is nearly negligible in all cases except for CIFAR-100, in the non-iid setting. This is motivated by the algorithm being still far from the convergence point in the given round budget, and by the error introduced by using past gradients (see discussion in Sec. 3.3).
> > >
> > > We would also like to point to the additional theoretical experiments (section C.9) we added in the first revision, comparing the convergence rate of GHBM and FedCM in cyclic participation. Those results demonstrate that GHBM in cyclic participation closely matches the convergence of FedCM in full participation, confirming our theoretical results.
> > >
> > >
> > > [4] Karimireddy et al., SCAFFOLD: Stochastic Controlled Averaging for Federated Learning, ICML 2019
> > >
> > > [5] Karimireddy et al., Breaking the centralized barrier for cross-device federated learning, NeurIPS 2021

---

### Official Review · Reviewer_f1jY · 2024-11-04

**Soundness:** 3
**Presentation:** 3
**Contribution:** 3
**Rating:** 6
**Confidence:** 3

**Summary:**

This paper introduces a novel Generalized Heavy-Ball Momentum (GHBM) technique for federated optimization with partial participation, enhancing convergence without relying on the bounded heterogeneity assumption. The proposed method is both theoretically and experimentally shown to improve convergence rates. Empirical results across various federated learning benchmarks demonstrate benefits over previous approaches.

**Strengths:**

1. The paper tackles the important problem of federated learning optimization in heterogeneous settings and under partial participation.

2. The proposed technique builds on the classical momentum mechanism in a straightforward manner, making it promising for practical implementation in federated learning systems.

3. The authors conduct diverse experiments showing the advantages of GHBM (and variations) across multiple realistic computer vision and language tasks.

**Weaknesses:**

**W1.** The **literature overview** in Subsection 3.2 omits some recent works, such as ScaffNew [1] and related papers, that achieve accelerated communication complexity in heterogeneous settings through control variates similar to SCAFFOLD. In addtion, there are recent papers on federated optimization [2, 3], which leverage momentum as a local correction term to benefit from second-order similarity (such as MIME and CE-LSGD).

**W2.** On **presentation and clarity.** Many equation links, particularly those referencing the appendix (e.g., Eq. (122)), are not functional, making the paper cumbersome to follow.
The symbol $G$ is used without definition in Lemma 4.4.
Referring to notations introduced in the appendix, such as equations (11) and (12), makes understanding Lemma 4.6 challenging without flipping back and forth between sections.

A clear, intuitive explanation of Assumption 4.3 would be beneficial to the reader.

**W3.** The local version (LocalGHBM) may not be well-suited for cross-device settings, as it necessitates storing local states similarly to SCAFFOLD, which has been identified as a limitation by Reddi et al. (2021).


The following work has already been published:

> Konstantin Mishchenko, Eduard Gorbunov, Martin Takác, and Peter Richtárik. Distributed learning with compressed gradient differences, 2019.
___

[1] Mishchenko, Konstantin, et al. “Proxskip: Yes! local gradient steps provably lead to communication acceleration! finally!.” International Conference on Machine Learning. PMLR, 2022.

[2] Mishchenko, Konstantin, et al. “Federated learning under second-order data heterogeneity.” https://openreview.net/forum?id=jkhVrIllKg, 2023.

[3] Karagulyan, Avetik, et al. “SPAM: Stochastic Proximal Point Method with Momentum Variance Reduction for Non-convex Cross-Device Federated Learning.” arXiv preprint, 2024.

**Questions:**

1. What is actually the problem being tackled in this paper: cross-device or cross-silo?

2. The results in Figure 1 are somewhat counterintuitive, showing higher variation for the “iid” case. Could the authors clarify this observation?

3. The work by Patel et al., (2022) suggested an algorithm CE-LSGD and provided some lower bounds, which seem to be circumvented by this submission. Please explain how is it possible? Please also provide a comparison to that paper.

4. How important is the cyclic participation Assumption 4.3 for achieving the main theoretical contribution of the paper?

5. Do the convergence rates benefit from introducing local steps, as seen in ScaffNew? From equation (122), it seems that the local step size $\eta_l$ must decrease inversely with the number of local steps $J$.

6. How does GHBM interact with server momentum, which has been shown to be essential for achieving good convergence rates?

---

> ### Author Response · Authors · 2024-11-23
> **Response 1/2**
>
> We thank the reviewer for the time spent reviewing our manuscript. In the following, we respond pointwise to the reported weaknesses (W) and questions (Q).
>
> **W1-Q3: On literature overview and missing comparison with CE-LSGD**
> Thanks for pointing out those related works, due to space constraints we added a discussion on them in section A.1, but will integrate it into the main paper after the discussion phase. The CE-LSGD and Mime are in the "streaming" or "stochastic" setting where there are potentially an infinite number of clients. In our work, we only focus on settings with a finite number of clients (similar to Scaffold and ScaffNew). This is why the lower bound results of Patel et al. 2022 do not apply to us. We also note that, unlike Scaffold or Scaffnew, our algorithm does not require the clients to store any state, meaning it is easier to implement in practice.
>
> **W2: On enhancing presentation**
> We appreciate the reviewer’s feedback on how to improve the readability. We fixed all the equation links and reported in the main text the definitions in eq. (11) and (12) to avoid requiring the reader to go back and forth to grasp the meaning of lemma 4.6. The symbol $G$ has been introduced in the caption of Table 1, but we do agree with the reviewers and will report the definition also in Lemma 4.4
>
> **Intuitive explanation of Assumption 4.3**
> Thanks for the suggestion, assumption 4.3 simply states that there is a fixed order in which clients appear across rounds in the training. For example, if $K=10$ clients $i \in [1,K]$ are present in the system and $C=20\%$ of them participate at each round, a sequence like $\{(1,2), (3,4), (5,6), (7,8), (9,10), (1,2), (3,4), …\}$ satisfies the assumption. It is easy to notice that the period between two subsequent sampling of any client is fixed and it is equal to $\frac{1}{C}$, i.e. the inverse of the participation frequency. We will add a graph in the supplementary for an immediate visual understanding of this pattern.
>
> **W3-Q1: On the FL settings our algorithms tackle**
> Thanks for the question, as it allows us to clarify that the general algorithm proposed in this work (GHBM) is designed for settings closer to cross-silo than to cross-device FL. However, we need to distinguish between what we can offer as theoretical guarantees and the expected empirical performance of our methods in realistic scenarios.
>
> From the **theoretical standpoint**, strong guarantees on statistical heterogeneity in **partial participation** come with the constraint that we can see each client multiple times (i.e. the $\tau=\frac{1}{C}$ constraint in theorem 4.7), which makes it an algorithm more suitable for settings with limited number of clients that participate often, as it is in cross-silo FL. In this scenario, GHBM is the sole momentum-only FL algorithm that can achieve this result.
>
> In practice, GHBM can also work in settings more akin to cross-device FL (e.g. the one considered in Table 3) since it is a stateless algorithm and the momentum calculation (i.e. the $\tau$ hyperparameter) can be controlled server-side. Indeed, from the **empirical perspective**, we observed that GHBM is currently the best algorithm in realistic scenarios akin to cross-device FL (e.g. the one considered in Table 3) with even extremely low participation (e.g. $C<1\%$ and the total number of clients on the order of $10^4-10^5$).
>
> LocalGHBM is proposed as a more communication-efficient variant of GHBM, that leverages stateful clients to avoid exchanging additional data (i.e. the momentum term). As such, is not intended and it is not well-suited for settings akin to cross-device, as the reviewer correctly notices.
>
> We will be more clear on these aspects in our revision.
>
> **Q2: Clarification about Figure 1**
> Thank you for pointing this out. The caption of Figure 1 contains a typo that is at the base of your (correct) concern: the caption should be “... in **non-iid** ($\alpha=0$, left) and **iid** ($\alpha=10k$, right) …”, i.e. $\alpha=0$ corresponds to non-iid and $\alpha=10k$ to iid, since low values of $\alpha$ indicate more heterogeneous splits, while higher values indicate lower heterogeneity (see section C.2 for additional details).
> Therefore, the presented results align with the reviewer’s observation - there is a higher variance in the non-iid scenario than iid. Apologies for the confusion, we fixed the typo.

---

> ### Author Response · Authors · 2024-11-23
> **Response 2/2**
>
> **Q4: About our use of cyclic participation assumption 4.3**
> The cyclic participation assumption (4.3) is used to determine how many different clients are being selected in $\tau$ subsequent rounds and, consequently, the gradients averaged when updating the momentum with the GHBM rule. It is used essentially to ease the proof because when sampling at random it can happen that the same clients are selected more than once in a given window of $\tau=\frac{1}{C}$ rounds, while some others may not be sampled.
> This is related to our main theoretical contribution in that, for achieving convergence under arbitrary heterogeneity, the proof technique we adopted requires that the momentum is updated with a gradient over all the clients, as it happens in full participation. We want to highlight that the assumption alone is not sufficient to enhance the guarantees of FedCM in partial participation: the reason behind the better result is the structure of GHBM update.
>
> Let us also note that the assumption impacts only the theoretical analysis; in practice, our experiments employ random sampling, and the results are nearly identical to those obtained under cyclic participation. Extending the analysis to account for random client sampling would likely require the development of new theoretical tools, which we leave as a direction for future work.
>
> **Q5: About convergence rate and local steps**
> As the reviewer correctly notes, our analysis does not show a benefit in the convergence rate from the local steps $J$, as ScaffNew does. In this regard, our proof shares the same result as the currently best-known theory for momentum-based FL methods [1].
> The main goal of our analysis is to show that GHBM can converge under arbitrary heterogeneity and with partial participation, a result previously achievable by FedCM only with full client participation. Improving the analysis with respect to the speedup of local steps is an interesting direction we leave as future work.
>
> **Q6: GHBM and server momentum**
> Thanks for the interesting question! GHBM uses (global) momentum on the client side and does not apply any server-side momentum. This is in line with recent momentum-based FL algorithms, such as FedCM [2] and MimeLiteMom [3], and previous work proving that it is important to incorporate client-drift correction locally at clients [3,4], especially in highly heterogeneous scenarios.
>
> However, to precisely answer the reviewer, we added section C.8 in the supplementary material with initial experiments not originally included in the submission, showing that GHBM used at the client level always outperforms its server-side variant.
>
> [1] Cheng et al., Momentum Benefits Non-IID Federated Learning Simply and Provably, ICLR 2024
>
> [2] Xu et al., FedCM: Federated Learning with Client-level Momentum, arXiv 2021
>
> [3] Karimireddy et al., Breaking the centralized barrier for cross-device federated learning, NeurIPS 2021
>
> [4] Karimireddy et al., SCAFFOLD: Stochastic Controlled Averaging for Federated Learning, ICML 2019
>
> [5] Hsu et al., Measuring the Effects of Non-Identical Data Distribution for Federated Visual Classification

---

> > ### Comment · Reviewer_f1jY · 2024-11-25
> >
> > I would like to thank the authors for addressing my questions and updating the paper.
> >
> > > The CE-LSGD and Mime are in the "streaming" or "stochastic" setting where there are potentially an infinite number of clients.
> >
> > > This is why the lower bound results of Patel et al. 2022 do not apply to us.
> >
> > As far as I can see, the CE-LSGD paper states that they consider a finite-sum problem on the first page (equation (1.1)) and later extend the results to a "stochastic" setting. Could you please elaborate on these points as currently, I can not see your response as correct.
> >
> > In light of the comments of Reviewer KZ3z, it also seems worth discussing the connections of the proposed Cyclic Participation to incremental and random reshuffling sampling-based methods (such as Malinovsky et al., 20223).
> >
> > ___
> >
> > Malinovsky, Grigory, et al. "Federated learning with regularized client participation." arXiv preprint arXiv:2302.03662 (2023).

---

> > > ### Author Response · Authors · 2024-11-29
> > >
> > > **Discussion about lower-bounds in CE-LSGD paper [6].**
> > > Thank you for the remark. Authors of [6] provide two different lower bounds, for two different classes of learning problems: one corresponds to the “finite-sum” setting and the other to the “stochastic” or “streaming” setting.
> > > * Theorem 2.1 (page 5) proves a lower bound for $\mathcal{F}_M^1, \mathcal{F}_M^2$ (definitions 1-2), where the global function $F$ is expressed as finite-sum of $M$ objectives (eq. 1.1). **This is the setting of our algorithm, and our rate does not circumvent the lower bound.**
> > > * Theorem 3.4 (page 9) instead proves a lower bound for $\mathcal{F}_P^1$, $\mathcal{F}_P^2$, (definitions 6-7), where $M$ clients are sampled at each round from a distribution $\mathcal{P}$, rather than from a finite set, accounting for possibly infinite clients. As such, that result does not apply to our setting (i.e. cyclic participation is not possible in that context).
> > >
> > > Please let us know if there are any more concerns on this matter.
> > >
> > > **Discussing the connections between cyclic participation and [7].**
> > > Thank you for bringing to our attention the paper [7].
> > > The cyclic participation assumption we use in our theoretical results matches the client participation scheme of [7], where $M$ clients are sampled in groups of $C$ cyclically at each round. Differently from [7], we use this rule to determine the contributions in the generalized momentum update, whereas [7] studies FedAvg under cyclic participation and random data reshuffling. In particular, the authors prove that FedAvg with cyclic participation has a faster rate than random client sampling - but it still depends on the client heterogeneity (though they use a weaker definition of heterogeneity since they seem to only analyze convex functions). Conversely, our work proves that, under the same cyclic participation assumption, our generalized formulation of momentum (GHBM) can guarantee convergence under unbounded client heterogeneity.
> > >
> > > Should you have more questions, we remain available for further clarification. We will include this discussion in the main paper.
> > >
> > > [6] Patel et al., Towards Optimal Communication Complexity in Distributed Non-Convex Optimization, NeurIPS 2022
> > >
> > > [7] Malinovsky et al., Federated Learning with Regularized Client Participation, arxiv

---

### Author Response · Authors · 2024-12-04
**Global message after authors-reviewers discussion period**

We thank the reviewers for all the suggestions and comments, in this global message we want to highlight the main points of our rebuttal and discussion, and summarize what changes have been made to the original manuscript.

---
# Contribution and Significance
The main contribution of this work is the GHBM algorithm, **a novel, principled momentum-based FL algorithm, with provable convergence guarantees under arbitrary heterogeneity in (cyclic) partial participation and significant empirical improvement of performance in various scenarios and heterogeneity levels.**

In particular, we clarified that **GHBM is the first algorithm that can use only momentum to converge under arbitrary heterogeneity in cyclic partial participation**. This is our theoretical contribution, as the **best known previous work on (classical) momentum in FL proves the same result only in full participation**.

**This result follows from the key novelty in the design of GHBM**: the momentum is updated with the average of the last $\tau$ pseudo-gradients such that, when $\tau=\frac{1}{C}, in cyclic participation the update is similar to the one classical momentum has in full participation. This is validated by additional experiments provided in section C.9, where **GHBM in partial participation is shown to behave similarly to FedCM in full participation**.

We believe this is a **significant contribution to the field**, as it expands the understanding of the effectiveness of momentum techniques in FL.

---
# Changes on the manuscript

### On the use of cyclic participation assumption.
We clarified that **the use of cyclic participation assumption is legitimate for GHBM** because, while it is needed for proving our theoretical result (as explicitly indicated in Table 1), in our analysis the convergence under unbounded heterogeneity comes explicitly from the algorithmic structure of GHBM (i.e. setting $\tau=\frac{1}{C}$ is necessary).
While we leave as an open problem proving a lower bound for FedCM, its best known analysis in [1] cannot prove a similar result in a context of partial participation. Moreover, **all the experimental results we present strongly indicate that we should expect FedCM to be inevitably affected by heterogeneity in partial participation**. In the final revision **we will make it clear that it is still open to formally prove.**

### Comparison with other methods.
Our work mainly compares with state-of-art FL algorithms that use only momentum to deal with heterogeneity, as the point of our paper is proving that (our generalized) momentum can extend strong convergence guarantees in cyclic participation.

As is, we initially left out of our study SCAFFOLD-M [1], which relies on SCAFFOLD controls variates to solve the issues of heterogeneity. Following the suggestion of Reviewer Zb5t, in the last revision **we added both theoretical and experimental comparison with SCAFFOLD-M**, showing that in difficult scenarios it inherits the empirical weaknesses of SCAFFOLD.

### Practical implications
In our work, we directly demonstrate the effectiveness of GHBM in **large-scale real-world scenarios**, by conducting extensive experimentation on challenging settings characterized by a **large number of clients** (on the order of $10^4$-$10^5$) and **very low participation** (e.g. $C<1\\%$). We clarified that, even if the theoretical guarantees are provided in cyclic participation, **all the experiments are carried under the common random uniform sampling**, to guarantee a fair comparison with other algorithms that do not require that assumption.
Together with strong empirical performance, **the simplicity of GHBM makes it  preferable to more complex alternatives** and further corroborates its value for practical application in FL systems.

**Other modifications.**
Other modifications include the correction of the caption of Figure 1, where we mistakenly wrote “iid” for the case of $\alpha=0$ and “non-iid” for the case of $\alpha=10.000$, as it is obviously the opposite. We want to clarify that **the presented results align with the reviewers’ expectations.**
As other modifications, we fixed all the non-functioning equation links and improved readability by reporting the definitions of the involved quantities lemmas 4.4 and 4.6.

---
# Discussion outcome
We thank the reviewers for engaging in discussion with us. Their valuable feedback has allowed us to enhance the overall quality of our paper.
Based on the most recent interactions with each reviewer, we believe we have adequately addressed all outstanding concerns that initially led to some negative scores.

**In light of our efforts to address these concerns, including providing new experiments, clarifications, and discussions of related work, we kindly encourage the reviewers to reconsider the significance and potential impact of our submission in the next discussion phase.**

[1] Cheng et al., Momentum Benefits Non-IID Federated Learning Simply and Provably, ICLR 2024

---

### Meta-Review · Area_Chair_kpF8 · 2024-12-22

**Metareview:**

The paper shows the benefits of computing momentum based on longer history in the FL setting, under heterogeneity and cyclic client participation. Overall, the results are mostly clearly presented and the assumptions clearly stated, especially after the discussion phase. The concerns from the reviewers primarily stem from the similarities with certain previous analysis and limited expiation of why the empirical results are better than closely related methods. The authors have provided detailed responses clarifying certain aspects, however some concerns remained. The concerns regarding the participation assumptions were discussed at length and while it can considered somewhat strong, it is arguably close to where sota analysis is.

**Additional Comments On Reviewer Discussion:**

The reviewers mostly engaged with the authors and there were active discussions which clarified several aspects of the work.

---

### Decision · Program_Chairs · 2025-01-22

Reject